# CD4+ T cell-induced inflammatory cell death controls immune-evasive tumours

Bastian Kruse[1,15], Anthony C. Buzzai[1,15], Naveen Shridhar[1,15], Andreas D. Braun[1,15], Susan Gellert[1], Kristin Knauth[1], Joanna Pozniak[2,3], Johannes Peters[1], Paulina Dittmann[1], Miriam Mengoni[1], Tetje Cornelia van der Sluis[1], Simon Höhn[1], Asier Antoranz[4], Anna Krone[5], Yan Fu[5], Di Yu[6], Magnus Essand[6], Robert Geffers[7], Dimitrios Mougiakakos[8], Sascha Kahlfuß[5], Hamid Kashkar[9], Evelyn Gaffal[1], Francesca M. Bosisio[10], Oliver Bechter[11], Florian Rambow[12,13], Jean-Christophe Marine[2,3], Wolfgang Kastenmüller[14], Andreas J. Müller[5,16] & Thomas Tüting[1,16 ✉]

Most clinically applied cancer immunotherapies rely on the ability of CD8+ cytolytic T cells to directly recognize and kill tumour cells[1–3]. These strategies are limited by the emergence of major histocompatibility complex (MHC)-deficient tumour cells and the formation of an immunosuppressive tumour microenvironment[4–6]. The ability of CD4+ effector cells to contribute to antitumour immunity independently of CD8+ T cells is increasingly recognized, but strategies to unleash their full potential remain to be identified[7–10]. Here, we describe a mechanism whereby a small number of CD4+ T cells is sufficient to eradicate MHC-deficient tumours that escape direct CD8+ T cell targeting. The CD4+ effector T cells preferentially cluster at tumour invasive margins where they interact with MHC-II+CD11c+ antigen-presenting cells. We show that T helper type 1 cell-directed CD4+ T cells and innate immune stimulation reprogramme the tumour-associated myeloid cell network towards interferon-activated antigen-presenting and iNOS-expressing tumouricidal effector phenotypes. Together, CD4+ T cells and tumouricidal myeloid cells orchestrate the induction of remote inflammatory cell death that indirectly eradicates interferon-unresponsive and MHC-deficient tumours. These results warrant the clinical exploitation of this ability of CD4+ T cells and innate immune stimulators in a strategy to complement the direct cytolytic activity of CD8+ T cells and natural killer cells and advance cancer immunotherapies.

Adoptive cell transfer (ACT) studies using tumour-infiltrating lymphocytes from patients that are expanded ex vivo before their reinfusion provided initial proof-of-principle for the clinical efficacy of T cell immunotherapy[11]. The recent success of the immune checkpoint blockade (ICB) with monoclonal antibodies targeting the immunoregulatory receptors CTLA4 and PD1 led to the clinical breakthrough of T cell-directed immunotherapies[12]. The efficacy of ICB is mainly attributed to reactivation of CD8+ T cells that specifically recognize tumour antigens in the form of processed peptide epitopes presented by major histocompatibility complex class I (MHC-I) molecules on tumour cells. Both antigen presentation and MHC expression are upregulated by interferons (IFNs). Following antigen recognition, CD8+ T cells release cytolytic granules and IFNγ

that initiate cell death. Despite its clinical efficacy, ICB is limited by the emergence of MHC-deficient and IFN-unresponsive tumour cell clones that escape recognition and destruction by CD8+ cytolytic T cells[4,5].

There is emerging evidence that CD4+ T cells can also contribute to antitumour immunity, independent of their role as helpers and regulators of CD8+ cytolytic T cells[13]. A subset of CD4+ T cells develops cytolytic effector functions towards MHC-II-expressing tumour cells[14,15]. In addition, CD4+ T cells were shown capable of eradicating tumour cells that do not express MHC-II by mobilizing myeloid cells, which are specialized to process and present peptide epitopes on their MHC-II molecules[16–18]. The therapeutic potential of this indirect CD4+ T cell effector mechanism has, however, remained unclear. Moreover, the

[1]Laboratory of Experimental Dermatology, Department of Dermatology, University Hospital and Health Campus Immunology Infectiology and Inflammation (GC-I3), Otto-von-Guericke University, Magdeburg, Germany. [2]Laboratory for Molecular Cancer Biology, Center for Cancer Biology, VIB, Leuven, Belgium. [3]Laboratory for Molecular Cancer Biology, Department of Oncology, KU Leuven, Leuven, Belgium. [4]Translational Cell and Tissue Research, Department of Imaging and Pathology, KU Leuven, Leuven, Belgium. [5]Institute of Molecular and Clinical Immunology, Health Campus Immunology Infectiology and Inflammation (GC-I3), Otto-von-Guericke University, Magdeburg, Germany. [6]Department of Immunology, Genetics and Pathology, Uppsala University, Uppsala, Sweden. [7]Helmholtz Centre for Infection Research, Brunswick, Germany. [8]Department of Hematology, University Hospital and Health Campus Immunology Infectiology and Inflammation (GC-I3), Otto-von-Guericke University, Magdeburg, Germany. [9]Institute for Molecular Immunology, Centre for Molecular Medicine Cologne and Cologne Excellence Cluster on Cellular Stress Responses in Ageing-Associated Diseases, University of Cologne, Cologne, Germany. [10]Department of Pathology, UZ Leuven, Leuven, Belgium. [11]Department of General Medical Oncology, UZ Leuven, Leuven, Belgium. [12]Department of Applied Computational Cancer Research, Institute for AI in Medicine (IKIM), University Hospital Essen, Essen, Germany. [13]University of Duisburg-Essen, Essen, Germany. [14]Institute for Systems Immunology, Wuerzburg, Germany. [15]These authors contributed equally: Bastian Kruse, Anthony Buzzai, Naveen Shridhar, Andreas Braun. [16]These authors jointly supervised this work: Andreas J. Müller, Thomas Tüting. ✉e-mail: andreas.mueller@med.ovgu.de; thomas.tueting@med.ovgu.de

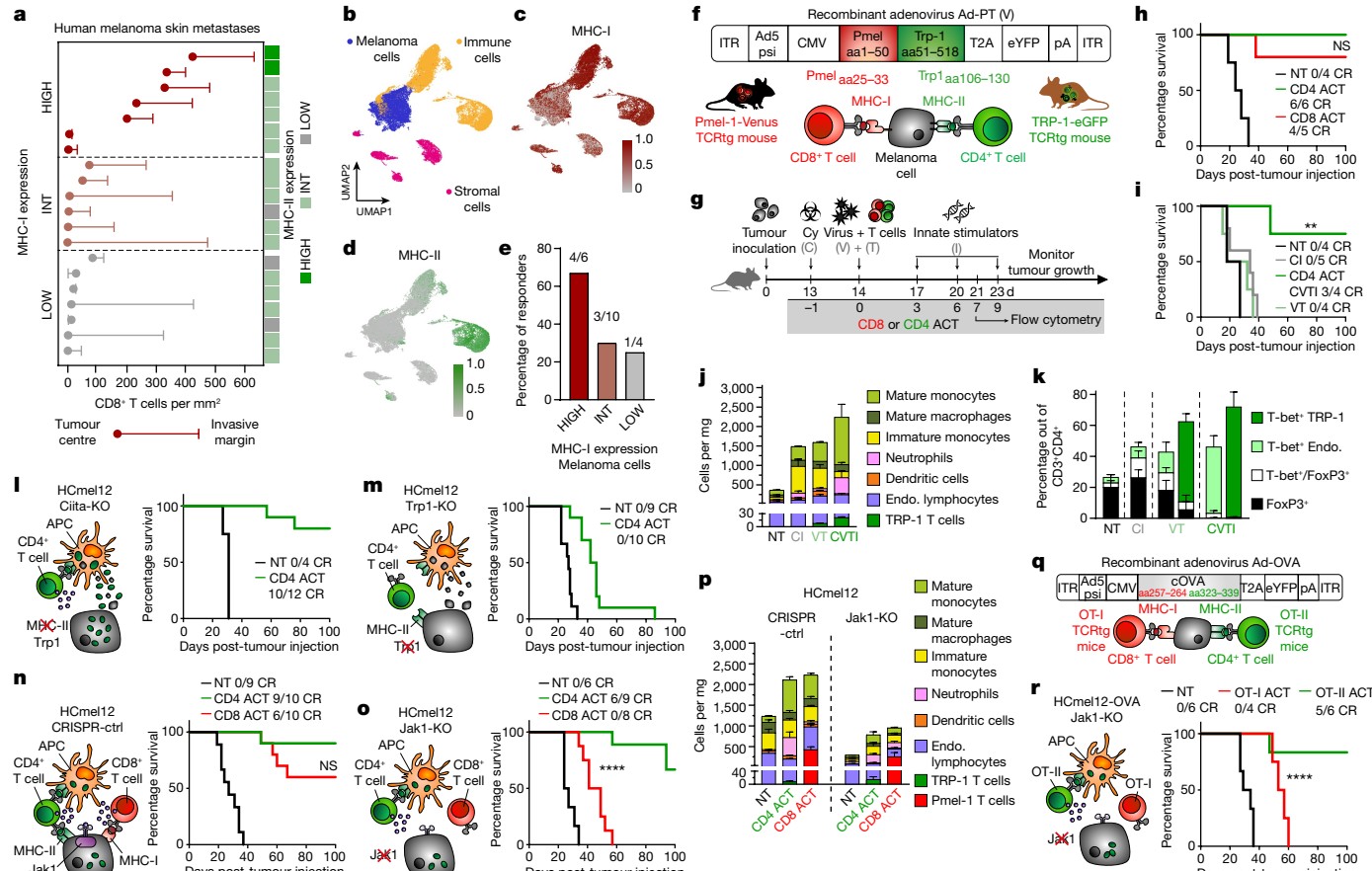

**Fig. 1 | A small population of CD4+ effector T cells can eradicate MHC-deficient and IFN-unresponsive melanomas that resist destruction by CD8+ cytotoxic T cells. a**, Density of CD8+ T cells infiltrating the tumour centre and the invasive margin of 20 human melanoma skin metastases and corresponding MHC-I and MHC-II expression, categorized into high, intermediate (int) and low expression. **b**, UMAP of single-cell transcriptomes from an extra set of 20 melanoma metastases in skin (*n* = 5), subcutis (*n* = 4) and lymph nodes (*n* = 11) annotated for melanoma, immune and stromal cell phenotypes. **c,d**, MHC-I (**c**) and MHC-II (**d**) gene set expression in single melanoma cells. **e**, ICB therapy responders in patients with high, intermediate and low MHC-I expression on melanoma cells. **f**, Structure of recombinant adenovirus Ad-PT. **g**, Experimental protocol for ACT immunotherapy of established tumours in mice (Cy, C, cyclophosphamide; V, Ad-PT; T, TCRtg Pmel-1 CD8+ or TRP-1 CD4+ T cells; I, innate stimuli, polyI:C and CpG) and time point for flow cytometric analyses. **h,i**, Kaplan–Meier survival curves of mice bearing established B16 melanomas and treated either with CD4 ACT or CD8 ACT (**h**) or with the indicated

components of the CD4 ACT protocol (**i**). NT, non-treated; CR, complete responders. **i**, \*\**P* = 0.0084. **j,k**, Immune cell composition (*n* = 2 biologically independent samples) (**j**) and phenotype of endogenous and transferred (VT, CVTI, right columns) CD4+ T cells (**k**) in tumours treated as indicated (mean ± s.e.m. from *n* = 4 biologically independent samples). **l–o**, Graphical representation of the immune cell interaction phenotypes (left) and Kaplan–Meier survival curves (right) of mice bearing established Ciita-KO (**l**), Trp1-KO (**m**), CRISPR-ctrl (**n**) or Jak1-KO (**o**) melanomas and treated as indicated. **o**, \*\*\*\**P* < 0.0001. **p**, Immune cell composition of tumours treated as indicated (mean ± s.e.m. from four biologically independent samples). **q**, Structure of recombinant adenovirus Ad-OVA. **r**, Graphical representation of the immune cell interaction phenotype of ovalbumin-expressing HCmel12 Jak1-KO cells (left) and Kaplan–Meier survival curves (right) of mice bearing established melanomas treated as indicated. \*\*\*\**P* < 0.0001. Survival was statistically compared using a log-rank Mantel–Cox test. NS, not significant.

spatiotemporal dynamics and the mechanism of action of CD4+ T cells within the tumour tissue have not been fully explained.

## Clinical relevance of immune evasion

Tumour cells can evade recognition and destruction by CD8+ T cells through MHC-I downregulation[19]. We reassessed the clinical relevance of this immune evasion mechanism in skin metastases of patients with melanoma by immunohistochemistry. Our results show very low MHC-I expression on melanoma cells in seven out of 20 samples that was associated with the absence of tumour-infiltrating CD8+ T cells (Fig. 1a and Extended Data Fig. 1a,b). Expression of MHC-II was mostly restricted to stromal and immune cells at the invasive margins in 15 out of 20 samples, often in association with infiltrating CD8+ T cells. Only two out of 20 samples that were densely infiltrated with CD8+ T cells showed high

expression levels of MHC-II on melanoma cells, whereas three out of 20 samples that lacked CD8+ T cell infiltrates barely expressed MHC-II (Fig. 1a and Extended Data Fig. 1c).

We independently analysed the expression of MHC molecules in single-cell RNA-sequencing (scRNA-seq) data obtained from a different cohort of 20 patients with treatment-naïve melanoma metastases (Fig. 1b–e)[20]. Transcriptional MHC-I downregulation in melanoma cells was evident in four out of 20 samples (Fig. 1c and Extended Data Fig. 1d). MHC-II expression was largely absent in melanoma cells and restricted to antigen-presenting immune cells (Fig. 1d and Extended Data Fig. 1e–g). Transcriptional downregulation of MHC-I on melanoma cells was associated with poor response to ICB (Fig. 1e). In aggregate, these results indicate that MHC-I downregulation in tumour cells is a frequent phenomenon during tumour evolution that favours immune escape.

## Establishment of a CD4 ACT model

To experimentally investigate how CD4[+] T cell effector functions could be therapeutically directed against MHC-deficient tumours that evade CD8[+] T cell immunity, we used enhanced green fluorescent protein (eGFP[+]) TRP-1 CD4[+] TCRtg T cells and Venus[+] Pmel-1 CD8[+] TCRtg T cells for ACT immunotherapies[21,22]. Vaccination with the recombinant adenovirus Ad-PT encoding a Pmel/gp100-TRP-1 fusion protein capable of stimulating both CD4[+] and CD8[+] TCRtg T cells (Fig. 1f) allowed us to directly compare their antitumour efficacy and their mechanism of action under identical experimental conditions using the ACT therapy protocol established in our previous work[23] (Fig. 1g). This protocol includes chemotherapeutic preconditioning with cyclophosphamide, a procedure also used in clinical ACT approaches, and adjuvant injections of the immunostimulatory oligonucleotides polyI:C and CpG that activate innate immunity through TLR3 and TLR9, respectively[24]. Similar oligonucleotides are currently explored in early phase clinical trials[25,26].

Initial experiments demonstrated that adoptively transferred TRP-1 CD4[+] T cells expanded much less efficiently in lymph nodes, peripheral blood and spleens when compared with adoptively transferred Pmel-1 CD8[+] T cells (Extended Data Fig. 2a,b). This observation is in line with the previously described intrinsic difference in the proliferative capacity between CD4[+] and CD8[+] T cells[27]. Despite their relatively poor in vivo expansion, adoptively transferred TRP-1 CD4[+] T cells eradicated established B16 melanomas as efficiently as Pmel-1 CD8[+] T cells (Fig. 1h and Extended Data Fig. 2c,d).

Cyclophosphamide pretreatment and innate immune stimulation were required in our CD4 ACT protocol to eradicate established tumours (Fig. 1i and Extended Data Fig. 2e,f), similar to our findings for CD8 ACT[24]. Flow cytometric analyses of tumour-infiltrating CD45[+] cells at day 7 after adoptive T cell transfer revealed a comparatively small subpopulation of adoptively transferred TRP-1 CD4[+] T cells representing only 1% of immune cells in treated mice (Fig. 1j and Extended Data Fig. 2g,h). The combination of cyclophosphamide pretreatment and adjuvant innate immune stimulation strongly promoted the differentiation of transferred and endogenous CD4[+] T cells towards a T helper (T$_H$) type 1 (T$_H$1)-directed phenotype and prevented the accumulation of regulatory T cells (Fig. 1k and Extended Data Fig. 2i). The adoptive transfer of T cells and the injection of innate immune stimuli independently increased the myeloid immune infiltrate that consisted predominantly of monocytes and macrophages. The full ACT protocol further increased the accumulation of monocytes (Fig. 1j).

## Indirect tumour recognition

TRP-1 CD4[+] T cells can eradicate B16 melanomas through direct recognition and cytolytic destruction[14]. However, most human melanoma cells do not express MHC-II molecules[10] (Extended Data Fig. 1c,e). We therefore investigated the ability of TRP-1 CD4[+] T cells to control MHC-II-deficient tumour cells by disrupting the *Ciita* gene encoding the MHC-II transactivator in HCmel12 mouse melanoma cells using CRISPR–Cas9 gene editing (Extended Data Fig. 2j). As controls, we also generated HCmel12 Trp1-knockout (Trp1-KO) cells that lack expression of the CD4[+] T cell target antigen (Extended Data Fig. 2k). In vitro experiments confirmed that TRP-1 CD4[+] T cells can directly respond to MHC-II-expressing HCmel12 cells in an antigen-specific manner (Extended Data Fig. 2l), but are more efficiently activated indirectly by MHC-II+ dendritic cells pulsed with HCmel12 cell lysates (Extended Data Fig. 2m). Of note, 40% of TRP-1 CD4[+] T cells isolated from CD4 ACT-treated mice produced IFNγ following stimulation with tumour lysate-pulsed dendritic cells, confirming their T$_H$1 phenotype.

Subsequent in vivo experiments showed that TRP-1 CD4[+] T cells were able to eradicate established MHC-II-deficient, but not TRP-1-deficient HCmel12 melanomas (Fig. 1l,m and Extended Data Fig. 2n–p). Treatment of tumours consisting of HCmel12 CRISPR-control (ctrl) and

HCmel12 Trp1-KO mixtures demonstrated that TRP-1 CD4[+] T cells also exerted substantial bystander killing activity, but could not prevent the outgrowth of HCmel12 Trp1-KO cells in all mice (Extended Data Fig. 2q,r). Furthermore, antibody-mediated depletion experiments confirmed that the treatment efficacy of TRP-1 CD4[+] T cells did not require the presence of CD8[+] T cells (Extended Data Fig. 2s,t). Thus, TRP-1 CD4[+] T cells can indirectly recognize and kill MHC-II-deficient tumour cells in the absence of CD8[+] T cells.

## Eradication of immune-evasive tumours

In subsequent experiments, we took advantage of the unique properties of HCmel12 melanoma cells that constitutively lack MHC-I and MHC-II expression unless exposed to IFNs. CRISPR–Cas9-mediated disruption of the *Jak1* gene encoding for a central mediator of the IFN signalling pathway leads to IFN-unresponsive and completely MHC-deficient tumour cells (Extended Data Fig. 3a). Robust in vivo growth of HCmel12 Jak1-KO melanoma cells required antibody-mediated depletion of natural killer (NK) cells before tumour inoculation, in line with the known ability of NK cells to directly recognize and kill MHC-I-deficient cells by cytolysis. This experimental setting allowed us to investigate the capacity of adoptively transferred TRP-1 CD4[+] T cells to indirectly recognize and kill IFN-unresponsive, MHC-deficient tumour cells independent from their ability to directly target and lyse MHC-II-expressing tumour cells and to provide help for the cytolytic activity of CD8[+] T and NK cells. Our results demonstrate that adoptively transferred TRP-1 CD4[+] T cells can indirectly eradicate established tumours that evade CD8[+] T cell control in the absence of NK cells (Fig. 1n,o and Extended Data Fig. 3b,c).

Subsequent flow cytometric analyses revealed that adoptively transferred Pmel-1 CD8[+] T cells represented a much larger subpopulation of tumour-infiltrating immune cells when compared to adoptively transferred TRP-1 CD4[+] T cells, both in HCmel12 CRISPR-ctrl and HCmel12 Jak1-KO tumours, consistent with their differential in vivo expansion dynamics. Nevertheless, both CD4 and CD8 ACT therapies substantially increased the number of myeloid cells in HCmel12 CRISPR-ctrl and HCmel12 Jak1-KO tumours (Fig. 1p and Extended Data Fig. 3d,e). The increased immune cell infiltrate was less pronounced in MHC-deficient HCmel12 Jak1-KO tumours when compared to HCmel12 CRISPR-ctrl tumours, in line with our observation in patient samples (Fig. 1a and Extended Data Fig. 1a,b).

Using ovalbumin as a second tumour antigen model (Fig. 1q), we recapitulated the different in vivo expansion dynamics of adoptively transferred ovalbumin-specific OT-II TCRtg CD4[+] T cells and OT-I TCRtg CD8[+] T cells (Extended Data Fig. 3f,g). Again, ovalbumin-specific OT-II TCRtg CD4[+] T cells were able to eradicate ovalbumin-expressing MHC-deficient HCmel12 Jak1-KO tumours, whereas ovalbumin-specific OT-I TCRtg CD8[+] T cells were ineffective (Fig. 1r and Extended Data Fig. 3h,i). Taken together, these results confirmed that a few CD4[+] effector T cells can indirectly eradicate established IFN-unresponsive, MHC-deficient tumours that evade CD8[+] T cell immunity independent of NK cells.

## Intratumoural CD4[+] T cell dynamics

We proposed that CD4[+] effector T cells are efficiently activated in tumour tissues by antigen-presenting immune cells that constitutively express MHC-II, whereas CD8[+] T cells require MHC-I-restricted antigen presentation by tumour cells. The expression pattern of MHC molecules in tumour tissues should therefore also determine the spatial distribution and migratory behaviour of adoptively transferred CD4[+] and CD8[+] T cells. To address these hypotheses, we generated amelanotic (Tyr-KO) HCmel12 CRISPR-ctrl and HCmel12 Jak1-KO cells that express tag-blue fluorescent protein (tagBFP) for immunofluorescence microscopy imaging (Extended Data Fig. 4a). Confocal microscopy showed only very few adoptively transferred eGFP[+] TRP-1 CD4[+] T cells in local clusters at the invasive margin of established amelanotic tagBFP-labelled

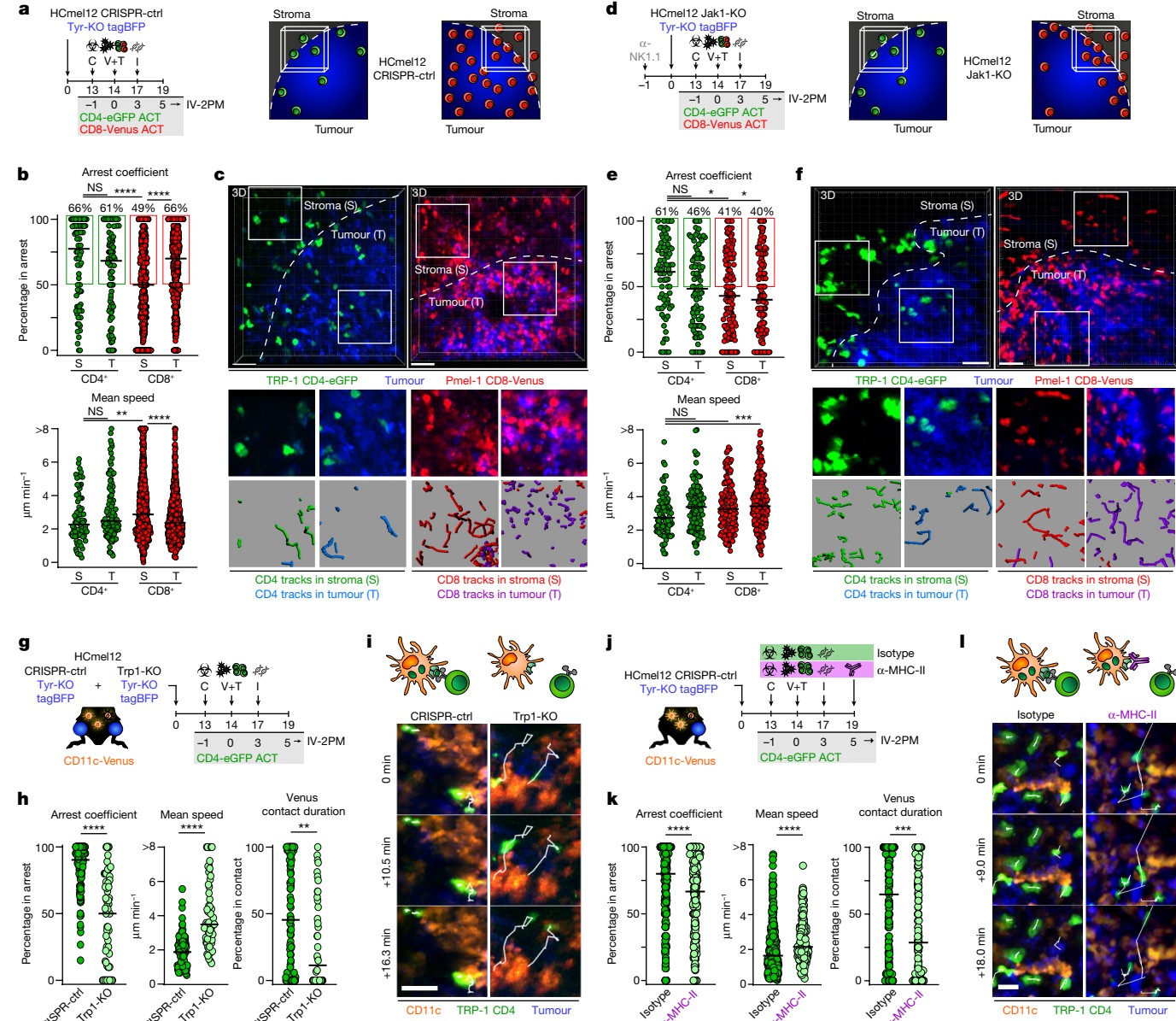

**Fig. 2 | CD4⁺ effector T cells interact with MHC-II-expressing CD11c⁺ antigen-presenting cells in clusters within the invasive tumour margin.** **a**,**d**, Experimental protocols to assess the distribution of adoptively transferred T cells (left) and graphics depicting the invasive tumour margin (right) of CRISPR-ctrl (**a**) or Jak1-KO (**d**) tumours. **b**,**e**, Arrest coefficient and mean speed of adoptively transferred Venus⁺ Pmel-1 CD8⁺ (red) and eGFP⁺ TRP-1 CD4⁺ T cells (green) in the stromal (S) and tumoural (T) compartment at the invasive margin (bars indicate the median) of CRISPR-ctrl (**b**) or Jak1-KO (**e**) tumours (75–842 cells examined from three independent experiments; ****$P < 0.0001$, ***$P = 0.0007$, **$P = 0.0068$, CD4⁺ versus CD8⁺ in stroma *$P = 0.0204$, CD4⁺ in stroma versus CD8⁺ in tumour *$P = 0.0107$ using a Kruskal–Wallis test with Dunn's multiple comparison test). **c**,**f**, Representative intravital microscopic images (scale bars, 100 μm) and insets exemplifying 450 s motion tracks of Venus⁺ Pmel-1 CD8⁺ and eGFP⁺ TRP-1 CD4⁺ T cells at the stromal (S) and tumoural (T) area of the invasive tumour margin of CRISPR-ctrl (**c**) or Jak1-KO (**f**) melanomas. **g**, Experimental protocol to assess antigen-dependent interactions between eGFP⁺ TRP-1 CD4⁺ T cells and CD11c⁺ immune cells. **h**, Arrest coefficient, mean speed and relative contact duration between eGFP⁺ TRP-1 CD4⁺ T cells and CD11c-Venus cells (the bars indicate the median; 43–132 cells examined from three independent experiments; ****$P < 0.0001$, **$P = 0.0022$ with a Mann–Whitney $U$-test). **i**, Representative motion tracks of eGFP⁺ TRP-1 CD4⁺ T cells interacting with CD11c-Venus cells in CRISPR-ctrl and Trp1-KO melanomas. Scale bars, 20 μm. **j**, Experimental protocol to assess the impact of MHC-II blockade on antigen-dependent interactions between eGFP⁺ TRP-1 CD4⁺ T cells and CD11c⁺ immune cells. **k**, Arrest coefficient, mean speed and relative contact duration between eGFP⁺ TRP-1 CD4⁺ T cells and CD11c-Venus cells (the bars indicate the median; 203–273 cells examined from three independent experiments; ****$P < 0.0001$, ***$P = 0.0003$, with a Mann–Whitney $U$-test). **l**, Representative motion tracks of eGFP⁺ TRP-1 CD4⁺ T cells interacting with CD11c-Venus cells in CRISPR-ctrl Tyr-KO tumours with and without MHC-II blockade. Scale bars, 20 μm. Data were pooled from at least three independent mice and groups statistically compared using a one-way ANOVA with Tukey post hoc.

HCmel12 CRISPR-ctrl tumours. By contrast, large numbers of Venus⁺ Pmel-1 CD8⁺ T cells infiltrated both the invasive margin and the tumour centre (Extended Data Fig. 4b,d), consistent with the flow cytometry results (Fig. 1j,p). TRP-1 CD4⁺ T cells also locally clustered at the invasive

margin of MHC-deficient HCmel12 Jak1-KO tumours, whereas Pmel-1 CD8⁺ T cells only infiltrated the invasive margin but not the tumour centre (Extended Data Fig. 4c,e). These results showed that the distribution of CD8⁺ T cells in tumour tissues of our preclinical model depends

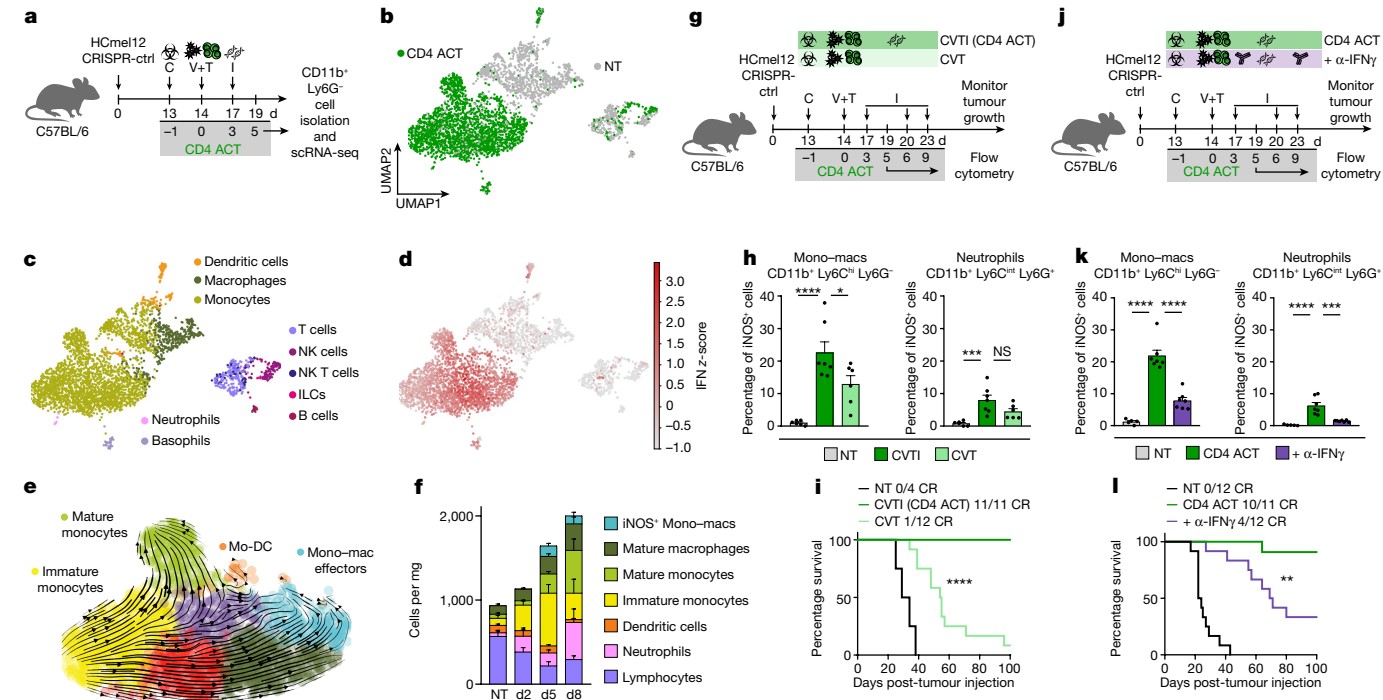

**Fig. 3 | CD4⁺ effector T cells and innate immune stimulation promote the recruitment and IFN-dependent activation of monocytes to eradicate established tumours. a**, Experimental protocol for scRNA-seq analyses of tumour-infiltrating CD11b⁺ Ly6G⁻ cells. **b**, Visualization and dimensionality reduction of single-cell transcriptomes from CD4 ACT-treated and non-treated (NT) mice using UMAP. **c,d**, UMAP plots with cell types assigned using SingleR (**c**) and z-score for the hallmark IFN gamma response gene set (MSigDB) of each cell (**d**). **e**, RNA velocity projected on UMAP plots for monocytes and macrophages (Mono–macs) of CD4 ACT-treated tumours. Arrows point towards the predicted course of cell maturation dynamics. Arrow sizes indicate the strength of predicted directionality. **f**, Immune cell composition of HCmel12 tumours treated as indicated (mean ± s.e.m. from n = 6 biologically

independent samples). **g,j**, Experimental treatment protocol to investigate the impact of innate stimuli (**g**) or IFNγ-blockade (**j**) on myeloid cell activation and tumour control. **h,k**, Percentage of intratumoural iNOS⁺ mono–macs and neutrophils (mean ± s.e.m. from n = 5–7 biologically independent samples). **h**, ****P < 0.0001, *P = 0.0111, ***P = 0.0005; **k**, ****P < 0.0001, ***P = 0.0002). **i,l**, Kaplan–Meier survival curves of mice bearing established HCmel12 CRISPR-ctrl tumours, treated as indicated (**i**, ****P < 0.0001; **l**, **P = 0.0053). Means between groups were statistically compared using a one-way ANOVA with Tukey post hoc. Survival was statistically compared using log-rank Mantel–Cox test. NT, non-treated; C, cyclophosphamide; V, Ad-PT; T, TCRtg TRP-1 CD4⁺ T cells; I, innate stimuli, polyI:C and CpG; CR, complete responders.

on MHC-I expression, reminiscent of the spatial distribution of CD8⁺ T cells in human melanomas (Extended Data Fig. 1b).

Intravital two-photon microscopy at the invasive margin confirmed the differential intratumoural localization of adoptively transferred CD4⁺ and CD8⁺ T cells and revealed substantial differences in their migratory behaviour. In HCmel12 CRISPR-ctrl tumours, CD4⁺ T cells arrested both in the stromal and the tumoural compartment within the invasive margin, whereas CD8⁺ T cells remained highly motile in the stroma and only arrested in the tumoural compartment (Fig. 2a–c and Supplementary Videos 1 and 2). In MHC-deficient HCmel12 Jak1-KO tumours, CD4⁺ T cells showed no changes in their migratory behaviour in the stroma and a slightly increased motility in the tumoural compartment of the invasive margin. By contrast, CD8⁺ T cells failed to arrest at all and were always highly motile in MHC-deficient HCmel12 Jak1-KO melanomas (Fig. 2d–f and Supplementary Videos 3 and 4). Together, these observations indicated that CD4⁺ T cells can interact with MHC-II⁺ immune and tumour cells at the invasive margin, whereas CD8⁺ T cells predominantly interact with MHC-I⁺ tumour cells.

We confirmed the fundamental difference in the spatial distribution and migratory behaviour of CD4⁺ and CD8⁺ T cells in tumour tissues using amelanotic (Tyr-KO) MHC-deficient HCmel12 Jak1-KO tumours that express tagBFP-Ova. Very few adoptively transferred ovalbumin-specific dsRed⁺ OT-II CD4⁺ TCRtg T cells clustered locally within the tumour invasive margin and arrested both in the stromal and the tumoural compartment. By contrast, large numbers of ovalbumin-specific Venus⁺ OT-I CD8⁺ TCRtg T cells infiltrated only the

tumour invasive margin but not the tumour centre and were always highly motile (Extended Data Fig. 4f,g). These observations confirmed that CD4⁺ T cells preferentially interact with MHC-II⁺ antigen-presenting cells at the invasive tumour margin, whereas CD8⁺ T cells require MHC-I expression on tumour cells to exert their effector functions in vivo.

## MHC-II-restricted antigen recognition

A likely interaction partner for CD4⁺ T cells are dendritic cells due to their ability to efficiently ingest and process tumour antigens for MHC-II-dependent antigen presentation[28–30]. To visualize antigen-specific interactions between TRP-1 CD4⁺ T cells and MHC-II⁺ CD11c⁺ antigen-presenting cells, we further generated amelanotic (Tyr-KO) HCmel12 Trp1-KO cells expressing tagBFP, injected them into opposite legs of CD11c-Venus transgenic mice that harbour fluorescent antigen-presenting immune cells[31] and treated established tumours with adoptively transferred eGFP⁺ TRP-1 CD4⁺ T cells (Fig. 2g and Extended Data Fig. 5a,b). Confocal microscopy revealed local accumulations of eGFP⁺ TRP-1 CD4⁺ T cells in association with MHC-II-expressing CD11c-Venus⁺ immune cells within tumour invasive margins in HCmel12 CRISPR-ctrl but not in HCmel12 Trp1-KO tumours (Extended Data Fig. 5c–f). Tumour cells surrounding clusters of CD11c-Venus⁺ immune cells with CD4⁺ T cells upregulated the expression of MHC-II exclusively in mice bearing HCmel12 CRISPR-ctrl tumours, consistent with the notion that CD4⁺ T cells were activated locally and secreted IFNγ in an antigen-dependent manner.

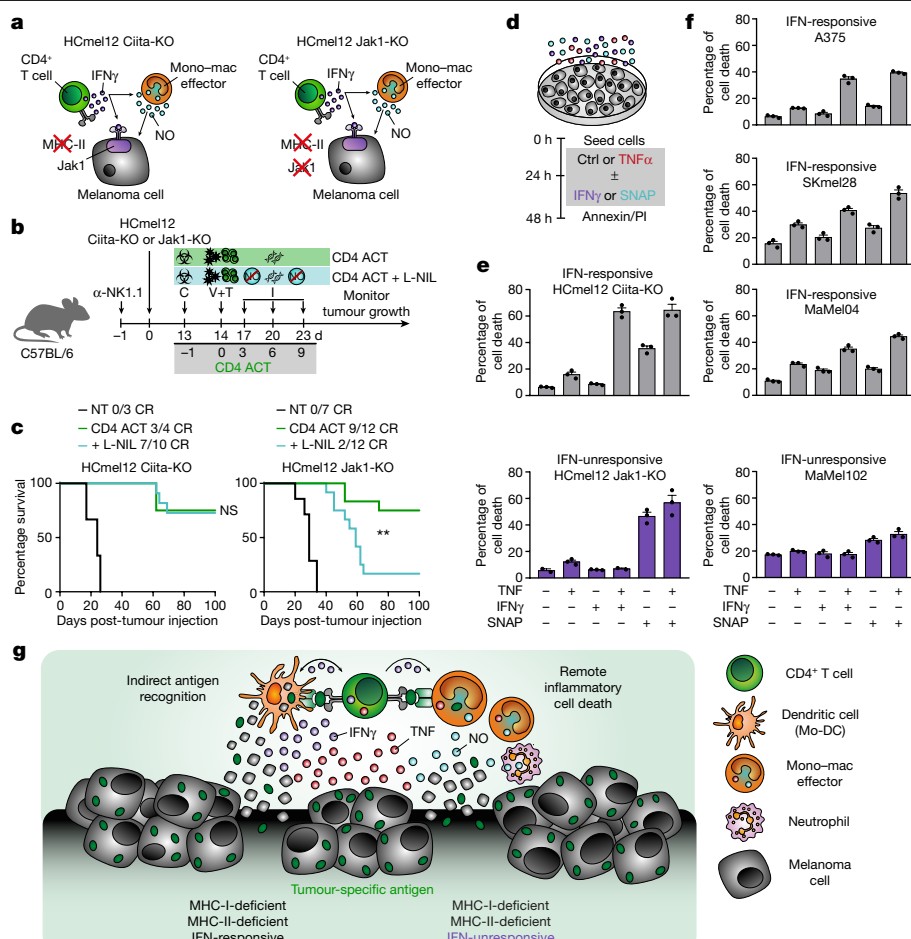

**Fig. 4 | CD4[+] effector T cells cooperate with activated iNOS-expressing tumouricidal monocytes and macrophages to orchestrate remote inflammatory cell death of MHC-deficient and IFN-unresponsive tumours.** **a**, Graphical representation of interaction phenotypes of indicated HCmel12 variants. **b**, Experimental treatment protocol to study the impact of chemical iNOS inhibition using L-NIL on CD4 ACT-mediated tumour control. **c**, Kaplan–Meier survival graphs of mice bearing established HCmel12 Ciita-KO melanomas (left) or HCmel12 Jak1-KO melanomas (right) and treated as indicated (NT, non-treated; CR, complete responders; **P = 0.0033). Survival was statistically compared using a log-rank Mantel–Cox test. Means between groups were statistically compared using a one-way ANOVA with Tukey post hoc. **d**, Experimental protocol to assess the ability of the inflammatory mediators TNF, IFNγ and the nitric oxide donor SNAP to induce melanoma cell death. **e**,**f**, Percentage of cell death in mouse (**e**) and human (**f**) melanoma cells treated as indicated (mean ± s.e.m. from 2–3 technical replicates). **g**, Graphical summary of inflammatory cell death induction of MHC-deficient and IFN-unresponsive tumours by CD4[+] T cells in cooperation with iNOS-expressing myeloid cells.

Intravital two-photon microscopy demonstrated that eGFP[+] TRP-1 CD4[+] T cells preferentially arrested and engaged in long-lasting close interactions with CD11c-Venus[+] immune cells only in HCmel12 CRISPR-ctrl tumours, but not in HCmel12 Trp1-KO tumours (Fig. 2h,i, Extended Data Fig. 5g,h and Supplementary Video 5). Antibody-mediated blockade of MHC-II-restricted antigen presentation abrogated the interaction between CD4[+] T cells and CD11c-Venus[+] immune cells, confirming the specificity of our findings (Fig. 2j–l and Supplementary Video 6). These observations demonstrate that CD4[+] effector T cells cluster with CD11c-Venus[+] immune cells at the tumour invasive margin where they maintain prolonged antigen-specific and MHC-II-restricted interactions that enable them to eradicate MHC-deficient tumours. Of note, CD4[+] effector T cells also clustered with MHC-II[+] dendritic antigen-presenting cells and macrophages in human melanomas (Extended Data Fig. 6).

## Recruitment of IFN-activated monocytes

Next, we investigated how a comparatively small subpopulation of CD4[+] effector T cells can eradicate large established tumours. We proposed that adoptively transferred CD4[+] T cells and injections of synthetic nucleic acids direct a strong T_H1-associated pathogen defence mechanism that also engages mononuclear phagocytes towards tumour destruction. To explore this hypothesis, we performed scRNA-seq analyses of sorted CD11b[+] Ly6G- tumour-infiltrating mononuclear phagocytes from mice treated with our CD4 ACT protocol and from non-treated controls. Dimensionality reduction, visualization using uniform manifold approximation and projection (UMAP) and cell type annotation using SingleR showed a clear separation between the monocyte–macrophage clusters derived from CD4 ACT-treated and non-treated tumours (Fig. 3a–c). Differential gene expression and gene set enrichment analyses revealed a strong activation of IFN-response genes on therapy (Fig. 3d and Extended Data Fig. 7a,b).

Unsupervised Leiden clustering of the monocyte–macrophage lineage for CD4 ACT-treated and non-treated groups separated four and seven cell states, respectively. The four cell states in non-treated mice express marker genes characteristic for immature monocytes (NT0), monocyte-derived dendritic cells (NT1), monocyte–macrophage effector cells (NT2) and mature monocytes (NT3). The seven cell states in CD4 ACT-treated mice represent IFN-activated counterparts of the intratumoural monocyte–macrophage network found in non-treated controls (Extended Data Fig. 7c,d). Computation of the

RNA velocity and pseudotime inference revealed a dynamic development of Ly6c-hi inflammatory immature monocytes towards phenotypes of IFN-activated monocyte-derived dendritic cells (ACT1, antigen-presentation phenotype), monocyte–macrophage effectors (ACT2a-c, tumouricidal phenotype) and Ly6c-lo mature monocytes (ACT3a,b, patrolling phenotype) (Fig. 3e and Extended Data Fig. 7e,f).

Flow cytometric analyses of tumour-infiltrating immune cells at days 2, 5 and 8 after adoptive CD4+ T cell transfer confirmed the dynamic recruitment of Ly6c-hi immature monocytes into the tumour microenvironment on CD4 ACT therapy and their differentiation into iNOS-expressing tumouricidal mononuclear phagocytes (Fig. 3f and Extended Data Fig. 7g,h). Together, the flow cytometric and transcriptomic analyses indicated that CD4+ T cells and innate immune stimuli reprogramme the myeloid network in treated tumours through the recruitment of immature monocytes that acquire IFN-activated cellular states and dynamically differentiate towards MHC-II antigen-presenting and iNOS-expressing tumouricidal effector phenotypes.

Our initial data showed that CD4+ effector T cells and innate immune stimulation independently promoted the recruitment of immature monocytes into the tumour microenvironment (Fig. 1j and Extended Data Fig. 2h). Next, we asked whether CD4+ T cells and innate immune stimuli synergized on a quantitative or qualitative level for the acquisition of tumouricidal monocyte effector functions. Omitting innate stimuli from our combined ACT therapy regimen reduced the recruitment of neutrophils, but not of immature Ly6C-hi monocytes (Extended Data Fig. 8a,b). However, both CD4+ T cells and innate immune stimuli were indispensable for full iNOS induction in the recruited monocytes (Fig. 3g,h and Extended Data Fig. 8b). Functionally, the synergism of the combined therapy was required locally in tumour tissues for the eradication of established tumours, leading to a striking increase in tumour-free survival (Fig. 3i and Extended Data Fig. 8c–e). We proposed that the release of IFNγ was responsible for the CD4+ T cell-driven qualitative enhancement of tumouricidal monocyte effector functions on the molecular level. In support of this hypothesis, we found that antibody-mediated neutralization of IFNγ did not influence the absolute number of tumour-infiltrating monocytes and neutrophils, but significantly reduced the frequency of iNOS-expressing monocytes (Fig. 3j,k and Extended Data Fig. 8f,g). IFNγ was essential to eradicate established tumours (Fig. 3l and Extended Data Fig. 8h).

## Inflammatory tumour cell death

CD4+ T cell-derived IFNγ can either act on tumour cells or through IFN-dependent activation of iNOS-expressing tumouricidal myeloid cells[18,32–34] (Fig. 4a). Treatment with the highly specific iNOS inhibitor (N6-(1-iminoethyl)-L-lysine, L-NIL) abrogated the ability of CD4 ACT treatment to control IFN-unresponsive, MHC-deficient HCmel12 Jak1-KO tumours, but had no effect on CD4 ACT treatment of IFN-responsive, MHC-deficient HCmel12 Ciita-KO tumours (Fig. 4b,c and Extended Data Fig. 9a,b). Transient antibody-mediated depletion of CCR2+ monocytes impaired the efficacy of CD4 ACT therapy to a greater extent than the depletion of Ly6G+ neutrophils, supporting a predominant role of iNOS-expressing monocytes and macrophages for tumour eradication (Extended Data Fig. 9c–e). In aggregate, these results indicated that the ability of adoptively transferred CD4+ T cells to indirectly eradicate IFN-unresponsive, MHC-deficient tumour cells involved the remote action of nitric oxide released by IFN-activated tumouricidal myeloid cells.

Our results raised the question as to how myeloid cell-derived nitric oxide contributes to death of IFN-unresponsive tumour cells. Recent data that explained the cytokine driven immunopathology in patients with COVID-19 revealed an inflammatory mode of cell death driven by the concerted action of IFNγ, tumour necrosis factor (TNF) and nitric oxide[35]. In experiments inspired by these observations, we found that the nitric oxide donor S-nitroso-N-acetylpenicillamine (SNAP)

effectively induced death of both IFN-responsive HCmel12 Ciita-KO and IFN-unresponsive HCmel12 Jak1-KO melanoma cells in vitro (Fig. 4d,e and Extended Data Fig. 9f,g). IFNγ and TNF were not effective alone and only induced death of IFN-responsive HCmel12 Ciita-KO melanoma cells when used in combination. The ability of the inflammatory mediators to induce cell death was fully recapitulated in a panel of IFN-responsive and IFN-unresponsive human melanoma cell lines (Fig. 4f and Extended Data Fig. 9h). These in vitro results demonstrated that IFNγ sensitizes IFN-responsive melanoma cells towards TNF-induced cell death and suggested that myeloid cell-derived nitric oxide contributes to efficient inflammatory cell death of IFN-unresponsive melanoma cells (Fig. 4g).

## Discussion

CD4+ T cells have primarily been perceived as helper cells for the activation of CD8+ effector T cells[36], which kill tumour cells by direct cytolysis. In our work, we showed that CD4+ effector T cells are also able to independently eradicate established tumours as efficiently as CD8+ cytolytic T cells. Using intravital microscopy, we found that CD4+ and CD8+ effector T cells differ fundamentally in their mode and their site of action in tumour tissues. Only very few CD4+ effector T cells, representing 1% of tumour-infiltrating immune cells, locate at the tumour invasive margins where they interact with CD11c+MHC-II+ antigen-presenting immune cells and indirectly eliminate tumours. By contrast, much larger numbers of CD8+ cytolytic T cells infiltrate the tumour centre where they directly target and kill MHC-I-expressing tumour cells.

Further innate immune stimulation boosted the $T_H1$-directed differentiation of CD4+ T cells, increased the recruitment of immature monocytes into the tumour microenvironment and supported their IFN-dependent activation and differentiation towards antigen-presenting and iNOS-expressing tumouricidal effector phenotypes. Together, CD4+ T cells and IFN-activated mononuclear phagocytes initiate an indirect inflammatory tumour cell death process that acts from the 'outside-in' and can be abrogated by neutralizing IFNγ. This unique mode of action operates in parallel to and independent of the direct cytolytic activities of NK cells[37] and enables the eradication of MHC-deficient and IFN-unresponsive tumours that evade direct recognition and destruction by CD8+ cytolytic T cells (Extended Data Fig. 10).

The ability of lymphocytes to cooperate with mononuclear phagocytes for immune defence was first observed in experiments with bacterial infections by investigators trying to understand immune resistance of mice to pathogens and tumours more than 50 years ago[38,39]. We faithfully recapitulate in our experimental model the cellular and molecular mechanisms underlying this cooperation. MHC-II-dependent antigen presentation by myeloid cells, dynamic IFN-dependent activation and differentiation of the monocyte–macrophage and dendritic cell lineages[40], and the induction of remote cell death through inflammatory mediators may represent shared immune defence mechanisms that critically contribute to the control of tumours[33,34,41] and pathogens[35,42–46].

Our results have important implications for the future of patient care, as ACT immunotherapies using CD4+ T cells have already been successfully implemented in clinical studies. ACT with NY-ESO-1 specific CD4+ T cells were used to successfully treat metastatic melanoma[47] and ACT with CD4+ T cells genetically engineered to express an MHC-II–restricted T cell receptor specifically recognizing the cancer germline antigen MAGE-A3 demonstrated clinical efficacy[48]. Our results strongly support the clinical development of ACT-based therapeutic strategies that include appropriate activating stimuli for myeloid cells to unleash the full potential of CD4+ T cell effector functions against immune-evasive tumours.

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

# Methods

## Patient biopsies

Skin metastases of 20 patients with melanoma (clinical stage III–IV), obtained during routine histopathological diagnostic procedures at the Department of Dermatology of the University Medical Centre Magdeburg, were analysed by immunohistochemistry for the expression of MHC-I (1:100), MHC-II (1:200) and CD8 (undiluted), in addition to the melanoma markers MART-1 (undiluted), gp100 (undiluted), S-100 (undiluted) and Sox10 (undiluted) using the automated Ventana BenchMark platform and standard protocols. These studies were performed in the context of routine clinical workup and were approved by the ethics committee of the Otto-von-Guericke University Hospital Magdeburg (approval number 162/20).

Cell suspensions derived from immunotherapy naive tumour biopsies of 20 melanoma metastases in skin ($n = 5$), subcutis ($n = 4$) and lymph nodes ($n = 11$) from 19 patients (clinical stage IIIB–IV) were analysed for their messenger RNA expression profile in single cells. Methods for tumour dissociation, library construction, scRNA-seq data acquisition and analysis were described previously[20]. This study was approved by the UZ Leuven Medical Ethical Committee and written consent obtained from all patients. The immune cells were distinguished from other tumour microenvironment cells by high immune signature score and low copy number variation score. Next, the cells were reclustered and annotated using SingleR. The MHC-I and MHC-II gene signature scores were measured using the AUCell R package[49].

## MILAN (mIHC)

Multiplex immunofluorescent images were generated by sequential immunostaining and antibody removal according to the published MILAN protocol[50] from human melanoma biopsies as described previously[50]. From the complete 41 protein markers included in the published panel, a reduced panel including panCK ($1 \mu g \, ml^{-1}$), CD3 ($1 \mu g \, ml^{-1}$), CD4 (1:200), CD8, ($1 \mu g \, ml^{-1}$), FOXP3 ($1 \mu g \, ml^{-1}$), MHC-II ($1 \mu g \, ml^{-1}$), CD11c ($1 \mu g \, ml^{-1}$), CD68 (1:200), MLANA (1:500), MITF ($1 \mu g \, ml^{-1}$) and CD31 ($1 \mu g \, ml^{-1}$) for staining keratinocytes, effector T cells, MHC-II expressing myeloid cell subsets, melanoma cells and vessels respectively, is shown. Image analysis was performed as described previously[51]. Briefly, stains were visually evaluated for quality by an experienced pathologist. Flat field correction was performed using a custom implementation of the methodology[52]. Consecutive staining rounds were registered using a previously described algorithm[53]. Tissue autofluorescence was subtracted using a baseline image stained only with a secondary antibody.

## Mice

Mice were housed in an ambient temperature- and humidity-controlled environment on a 12-h light/dark cycle to mimic natural conditions. Wild type C57BL/6J mice were purchased from Janvier or Charles River. The T cell receptor-transgenic Pmel-1 (B6.Cg-Thy1a/Cy Tg(TcraTcrb)8Rest/J), TRP-1 (B6.Cg-Rag1tm1Mom Tyrp1B-w Tg (Tcra,Tcrb) 9Rest/J), OT-I (C57BL/6-Tg(TcraTcrb)1100Mjb/J) and OT-II (B6.Cg-Tg(TcraTcrb)425Cbn/J) mice, and the fluorescent B6-eGFP (C57BL/6-Tg (UBC-GFP) 30Scha/J) and CD11c-eYFP (B6.Cg-Tg (Itgax-Venus) 1Mnz) mice were purchased from Jackson Laboratories. Pmel-1-Venus mice were generated by crossing CAG-Venus mice with Pmel-1 mice. TRP-1-eGFP mice were generated by crossing B6-eGFP mice into the TRP-1-deficient Rag1-KO background of TRP-1 mice. OT-I-Venus mice were generated by crossing CAG-Venus mice with OT-I mice. OT-II-dsRed were generated by crossing OT-II mice with hCD2-dsRed mice (kindly provided by C. Halin). All transgenic mice were bred in house. Age matched cohorts of tumour developing mice were randomly allocated to the different experimental groups. All animal experiments were conducted with male mice on

the C57BL/6 background under specific pathogen-free conditions in individually ventilated cages according to the institutional and national guidelines for the care and use of laboratory animals with approval by the Ethics Committee of the Office for Veterinary Affairs of the State of Saxony-Anhalt, Germany (permit licence numbers 42502-2-1393 Uni MD, 42502-2-1586 Uni MD, 42502-2-1615 Uni MD, 42502-2-1672 Uni MD) in accordance with legislation of both the European Union (Council Directive 499 2010/63/EU) and the Federal Republic of Germany (according to §8, section 1 TierSchG and TierSchVersV).

## Cell lines and cell culture

The mouse melanoma cell line HCmel12 was established from a primary melanoma in the Hgf-Cd4k$^{R24C}$ mouse model by serial transplantation in our laboratory as described previously[54]. The mouse melanoma cell line B16 and the human melanoma cell lines A375 and SKmel28 were purchased from ATCC. The human melanoma cell lines MaMel04 and MaMel102 were kindly provided by D. Schadendorf. All cell lines were cultured in complete Roswell Park Memorial Institute (RPMI) medium consisting of RPMI 1640 medium (Life Technologies) supplemented with 10% foetal calf serum (Biochrome), 2 mM L-glutamine, 10 mM non-essential amino acids, 1 mM HEPES (all form Life Technologies), 20 µM 2-mercoptoethanol (Sigma), 100 IU ml$^{-1}$ penicillin and 100 µg ml$^{-1}$ streptomycin (Invitrogen) in a humidified incubator with 5% $CO_2$. The cell lines were routinely screened for mycoplasma contamination and were authenticated by the commercial provider or by short tandem repeat fingerprinting.

## In vitro cell death assays

For the measurements of cell death in mouse and human melanoma cell lines, cells were first seeded in 96-well plates in complete RPMI medium. Inflammatory mediators were added after 24 h (10 U ml$^{-1}$ recombinant mouse IFNγ (Peprotech); 1,000 U ml$^{-1}$ recombinant mouse TNF (Peprotech); 100 U ml$^{-1}$ animal-free recombinant human IFNγ (Peprotech); 1,000 U ml$^{-1}$ recombinant human TNF (Peprotech) and 100 µM SNAP (Cayman Chemicals)). After 24 h, floating and adherent cells were gathered and stained using the fluorescein isothiocyanate (FITC) Annexin V Apoptosis Detection Kit I (BD Pharmingen) and analysed using the Attune NxT acoustic focusing flow cytometer (ThermoFisher).

## Adenovirus generation and expansion

To generate the adenoviral vaccine Ad-PT, a fusion construct was generated consisting of the first 150 base pairs of the human *PMEL* complementary DNA (coding for amino acids 1–50 of the human PMEL/gp100 protein including the CD8$^+$ T cell epitope KVPRNQDWL) and 1,404 base pairs of the mouse *Trp1* cDNA (coding for amino acids 51–518 including the CD4$^+$ T cell epitope SGHNCGTCRPGWRGAACNQKILTVR) followed by sequences coding for a T2A viral self-cleaving peptide and the yellow fluorescent marker protein eYFP. This vaccine construct was cloned into the pShuttle vector (termed pShuttle-PT-YFP). A recombinant adenovirus vector with this sequence was then generated by a recombineering technique in *Escherichia coli* strain SW102 using bacmid pAdZ5-CV5-E3$^+$. The E1 region of this bacmid is replaced by a selection/counter-selection cassette called ampicillin, LacZ, SacB or the ALS cassette. Next, *E. coli* with this bacmid were electroporated with the PT-YFP transgene with homology arms flanking the ALS cassette obtained by PCR amplification using pShuttle-PT-YFP as a template. Positive colonies were isolated after antibiotic selection on LB-sucrose plates. Ad-PT and Ad-OVA were expanded using the 911 human embryonic retinoblast cell line. A confluent monolayer of the cells in T175 cell culture flasks was infected with Ad-PT or Ad-OVA at MOI 1. The cytopathic effects were observed at around 36 h of incubation at 37 °C. Next, cells were scraped, freeze–thawed three times

and the lysates were cleared by centrifuging at the speed of 7,000*g* for 45 min. The crude virus was then titrated by the $TCID_{50}$ method according to standard protocols.

## CRISPR–Cas9-mediated genetic cell engineering

To generate Ciita-KO, Trp1-KO, Jak1-KO and Tyr-KO HCmel12 variants, HCmel12 melanoma cells were used that can be readily genetically modified using CRISPR–Cas9-mediated gene editing[55]. Cells were seeded into a 12-well plate at a density of $5 \times 10^5$ cells per well and cotransfected with 1.6 μg pX330-sgRNA and 0.4 μg plasmid expressing GFP (pRp-GFP) using Fugene HD transfection reagent (Promega) according to the manufacturer's instructions. GFP positive cells were single-cell sorted using a FACSAria III Cell Sorter (BD) to generate polyclonal and monoclonal populations per targeted gene. HCmel12 cells were mock transfected with pX330 plasmid without single-guide RNA and the polyclonal cell line was used as a CRISPR-control in all performed experiments. Genomic DNA from cultured knockout variants was extracted using the NucleoSpin Tissue kit (Macherey-Nagel) according to the manufacturer's protocol. A two-step PCR protocol was performed to generate targeted PCR amplicons for next-generation sequencing. In the first PCR, specific primers for the target gene with more adapter sequences complementary to the barcoding primers were used to amplify the genomic region of interest with Phusion HD polymerase (New England Biolabs). In a second PCR, adapter-specific universal primers containing barcode sequences and the Illumina adapter sequences P5 and P7 were used (Illumina barcodes D501-508 and D701-D712). Next-generation sequencing was performed with MiSeq Gene and Small Genome Sequencer (Illumina) according to manufacturer's standard protocols with a single-end read and 300 cycles (MiSeq Reagent Kit v.2 300 cycle). For the detection of insertions or deletions, the web-based program Outknocker (http://www.outknocker.org/) was used as previously described[56]. FASTQ files were imported, and the sequence of the target gene amplicons was used as reference sequence for alignment.

## Western blot

Melanoma cells were lysed using the M-PER mammalian protein reagent (Fermentas) with protease inhibitors (Thermo Scientific). The protein concentration was spectrophotometrically measured by a Bradford-based assay using Pierce BCA protein assay kit (Thermo Scientific) according to manufacturer's protocol. Laemmli buffer was added and lysates were boiled at 95 °C for 5 min. Then, 10 μg of protein was loaded and separated according to size by SDS–PAGE gel electrophoresis on a 3% stacking and 10% polyacrylamide gel. Proteins were transferred to polyvinyl difluoride membranes with a 0.2 μm pore size (GE Healthcare) by means of wet blotting for 1 h. Unspecific binding was blocked with 5% skimmed milk in PBS with Tween for 1 h. Blots were stained with a goat polyclonal Trp1 antibody (1:1,000, Novus Biologicals) overnight at 4 °C. Next, the blots were incubated with anti-goat IgG HRP (1:2,000, Santa Cruz) for 1 h at room temperature. Horseradish peroxidase conjugated β-actin (200 μg ml$^{-1}$) was used as loading control. Bound antibody was detected by SignalFire ECL reagent (Cell Signaling Technology) and chemiluminescence was visualized using an Octoplus QPLEX imager (NH DyeAgnostics).

## Retroviral transduction

To generate tagBFP, mCherry and OVA-tagBFP-expressing cell lines, retroviruses were produced by transfecting human embryonic kidney 293T cells with the retroviral packaging constructs pCMV-gag-pol and pMD.2G (expressing VSVg) and the retroviral plasmids pRp-tagBFP, pRp-mCherry and pRp-OVA-tagBFP, respectively, according to standard protocols. Retrovirus-containing supernatant was used to transduce the target cell lines and antibiotic selection of transduced cells was started 48 h after transduction using 10 μg ml$^{-1}$ Puromycin.

## Tumour transplantation experiments

For tumour inoculation, a total of $2 \times 10^5$ cells were injected intracutaneously (i.c.) into the shaved flanks or hindlegs of mice with a 30G ($0.3 \times 13$ mm) injection needle (BD). Tumour development was monitored by inspection and palpation. Tumour sizes were measured three times weekly and presented as mean diameter. Mice were euthanized when tumours exceeded 15 mm mean diameter or when mice showed signs of sickness in adherence with the local ethical regulations. All animal experiments were performed in groups of four to six mice and repeated independently at least twice.

## ACT therapy protocol

ACT therapy was performed as previously described[23,24]. In brief, when transplanted melanoma cell lines reached a mean diameter of 3–5 mm, mice were preconditioned for ACT by a single intraperitoneal (i.p.) injection of 2 mg (roughly 100 mg kg$^{-1}$) of cyclophosphamide in 100 μl of PBS 1 day before intravenous (i.v.) delivery of splenocytes isolated from TCR-transgenic Pmel-1 and/or TRP-1 donor mice harbouring naïve Pmel-1/gp100-specific CD8$^+$ T cells and/or naïve TRP-1-specific CD4$^+$ T cells (in 100 μl of PBS). Unless otherwise indicated, we transferred splenocytes containing $5 \times 10^5$ antigen-specific T cells. The adoptively transferred T cells were stimulated in vivo by a single i.p. injection of $2.5 \times 10^8$ PFU of the recombinant adenoviral vaccine Ad-PT in 100 μl of PBS. On day 3, 6 and 9 after T cell transfer, tumours were injected with 50 μg of CpG 1826 (MWG Biotech) and 50 μg of polyinosinic:polycytidylic acid (polyI:C, Invivogen) diluted in 100 μl of distilled water. Seven days after T cell transfer, blood was taken routinely from the *Vena facialis* to confirm successful expansion of transferred T cells by flow cytometry.

## Supplementary in vivo treatments

NK-cell depletion was performed by a single i.p. injection of 200 μg anti-NK1.1 antibody (clone PK136, BioXCell) in 100 μl, diluted in pH 7.0 Dilution Buffer (BioXCell). CD8$^+$ T cell depletion was performed by i.p. injections of initially 100 μg, followed by weekly injections of 50 μg of anti-CD8 antibody (clone 2.43, BioXCell). MHC-II blockade was performed by a single i.v. injection of 500 μg of anti-MHC-II antibody (clone Y-3P, BioXCell) directly after inducing anaesthesia for 2P-IVM and roughly 30 to 60 min before data acquisition. IFNγ blockade was performed by weekly i.p. injection of 500 μg of anti-IFNγ antibody (clone XMG1.2, BioXCell) in 100 μl, diluted in pH 8.0 buffer. Monocyte depletion was performed by i.p. injections of 20 μg of anti-CCR2 (clone MC21, provided by M. Mack) for five consecutive days. Neutrophil depletion was performed by i.p. injections of 100 μg of anti-Ly6G (clone 1A8, BioXCell) every fifth day. Inhibition of iNOS was performed by daily i.p. injection of 200 μg of L-NIL (Cayman Chemicals) diluted in 100 μl of PBS.

## Flow cytometry

Immunostaining of single-cell suspensions was performed according to standard protocols. Single suspensions were incubated with anti-CD16/CD32 (1:300, Biolegend) before staining with fluorochrome-conjugated monoclonal antibodies CD45-APC Fire 750 (1:1,600), CD11c-APC (1:200), F4/80-PE (1:300), CD11b-BV711 (1:200), Ly6C-PE-Cy7 (1:2,000), CD45R-PE (1:1,000), CD3ε-BV421 (1:500), CD4-BV605 (1:500), NK1.1-APC (1:400), CD45-FITC (1:1,000), F4/80-APC (1:200), Ly6C-BV421 (1:800), iNOS-PE, (1:300), I-A/I-E-BV510 (1:800), CD45-BV711 (1:200), CD11c-APC Fire 750 (1:100), Siglec H-FITC (1:400), CD4-PE (1:1,600), CD11b-PE-Cy7 (1:2,000), Ly6G-PE (1:800), CD3ε-BV711 (1:100), CD8α-APC Fire 750 (1:1,600), H2-Kb-PE (1:500), I-A/I-E-APC (1:2,000), CD3ε-FITC (1:100), CD335-APC (1:100), CD8α-PE (1:800), Vβ14-FITC (1:2,000), T-bet-PeCy7 (1:200) and Foxp3-Alexa Fluor 647 (1:100). Intracellular staining was carried out using a Fixation/Permeabilization Solution Kit (BD or Biolegend). Single-cell suspensions from tumours were

first stained with antibodies against-cell-surface antigens, then fixed and permeabilized, followed by intracellular staining. Dead cell exclusion was performed using 7-aminoactinomycin or propidium iodide. All data were acquired with an Attune NxT acoustic focusing flow cytometer (ThermoFisher). Gating and subsequent analyses were performed using FlowJo v.10.8.1 for Windows (Tree Star, Inc.). Fluorescence-activated cell sorting was performed using an Aria III (BD Biosciences).

### Quantification of tumour-infiltrating immune cells

To quantify the abundance of immune cell subpopulations in tumour tissues, 2,000 cells of interest per biological sample were concatenated to a single FCS file. The *t*-distributed stochastic neighbor embedding (*t*-SNE) plots were generated in FlowJo using the opt-SNE learning configuration[57]. The vantage-point tree *K*-nearest-neighbours algorithm and the Barnes–Hut gradient algorithm were set to 1,000 iterations, 30 perplexity and 840 learning rate. Immune cell subpopulations were annotated on the basis of heatmaps for characteristic marker combinations and their percentage in the tumour was calculated.

### Analysis of tumour cell MHC expression and antigen recognition by $CD4^+$ T cells

To quantify the expression of MHC molecules, tumour cells were pretreated with 100 U ml$^{-1}$ recombinant murine IFNγ (Peprotech) for 72 h and then analysed by flow cytometry. To assess antigen recognition by $CD4^+$ T cells, TRP-1 TCRtg mice were immunized with Ad-PT and subsequently injected with 50 µg of CpG and 50 µg of polyI:C i.c. 3 and 6 days after immunization. TRP-1 $CD4^+$ T cells were isolated from the spleen and purified by two rounds of magnetic cell sorting (Miltenyi). Direct antigen recognition was determined by coculturing purified $CD4^+$ T cells with IFNγ pretreated HCmel12 cells. Antigen recognition in proxy was assessed by initially generating bone marrow-derived dendritic cells with recombinant GM-CSF and IL-4 (Peprotech) as previously described. After 1 week, differentiated bone marrow-derived dendritic cells were then pulsed overnight with HCmel12 lysate, before coculture with purified $CD4^+$ T cells. For both direct and myeloid cell-dependent antigen-recognition assays, the production of IFNγ from the $CD4^+$ T cells was measured 16 h after coculture by intracellular cytokine staining using flow cytometry according to standard protocols.

### Calculations of absolute immune cell counts in tumour tissues

Tumours were excised with tweezers and scissors, then weighed using the Entris 224-1S analytical balance (Sartorius). Single-cell suspensions were created mechanically using 5-ml syringe plungers (BD) and 70 µm cell strainers (Greiner). After immunostaining, cells were suspended in a defined volume and analysed on the Attune NxT acoustic focusing flow cytometer that uses a unique volumetric sample and sheath fluid delivery system allowing for accurate measurements of the number of cells analysed in a defined sample volume. The total number of viable $CD45^+$ immune cells in an individual tumour can then be derived by multiplying the number of $CD45^+$ immune cells counted in a defined sample volume with the total volume of the respective single-cell suspension. Division of this total number by the total weight of the tumour yields the absolute immune cell count per mg tumour tissue. The absolute count of various immune cell subpopulations was calculated from their relative percentage in viable $CD45^+$ immune cells.

### Immunofluorescence microscopy

Tumours were harvested on day 5 after ACT and fixed in 4% paraformaldehyde for 24 h, then dehydrated in 20% sucrose before embedding in optimal cutting temperature freezing media (Sakura Finetek). Next, 6 µm sections were cut on a CM305S cryostat (Leica), adhered to Superfrost Plus slides (VWR) and stored at −20 °C until further use. When thawed, slides were either fixed with ice-cold acetone and stained with rat anti-mouse I-A/I-E (1:50) and anti-rat IgG-Alexa Fluor 594 (1:100) or directly mounted with Vectashield Antifade Mounting Medium (Vector Laboratories). Images were acquired on an Axio Imager.M2 with a Colibri 7 LED illumination system (Zeiss) and analysed with ImageJ v.1.52i (http://imageJ.nij.gov/ij).

### Intravital two-photon microscopy

Mice were anaesthetized with 100 mg kg$^{-1}$ ketamine and 10 mg kg$^{-1}$ xylazine i.p., complemented by 3 mg kg$^{-1}$ acepromazine s.c. after the onset of anaesthesia. The animals were placed and fixed to a heated stage. Transparent Vidisic carbomer gel was applied to moisten the eyes during anaesthesia. The hind leg was fixed in an elevated position and the skin covering the melanoma was detached using surgical scissors and forceps. One drop of transparent Vidisic carbomer gel was used on the exposed site as mounting medium. Two component STD putty (3M ESPE) placed on both sides of the leg was used create a level surface using a 24 × 60 mm cover slip, which was gently pressed on the putty in a way that the coverslip made slight contact with the exposed site, without exerting pressure on the tumour. After complete polymerization of the putty, the mice were transferred onto a 37 °C heating plate under the two-photon microscope.

Imaging was performed using distilled water or transparent Vidisic carbomer gel as immersion liquid with a W Plan-Apochromat ×20/1.0 DIC VIS-IR objective mounted to a Zeiss LSM 700 upright microscope with the ZEN software environment (v.2.1, Zeiss), or a LaVision TrimScope mounted to an Olympus BX50WI fluorescence microscope stand and a XLUMPlanFl ×20/0.95 objective. Excitation on the LSM700 setup was performed with Mai Tai DeepSee (tuned to 800 nm) and Insight X3 (tuned to 980 nm) Ti:Sa oscillators (both from Spectra-Physics). Fluorescence signals were read out on a long-pass dichroic mirror detector cascade as follows: dsRed, 980 nm excitation and 555 nm dichroic transmission with a 587/45 nm bandpass filter; Venus, 980 nm excitation and 520 nm dichroic transmission with a 534/30 nm bandpass filter; second-harmonic generation, 800 nm excitation and 445 nm dichroic deflection unfiltered; tagBFP, 800 nm excitation and 490 nm dichroic deflection with a 485 nm short-pass filter; and eGFP, 980 nm excitation, 520 nm dichroic deflection and 490 nm dichroic transmission with a 525/50 nm bandpass filter. Excitation on the TrimScope setup was performed with a Chamaeleon Ultra II Ti:Sa oscillator tuned to 880 nm. Fluorescence signals were read out with a double split detector array with a 495 nm main dichroic mirror and 445 and 520 nm secondary dichroic mirrors (all long-pass) as follows: second-harmonic generation, 495 nm and 445 nm dichroic deflection unfiltered; tagBFP, 495 nm dichroic deflection and 445 nm dichroic transmission with a 494/20 nm bandpass filter; eGFP, 495 nm dichroic transmission and 520 nm dichroic deflection with a 514/30 nm bandpass filter; and Venus, 495 nm and 520 nm dichroic transmission with a 542/27 nm bandpass filter. Non-descanned photomultiplier tubes (for second-harmonic generation, dsRed and Venus in all setups, and for eGFP and tagBFP in the TrimScope setup) and high sensitivity detectors (for tagBFP and eGFP in the Zeiss setup) were used for signal collection.

Typically, three to four representative fields of view of 353 µm$^2$ size in *x*-, *y*- and a *z*-range of 48 to 60 µm with 4 µm step sizes were chosen for data acquisition. *Z*-stacks were captured in 60–90 s intervals and individual video length was 15–30 min. Data analysis was performed with the Bitplane Imaris software (v.8.3 to 9.7). T cells were identified using the Imaris spot function. Tumour area was identified using the surface function with low surface detail. CD11c-Venus cells were identified using the surface function with high detail. T cell speed was calculated using the Imaris software. Cells were considered arrested when speed was less than 2 µm min$^{-1}$. Contact duration was measured as the time that the distance between the centre of mass of a T cell to the closest CD11c cell surface was less than 8 µm. Snapshot images of 3D rendering and

tracking were cropped, arranged and animated for time series using ImageJ v.1.52i (http://imageJ.nij.gov/ij).

## Cell preparation for scRNA-seq

Three individual tumours per group were harvested and processed into single suspensions. CD45$^+$ cells were enriched using a positive selection kit (Miltenyi). Next, individual samples were hashtagged with unique TotalSeq-B hashtag antibodies B0301-B0310 (1:300, Biolegend) and subsequently stained with fluorescently labelled antibodies. CD45$^+$CD11b$^+$Ly6G$^-$ cells were sorted with an Aria III fluorescence-activated cell sorter (BD). Isolated cells were loaded onto one lane of a 10X Chromium microfluidics controller. cDNA of hashtag and gene expression libraries were amplified, and indices added by means of PCR. Sequencing was performed on an Illumina Novaseq on two lanes of a S1 cartridge with 150 bp read length in paired end mode. Reading depth was calculated to obtain roughly 50,000 reads per cell for the gene expression library and 5,000 reads per cell for the hashtag library.

## scRNA-seq data processing and hashtag-demultiplexing

The scRNA-seq data generated using 10X Genomics Chromium technology were aligned and quantified using the Cell Ranger Single-Cell Software Suite against the mm10 mouse reference genome. The raw, unfiltered data generated from Cell Ranger were used for downstream analyses. Quality control was performed on cells on the basis of the three metrics: total unique molecular identifier (UMI) count, number of detected genes and proportion of mitochondrial gene count per cell. Specifically, cells with less than 1,000 UMIs, 1,000 detected genes and more than 25% mitochondrial UMIs were filtered out. To remove potential doublets, cells with UMI count above 40,000 were removed. Subsequently, we demultiplexed the samples tagged with distinct hashtag-oligonucleotides using Solo[58]. After quality control, we normalized raw counts by their size factors using scran[59] and subsequently performed $\log_2$ transformation. The logarithmized and normalized count matrix was used for the downstream analyses.

## Dimensionality reduction, unsupervised clustering and differential gene expression analyses

Analysis of normalized data was performed using the scanpy Python package[60]. Initially, the 4,000 most highly variable genes were selected for subsequent analysis using scanpy.pp.highly_variable_genes with the parameter 'n_top_genes=4000'. Next, a principal component analysis was performed with 50 components using scanpy.tl.pca with the parameters 'n_comps=50, use_highly_variable=True, svd_solver='arpack''. Subsequently, dimensionality reduction was performed using UMAP with scanpy.tl.umap. Single cells were automatically assigned using R package SingleR[61], with transcriptomes from the Immunological Genome Project as a reference. Clustering of single cells by their expression profiles was conducted by using the Leiden algorithm running scanpy.tl.leiden with the parameter 'resolution=1.0'. Clusters with fewer than 20 cells were removed from further analysis. Differential gene expression was performed between cells classified as macrophages and monocytes from non-treated and CD4 ACT-treated mice using a hurdle model implemented in the R package MAST. Subsequent gene set enrichment analysis was performed using gene set enrichment analysis in preranked mode using the $\log_2$ fold change as a ranking metric. The IFN score was derived by calculating a $z$-score for all genes from the MSigDB gene set 'HALLMARK_INTERFERON_GAMMA_RESPONSE' for each cell.

## RNA velocity

For RNA velocity, count matrices of spliced and unspliced RNA abundances were generated using the velocyto workflow for 10X chromium samples, with the genome annotation file supplied by 10X Genomics for the mm10 genome and a repeat annotation file retrieved from the UCSC genome browser. Subsequent analyses were performed using scVelo[62]. The count matrices were loaded into the scanpy environment, merged with the previously generated anndata objects and normalized using scvelo.pp.filter_and_normalize. Next, moments for velocity estimation were calculated, gene-specific velocities were estimated and the velocity graphs were computed. Furthermore, a partition-based graph abstraction was generated with velocity-directed edges.

## Statistical methods

Statistical analyses and number of samples ($n$) are given in each figure legend. Mann–Whitney $U$-tests, unpaired two tail $t$-tests, analysis of variance (ANOVA) and log-rank tests were performed in Graphpad Prism (v.8).

## Reporting summary

Further information on research design is available in the Nature Portfolio Reporting Summary linked to this article.

## Data availability

The raw sequencing mouse scRNA-seq data are available at the NCBI GEO under the accession GSE230427 without restrictions. The normalized and logarithmized count matrix used for the subsequent analyses is also available at the NCBI GEO under the accession GSE230427 without restrictions. Human scRNA-seq data used in this study are available at the European Genome-Phenome Archive with the identifier EGAS00001006488, available for non-commercial research purposes on reasonable request and subject to review of a project proposal that will be evaluated by the VIB-UZL Data Access Committee. Source data are provided with this paper.

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

**Acknowledgements** We thank the following individuals for their support: S. Bonifatius, J. Herz, J. Leipold, A. Ziems, K. Beinhoff, R. Hartig and J. Dudeck for managing the mouse colony, performing tumour analyses and assisting for intravital microscopy and cell sorting; M. Mack for providing us with the anti-CCR2 mAb; J. Ruotsalainen for supporting viral vector production and CRISPR–Cas9 gene editing and D. Schanze for sequencing of CRISPR–Cas9 engineered cells. T.T. was supported by funding from the German Research Foundation (grant nos. SFB854-P27 and SFB704-P22, FOR5489-P8, TU 90/10-1) and the German Cancer Aid (grant nos. 70112525 and 70114549). A.J.M. was supported by funding from the European Research

Council under the European Union's Horizon 2020 research and innovation programme (StG ImmProDynamics, grant agreement no. 714233) and the German Research Foundation DFG (grant nos. SFB854-Z01, SFB854-B31). W.K. was supported by funding from the German Research Foundation DFG (grant no. CRC TRR 338). J.C.M. received funding within the Grand Challenges Program of VIB, from FWO/KOTK (grant no. G0B1622N), Neftkens foundation, KULeuven (C1 grant) and the Belgian Excellence of Science programme. A.C.B. and M.M. were funded by the Else Kröner-Fresenius Forschungskolleg Magdeburg (grant nos. 2017_Kolleg.07; TP3 and TP4).

**Author contributions** B.K., A.C.B., N.S., S.G., K.K., P.D., S.H., T.C.v.D.S., Y.F. and A.K. performed experiments and analysed data. B.K., N.S. and J. Peters generated cell lines. N.S., D.Y. and M.E. generated recombinant adenoviruses. M.M., A.D.B. and O.B. collected clinical data. A.D.B., J. Pozniak, F.R., J.-C.M. and R.G. performed scRNA-seq analyses. B.K., A.C.B., N.S., A.A., S.G., A.J.M. and T.T. designed experiments. A.A. and F.M.B. generated the MILAN images. B.K. and A.J.M. performed and analysed the intravital images and videos. B.K., A.C.B., N.S., A.D.B., E.G., S.K., D.M., H.K., J.-C.M., W.K., A.J.M. and T.T. contributed intellectual input and helped to interpret data. A.J.M. and T.T. led the research programme. B.K., A.C.B., A.D.B., W.K., A.J.M. and T.T. wrote the manuscript.

**Competing interests** The authors declare no competing interests.

**Additional information**
**Correspondence and requests for materials** should be addressed to Andreas J. Müller or Thomas Tüting.

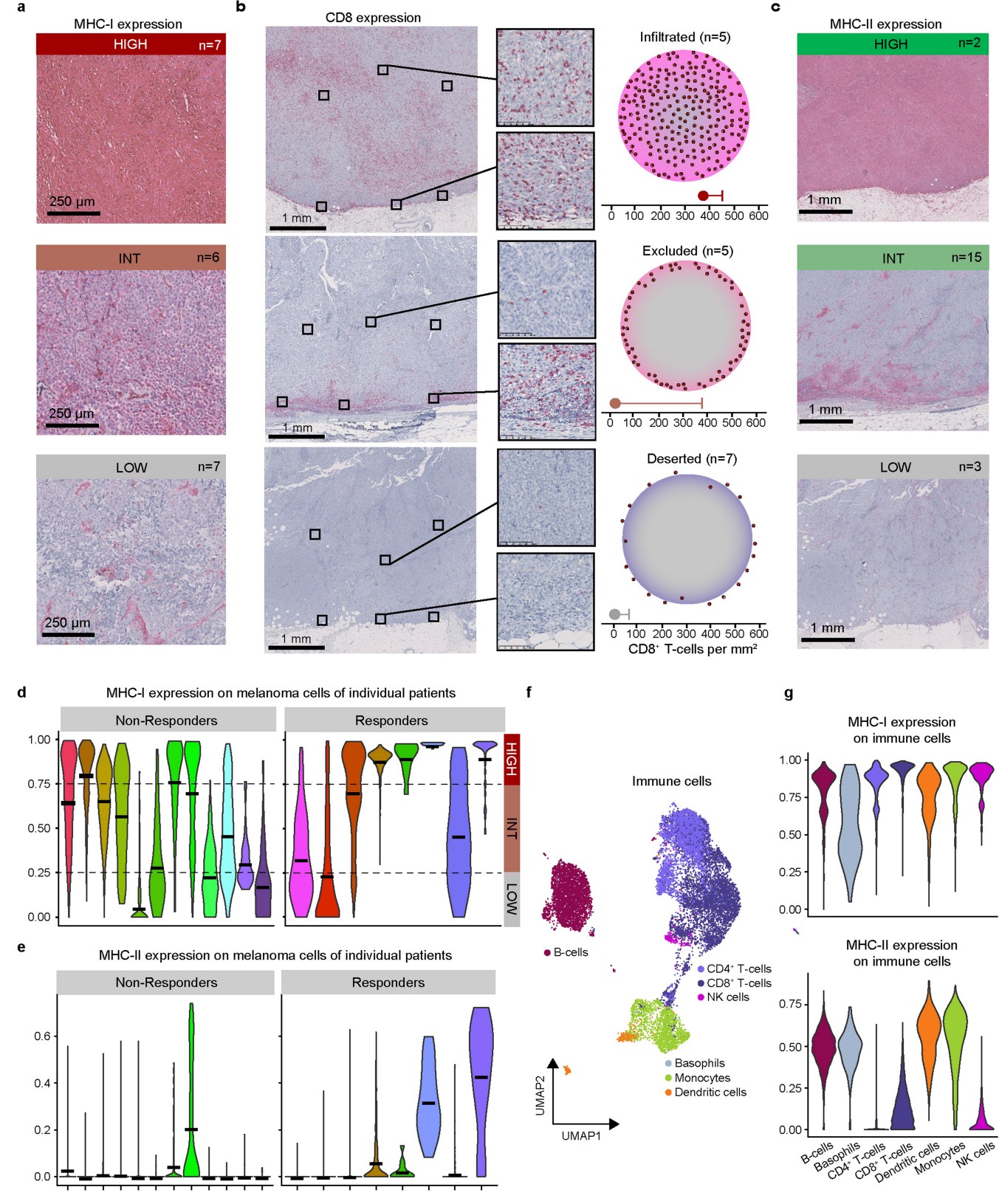

**Extended Data Fig. 1** | See next page for caption.

**Extended Data Fig. 1 | Landscapes of MHC expression, distribution of tumour infiltrating CD8+ T cells, and single cell transcriptomes of human melanoma metastases.** a, Representative immunohistochemical stains of human melanoma skin metastases with high, intermediate and low expression of MHC-I. b, Left: Overviews of representative immunohistochemistry stainings for CD8 and magnifications of 0.1 mm² squares in the tumour center and at the tumour invasive margins that were evaluated for the number of infiltrating CD8+ T cells; Right: graphical illustration of the three distinct patterns of immune infiltration observed. c, Representative immunohistochemical stains of human melanoma skin metastases with high, intermediate and low expression of MHC-II. d, MHC-I and (e) MHC-II gene set expression in single melanoma cells of an additional set of human melanoma metastases in skin (n = 5), subcutis (n = 4), and lymph nodes (n = 11), categorised into ICB therapy responders and non-responders. Median MHC-I gene set expression was used to categorize tumours according to high (>0.75), intermediate (0.25 - 0.75) and low (<0.25) expression levels. f, Uniform manifold approximation and projection (UMAP) clustering of the immune cell compartment in human melanomas annotated as indicated. g, MHC-I (top) and MHC-II (bottom) gene set expression in the indicated immune cell subpopulations.

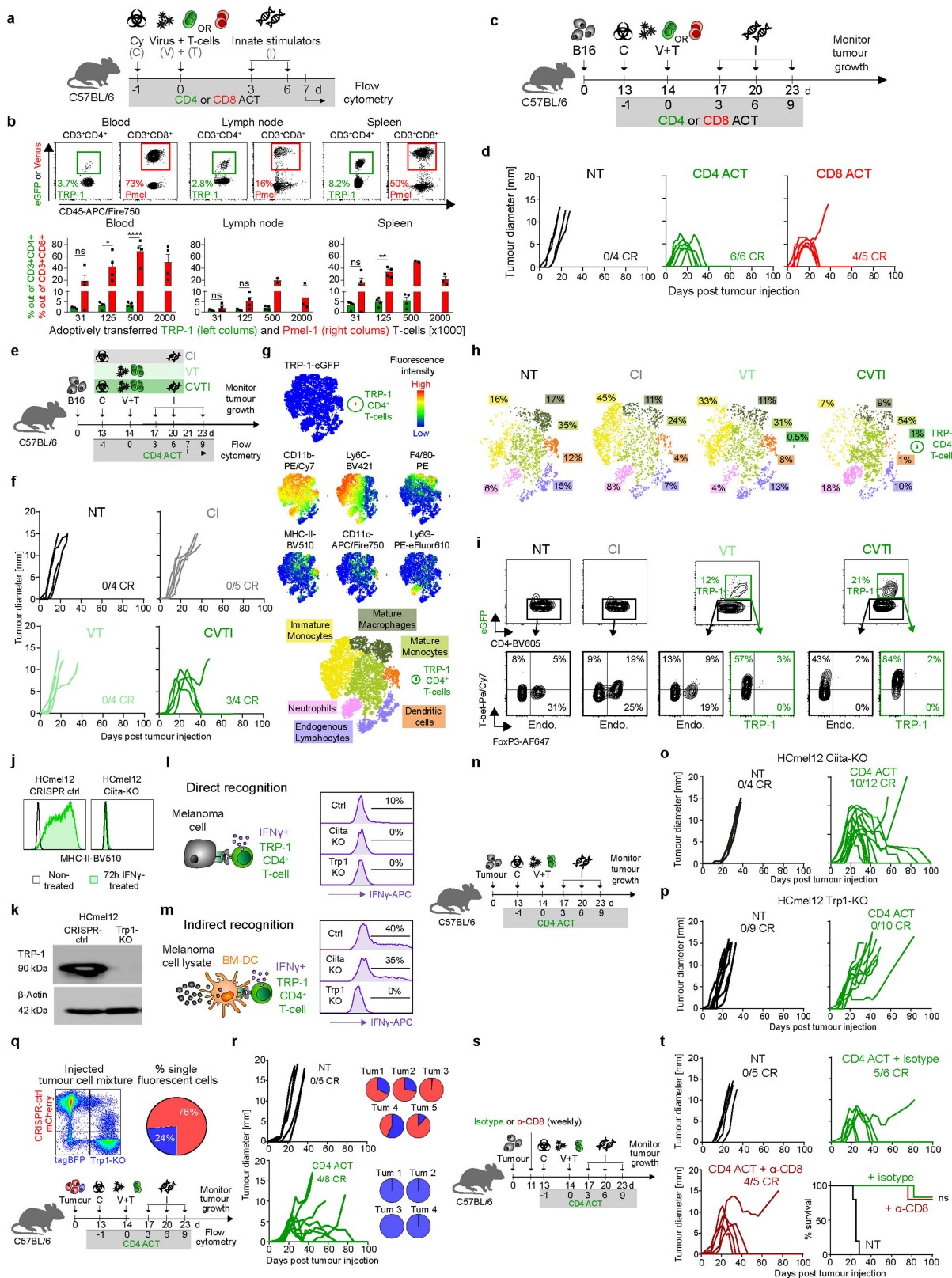

**Extended Data Fig. 2** | See next page for caption.

**Extended Data Fig. 2 | Establishment of an experimentally tractable adoptive cell transfer model to compare CD4+ and CD8+ T-cell effector functions against tumours and eradication of established MHC-II-deficient melanomas through indirect antigen recognition on MHC-II+ tumour-infiltrating immune cells.** a, Experimental protocol to assess the *in vivo* expansion of adoptively transferred CD8+ and CD4+ T cells. b, Representative flow cytometric contour plots with $0.5 \times 10^6$ transferred cells (top) and quantitation of Pmel-1 CD8+ and TRP-1 CD4+ TCRtg T-cell expansion in peripheral blood, lymph nodes and spleen 7 days after ACT (bottom, mean ± SEM from n = 2-4 biologically independent samples; ****p < 0.0001, **p = 0.0010, *p = 0.0146 by one-way ANOVA with Tukey's post-hoc test). c,e, Experimental protocols for adoptive cell transfer (ACT) immunotherapy of established tumours in mice and flow cytometric analyses of tumour-infiltrating immune cells. d,f, Individual tumour growth curves of established B16 melanomas treated as indicated. g, t-SNE heatmaps of multiparametric flow cytometry for B16 melanoma single cell suspensions showing the indicated markers (top) and corresponding annotation (bottom) of TRP-1 CD4+ T-cells (GFP+), immature monocytes (CD11b+ Ly6Chi), mature monocytes (CD11b+ Ly6Clo), mature macrophages (CD11b+ F4/80+), dendritic cells (CD11b+ MHC-II+ CD11c+), endogenous lymphocytes (CD11b- CD11c-), and neutrophils (CD11b+ Ly6G+). h, Annotated t-SNE plots quantifying the immune cell composition of B16 melanomas treated as indicated. i, Representative flow cytometric contour plots for the phenotyping of endogenous and transferred (VT, CVTI, green) CD4+ T-cells from mice treated as indicated. j, Representative flow cytometric histograms for MHC-II expression on indicated melanoma cells cultivated in the presence or absence of IFNγ. k, Representative western blot analysis for TRP-1 expression for the indicated melanoma cells (n = 2 biologically independent samples). Beta-actin was used as a loading control. For uncropped images see source data table. l, Graphical representation of direct and (m) indirect recognition of melanoma cells by CD4+ T-cells (left) and representative flow cytometry histograms showing IFNγ+ TRP-1 CD4+ T-cells following stimulation by the indicated melanoma cells (right). n, Experimental protocol and (o,p) individual tumour growth curves of established B16 melanomas treated as indicated. q, Injection of a tumour cell mixture consisting of ~75% HCmel12 CRISPR-ctrl cells and ~25% HCmel12 Trp1-KO cells (top) and treatment protocol (bottom). r, Individual tumour growth curves of mice bearing established melanomas and treated as indicated (left) and proportion of HCmel12 CRISPR-ctrl and HCmel12 Trp1-KO cells from escaping tumours (right). s, Experimental treatment protocol for depletion of CD8+ T-cells during CD4 ACT. t, Individual tumour growth curves and Kaplan-Meier survival graph of mice bearing established HCmel12 CRISPR-ctrl melanomas treated as indicated. Means between groups were statistically compared using one-way ANOVA with Tukey's post-hoc test. Survival was statistically compared using a log-rank Mantel-Cox test. NT, non-treated; C, cyclophosphamide; V, Ad-PT; T, TCRtg TRP-1 CD4+ T-cells; I, innate stimuli, polyI:C and CpG; CR, complete responders.

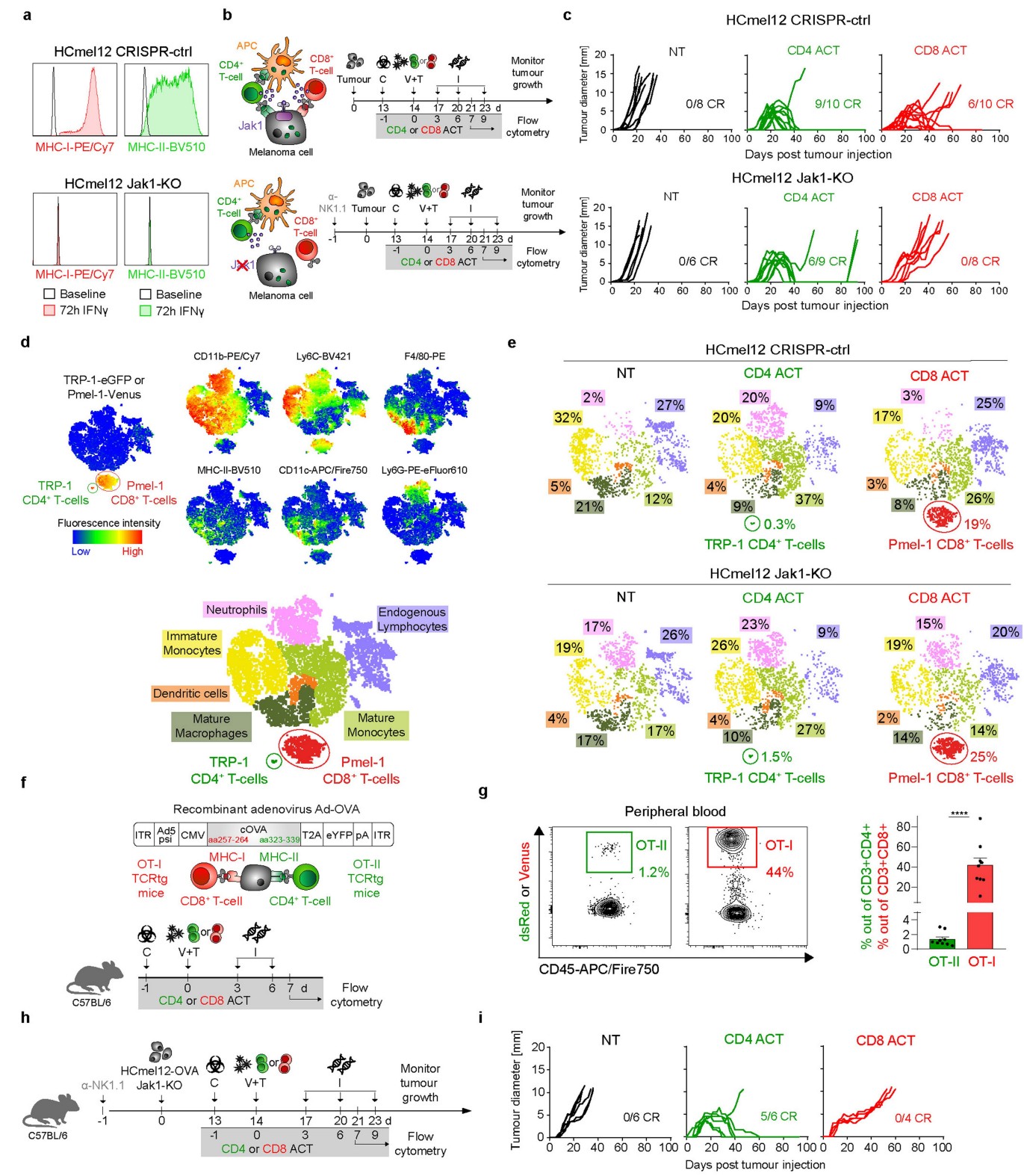

**Extended Data Fig. 3 |** See next page for caption.

**Extended Data Fig. 3 | Comparative evaluation of CD4+ and CD8+ T-cell effector functions against IFN-unresponsive tumours lacking MHC-I and MHC-II.** a, Representative flow cytometric histograms for MHC-I and MHC-II expression of indicated melanoma cells in the presence or absence of IFNγ. b, Graphical representation of the interaction phenotype of the indicated melanoma cells (left) and experimental treatment protocol (right). c, Individual tumour growth curves of mice bearing established melanomas and treated as indicated. d, t-SNE heatmaps of multiparametric flow cytometry for HCmel12 melanoma single cell suspensions showing the indicated markers (top) and corresponding annotation (bottom) of TRP-1 CD4+ T-cells (GFP⁺), Pmel-1 CD8+ T-cells (Venus⁺), immature monocytes (CD11b⁺ Ly6Cʰⁱ), mature monocytes (CD11b⁺ Ly6cˡᵒ), mature macrophages (CD11b⁺ F4/80⁺), dendritic cells (CD11b⁺ MHC-II⁺ CD11c⁺ F4/80⁻), endogenous lymphocytes (CD11b⁻ CD11c⁻), and neutrophils (CD11b⁺ Ly6G⁺). e, Quantification of the immune cell composition of HCmel12 CRISPR-ctrl and Jak1-KO tumours treated as indicated. f, Recombinant adenovirus Ad-OVA (top) and experimental protocol (bottom) to assess the *in vivo* expansion of adoptively transferred ovalbumin-specific CD8+ and CD4+ T-cells. g, Representative flow cytometric contour plots with 0.5 x 10⁶ transferred T-cells and quantitation of OT-I CD8+ and OT-II CD4+ TCRtg T-cell expansion in peripheral blood 7 days after ACT (mean ± SEM from n = 9 biologically independent samples, ****p < 0.0001 using a two-tailed paired t-test). h, Experimental protocol and (i) individual tumour growth curves of mice bearing established HCmel12-OVA Jak1-KO tumours, treated as indicated. Means between groups were statistically compared using a two-tailed paired t-test.

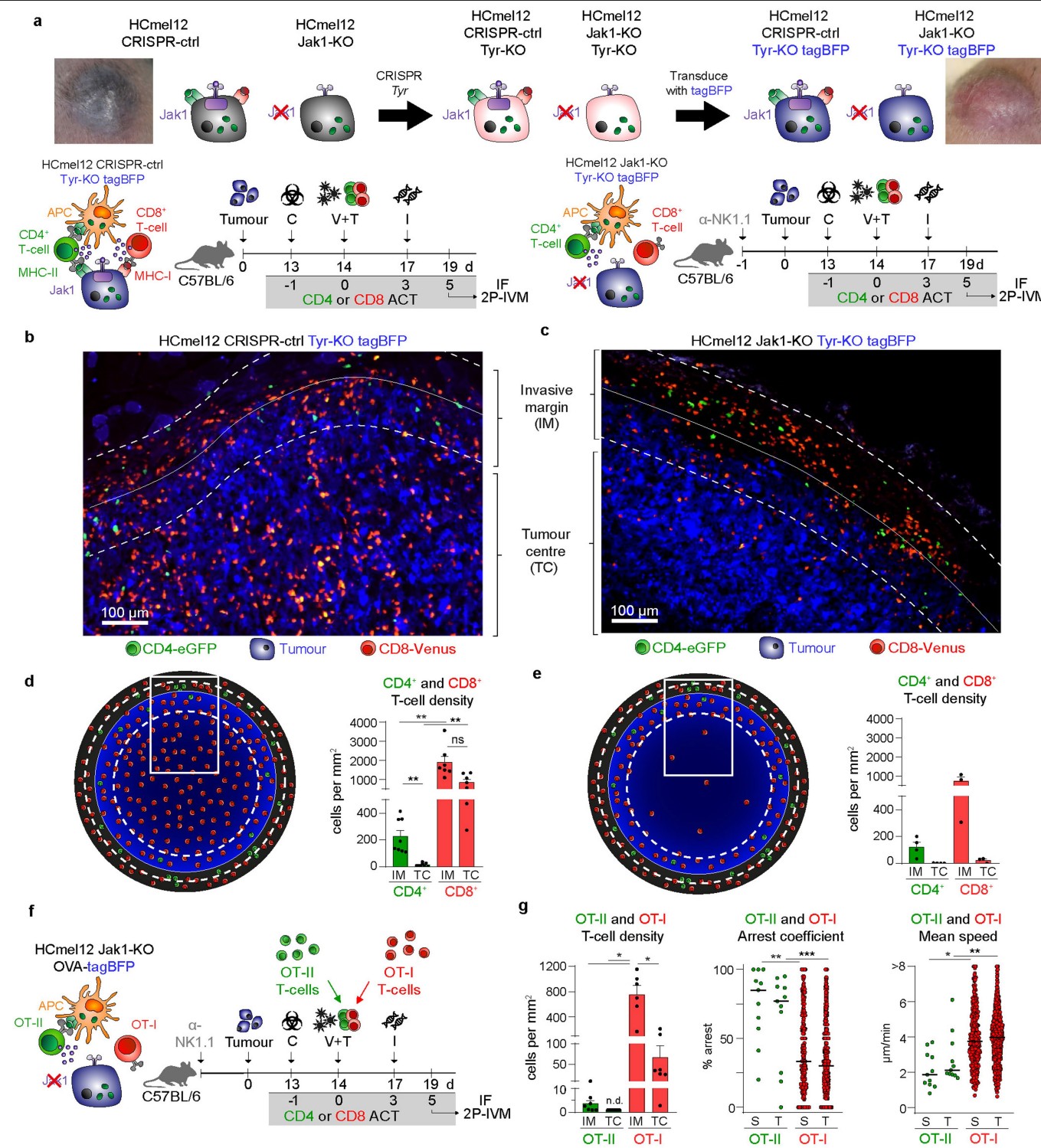

**Extended Data Fig. 4 | CD4+ effector T-cells show a different spatial distribution and migratory behaviour in tumour tissues when compared to CD8+ effector T-cells.** a, Macroscopic phenotype and graphical representation for the generation of amelanotic HCmel12 CRISPR-ctrl or Jak1-KO Tyr-KO tagBFP cell lines (top) and experimental treatment protocols (bottom). b,c, Representative fluorescence image for the distributions of Venus+ Pmel-1 CD8+ T-cells and eGFP+ TRP-1 CD4+ T-cells in indicated HCmel12 variants. d,e, Diagrammatic representation of the Venus+ Pmel-1 CD8+ T-cell and eGFP+ TRP-1 CD4+ T-cell distribution in a whole tumour cryosection of HCmel12 Tyr-KO CRISPR-ctrl (d) or Jak1-KO (e) melanomas (left) and corresponding quantitation (right) at the invasive margin (IM) and in the tumour centre (TC) (mean ± SEM for n = 6-7 biologically independent samples; CD4+ IM vs CD4+ TC **p = 0.0051, CD4+ IM vs CD8+ IM **p = 0.0035, CD4+ TC vs CD8+ TC **p = 0.068

using a one-way ANOVA with Tukey post-hoc). f, Diagrammatic representation and experimental protocol for treatment of HCmel12 Jak1-KO Tyr-KO OVA-tagBFP cells. g, Cell density (mean ± SEM from 7 biologically independent samples; CD4+ IM vs CD8 IM *p = 0.0122, CD4+ TC vs CD8+ IM *p = 0.0173, CD8 TC vs CD8 IM *p = 0.121 using a one-way ANOVA with Tukey post-hoc) at the IM and in the TC and arrest coefficient (***p = 0.0005, **p = 0.0046 using a Kruskal-Wallis test with Dunn's multiple comparison test) and mean speed (**p = 0.0012, *p = 0.0499 using a Kruskal-Wallis test with Dunn's multiple comparison test) of adoptively transferred of adoptively transferred Venus+ OT-I CD8+ and dsRed+ OT-II CD4+ T-cells in the stromal (S) and tumoural (T) compartment at the invasive margin (11-794 cells examined from 3 independent experiments; bar indicates the median).

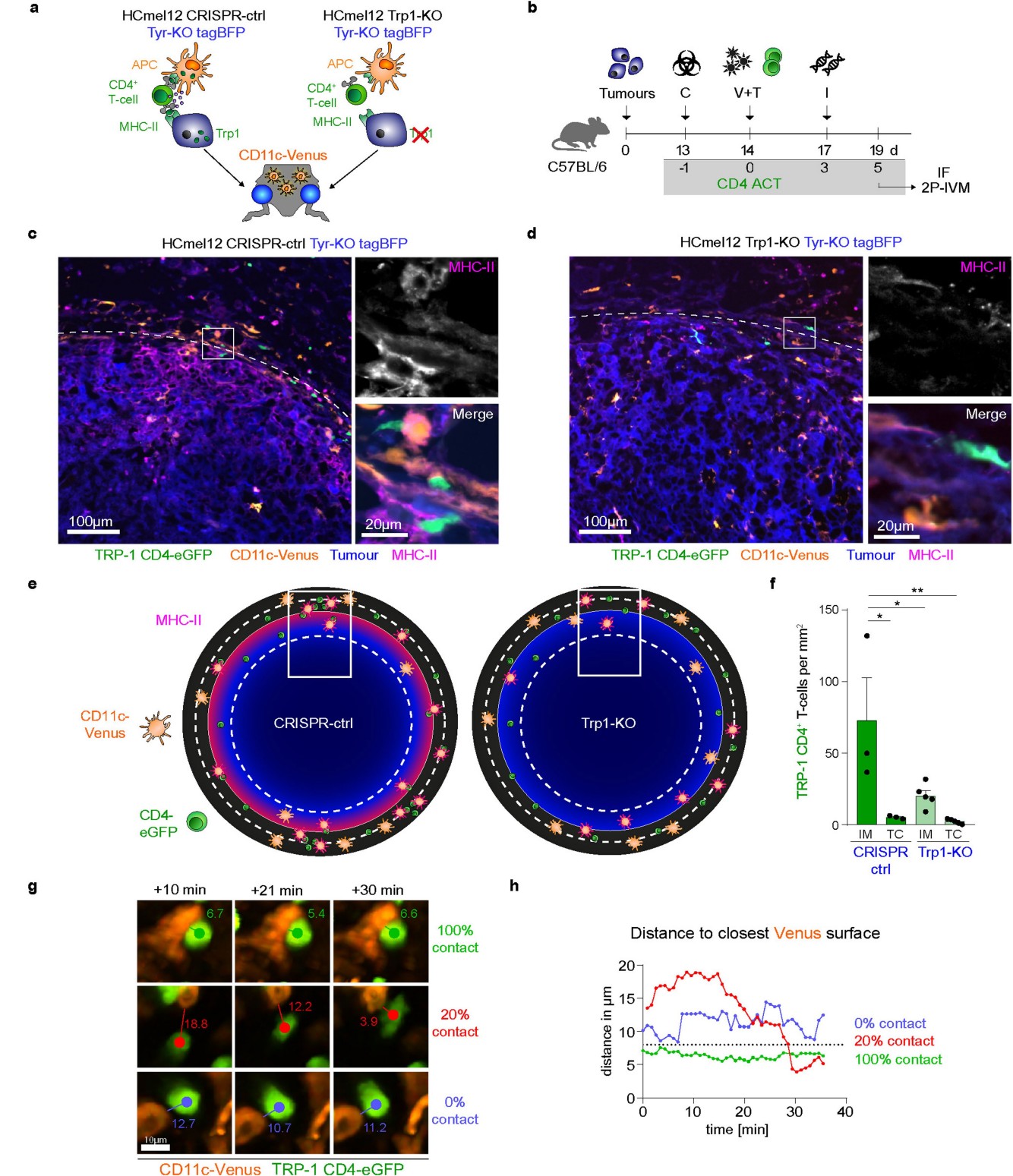

**Extended Data Fig. 5 | CD4+ effector T-cells cluster with MHC-II-expressing CD11c+ immune cells at the invasive margin of mouse melanomas.** a, Graphical representation of the interaction phenotype of the indicated HCmel12 variants and (b) experimental protocol to study antigen-specific interactions between eGFP+ TRP-1 CD4+ T-cells and CD11c+ cells in CD11c-Venus mice. c,d, Representative immunofluorescence images of MHC-II-stained cryosections from a (c) CRISPR-ctrl and a (d) Trp1-KO melanoma (mean ± SEM from n = 3-5 biologically independent samples). The dashed lines indicate the tumour border. e, Diagrammatic representation of MHC-II expression and interactions between eGFP+ TRP-1 CD4+ T-cells and CD11c-Venus antigen-presenting cells in CRISPR-ctrl and Trp1-KO melanomas. f, Density of eGFP+ TRP-1 CD4+ T-cells at the invasive margin (IM) and in the tumour centre (TC) of indicated tumours (mean ± SEM from n = 3-5 biologically independent samples, **p = 0.0037, CRISPR-ctrl IM vs Trp1-KO IM *p = 0.0267, CRISPR-ctrl IM vs TC *p = 0.0112). Means between groups were statistically compared using a one-way ANOVA with Tukey post-hoc. g,h Intravital 2P-microscopy images of three eGFP+ TRP-1 CD4+ T-cells and their distance to CD11c-Venus cells over time.

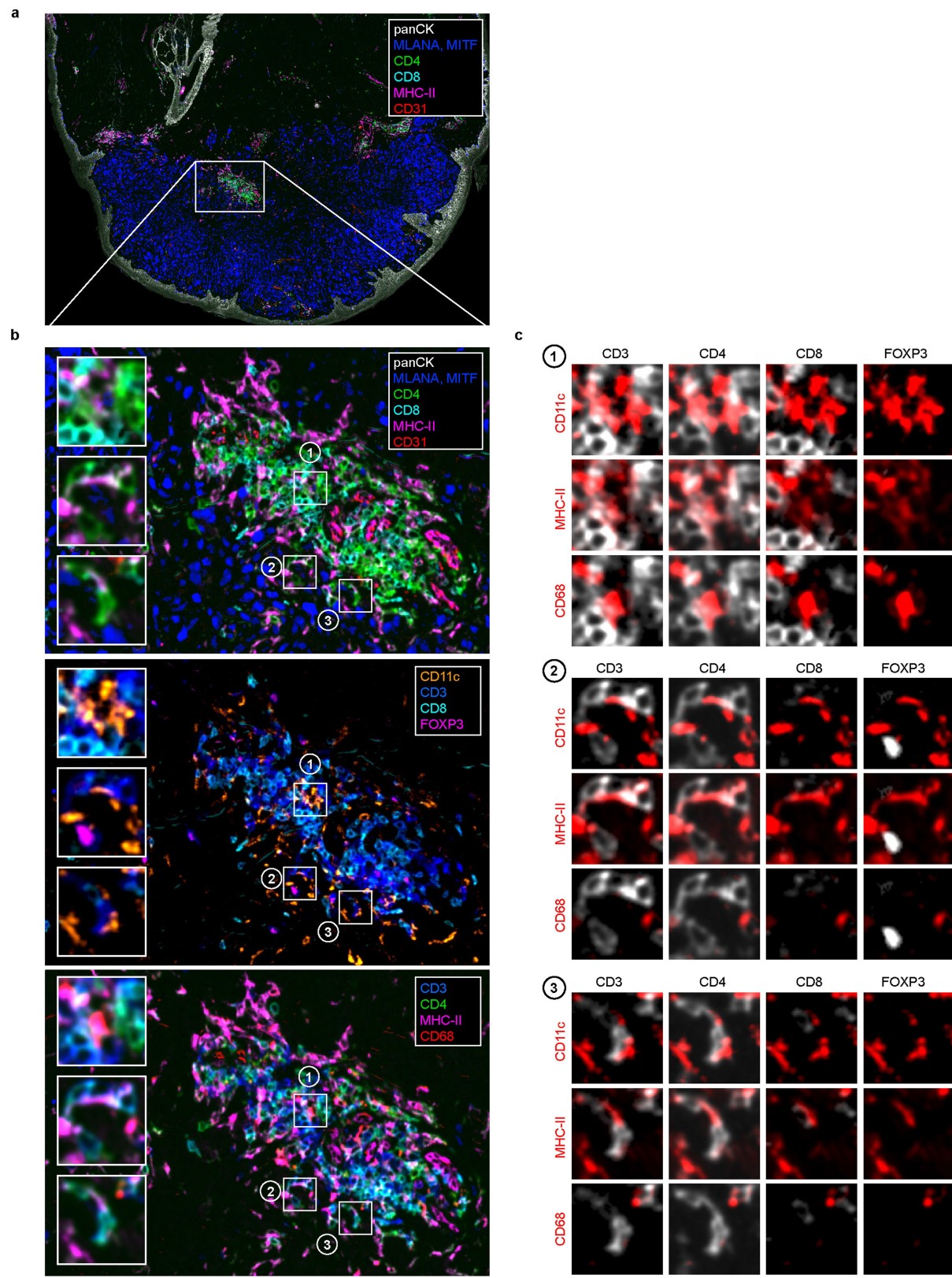

**Extended Data Fig. 6 | CD4+ effector T-cells cluster with antigen-presenting dendritic cells and macrophages at the invasive margin of a human melanoma.** Multiple iterative labeling by antibody neodeposition (MILAN) of a human melanoma obtained from (Pozniak, J., et al., 2022). An overview (a) over the whole tumour and a selected area of the tumour margin (b) are shown with multiple label combinations selected from the published panel. Insets (1-3) show exemplary sites of CD4+ T-cell juxtaposition with different myeloid subtypes expressing MHC-II, CD11c, and/or CD68. c, combinatorial overlays of different T-cell markers (white) with myeloid cell markers (red) in the insets (1-3).

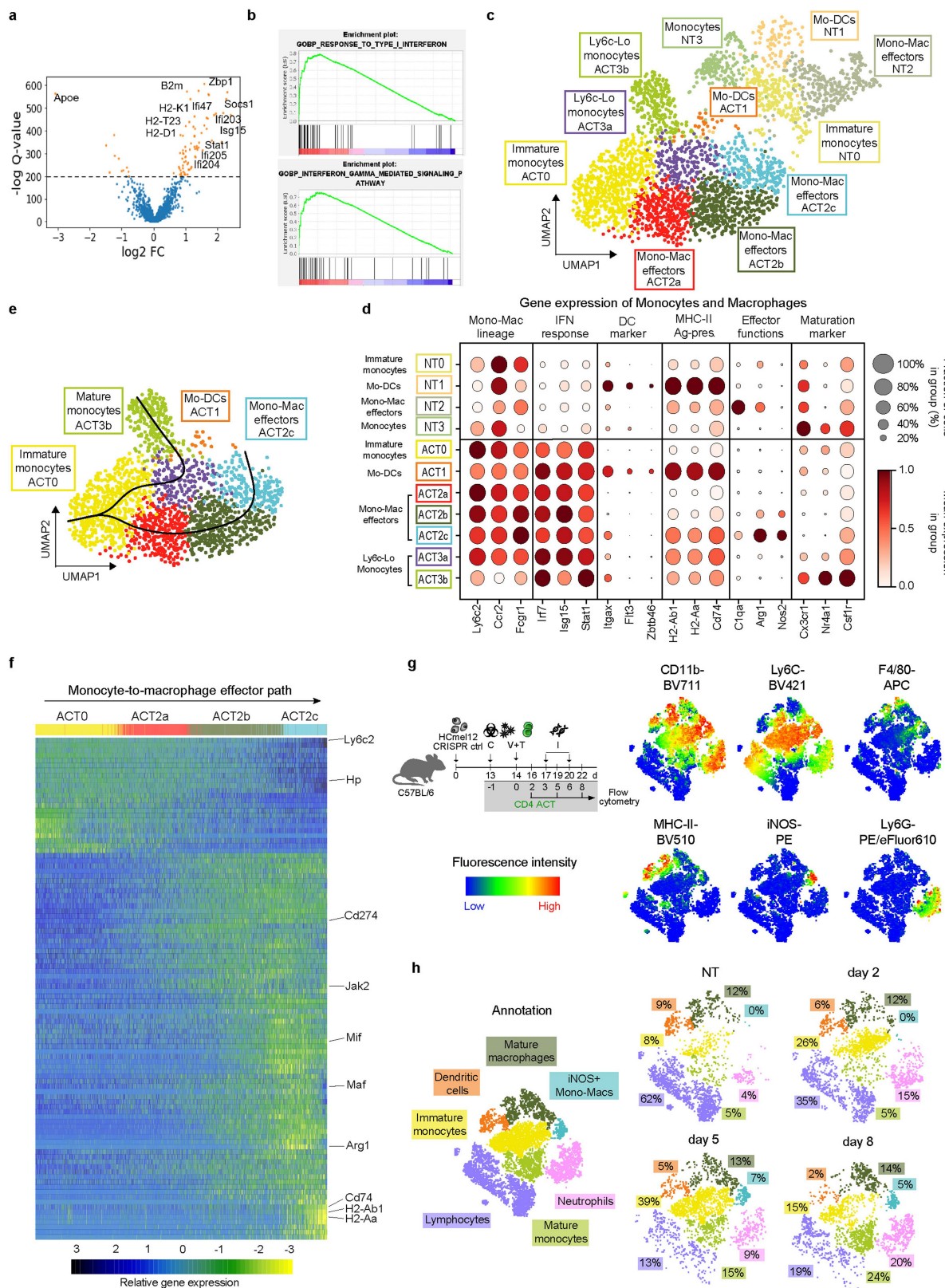

**Extended Data Fig. 7** | See next page for caption.

**Extended Data Fig. 7 | CD4 ACT therapy predominantly recruits immature monocytes into the tumour microenvironment and drives the acquisition of IFN-activated effector phenotypes.** a, Differentially expressed genes comparing samples from CD4 ACT-treated versus non-treated (NT) mice. Genes with –log Q-values >200 are shown in orange. b, Gene set enrichment analysis for the "GOBP_RESPONSE_TO_TYPE_I_INTERFERON" (top) and "GOBP_INTERFERON-GAMMA_MEDIATED_SIGNALING_PATHWAY" (bottom) gene sets. c, UMAP plots with Leiden clusters for monocytes and macrophages of CD4 ACT treated tumours. d, Corresponding expression levels and expression cell fractions of selected signature genes for the individual Leiden clusters and (e) pseudotime inference using slingshot. f, Heatmap of differentially expressed genes along the pseudotime trajectory of the indicated Leiden clusters representing the monocyte-macrophage (mono-mac) effector differentiation path. g, Experimental protocol (left), and t-SNE heatmaps of multiparametric flow cytometry for HCml12 melanoma single cell suspensions showing the indicated markers (right). h, Left: Corresponding annotation of immune of immature monocytes (CD11b+ Ly6Chi), mature monocytes (CD11b+ Ly6clo), mature macrophages (CD11b+ F4/80+ iNOS-), iNOS+ mono-macs (CD11b+ Ly6Chi iNOS+), dendritic cells (CD11b+ MHC-II+ CD11c+ F4/80-), endogenous lymphocytes (CD11b- CD11c-), and neutrophils (CD11b+ Ly6G+). Right: Annotated t-SNE plots quantifying the immune cell composition of HCmel12 melanomas at the indicated time points.

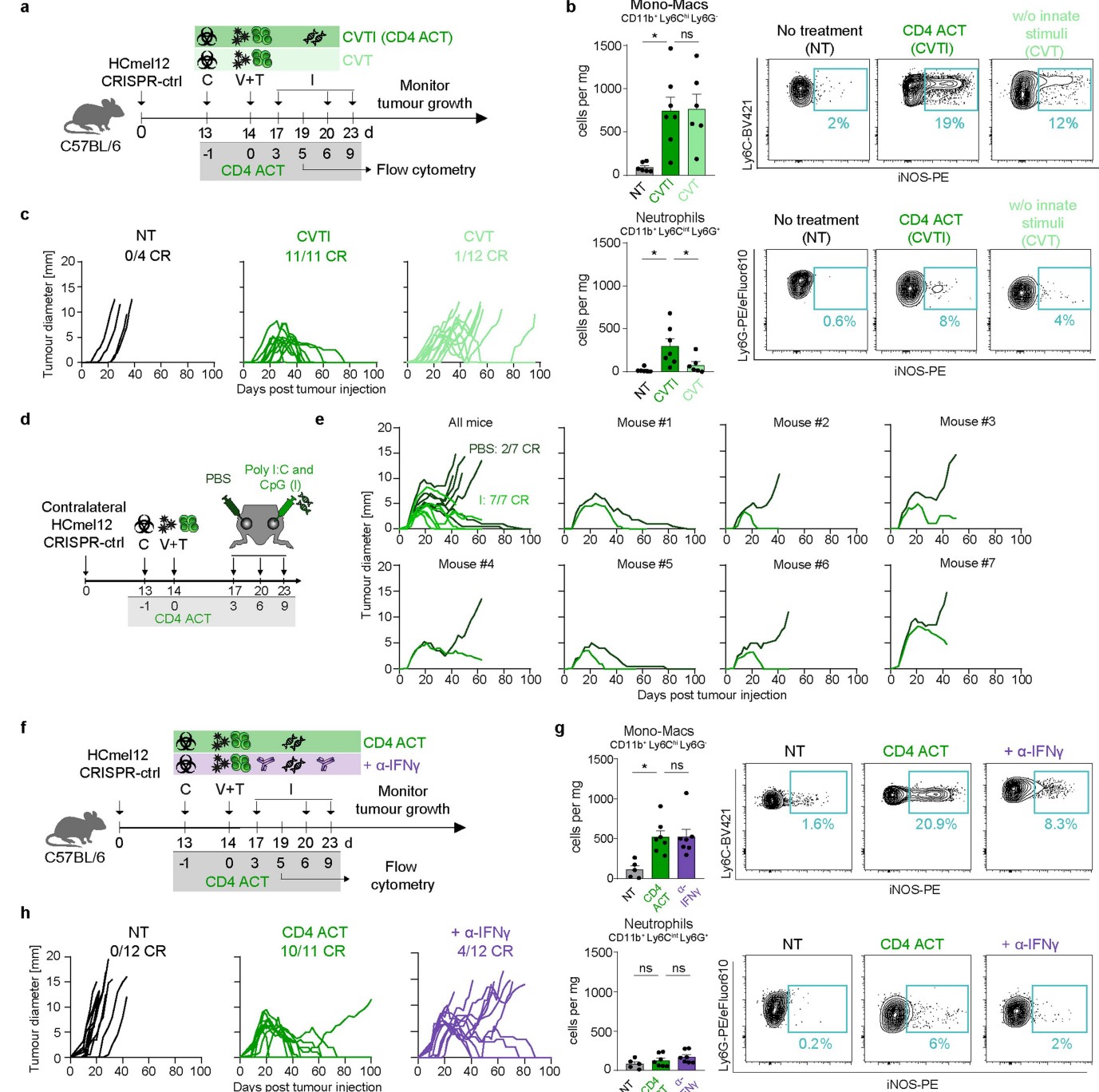

**Extended Data Fig. 8 | Robust IFNγ-dependent eradication of established melanomas requires local adjuvant innate immune stimulation.**
a, Experimental treatment protocol to address the impact of innate stimuli on myeloid cell activation and tumour control. b, Cell density (left) and representative contour plots quantifying the relative iNOS expression (right) in Ly6C^hi mono-macs (NT vs CVTI *p = 0.0109) and neutrophils (NT vs CVTI *p = 0.0114, CVTI vs CVT *p = 0.0474) in tumours 5 days post-ACT, treated as indicated (mean ± SEM from n = 6-7 biologcally independent samples). c, Individual tumour growth curves in mice bearing established melanomas and treated as indicated. d, Experimental treatment protocol to address the impact of local innate stimuli on tumour control and (e) individual growth curves of mice bearing contralateral HCmel12 tumours, treated as indicated. f, Experimental treatment protocol to address the impact of IFNγ-blockade. g, Cell density (left) and relative iNOS expression (right) in Ly6C^hi mono-macs and neutrophils in tumours 5 days post-ACT, treated as indicated (mean ± SEM from n = 6-7 biologically independent samples, *p = 0.0109). h, Individual tumour growth curves of mice bearing established melanomas and treated as indicated. Means between groups were statistically compared using a one-way ANOVA with Tukey post-hoc.

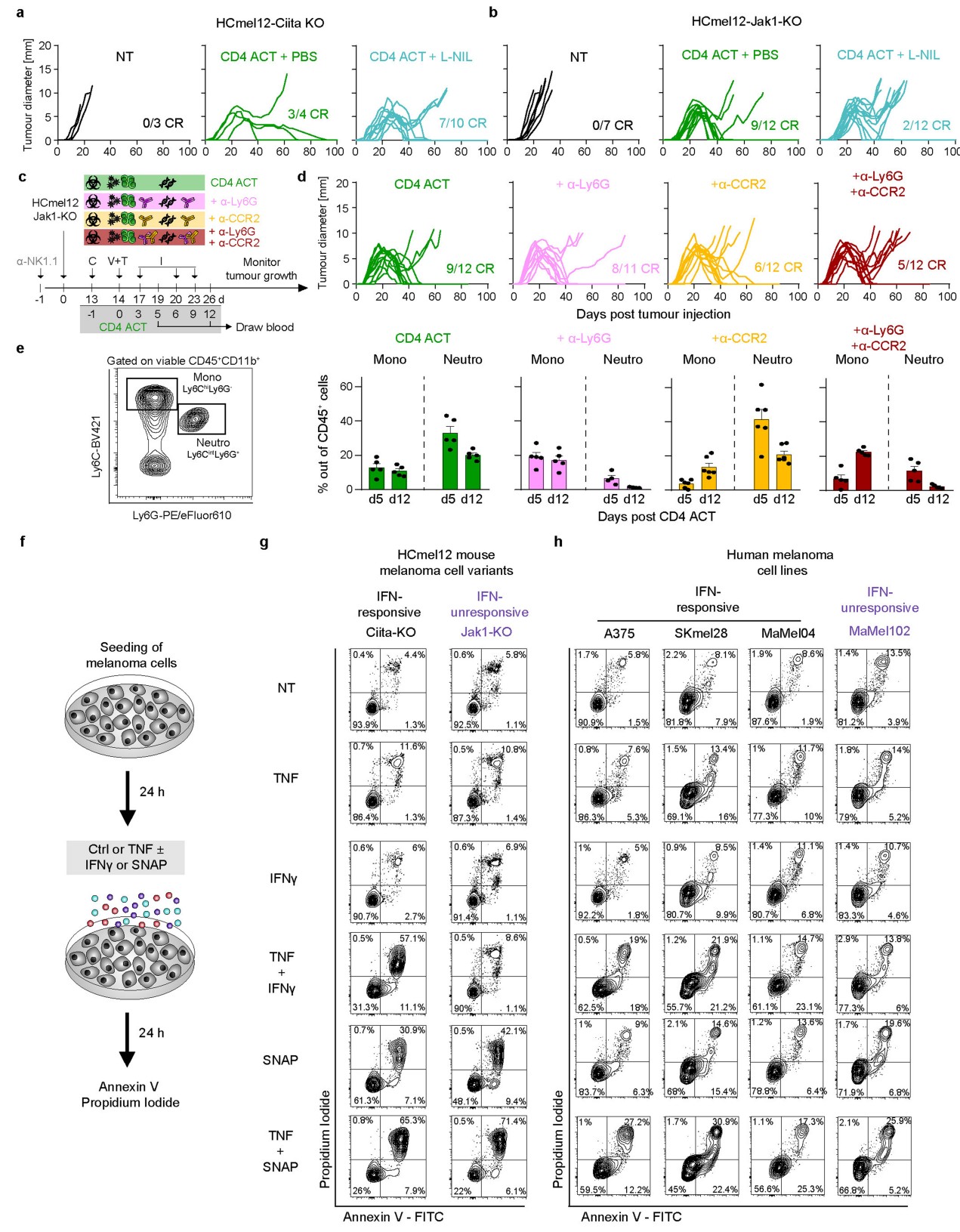

**Extended Data Fig. 9** | See next page for caption.

**Extended Data Fig. 9 | Chemical iNOS inhibition and antibody-mediated cell depletion of monocytes and neutrophils *in vivo* as well as treatment with the nitric oxide donor SNAP *in vitro* suggest a role for iNOS expressing myeloid cells in the control of established MHC-deficient and IFN-unresponsive melanomas.** a,b, Individual tumour growth curves of established HCmel12 Ciita-KO or HCmel12 Jak1-KO melanomas treated as indicated (L-NIL, iNOS-inhibitor). c, Experimental treatment protocol for antibody-mediated depletion of neutrophils and inflammatory monocytes. d, Individual tumour growth curves of established melanomas treated as indicated. e, Left: Representative gating strategy to evaluate the depletion of monocytes and neutrophils. Right: Flow cytometric percentages of monocytes and neutrophils in the blood 5 and 12 days post-CD4 ACT (mean ± SEM from 4-6 biologically independent samples). f, Experimental protocol to assess the ability of the inflammatory mediators TNF, IFNγ and the nitric oxide donor SNAP to induce melanoma cell death. g, h, Representative flow cytometric contour plots to assess cell death of mouse and human melanomas treated as indicated.

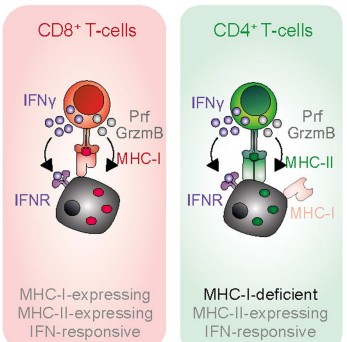

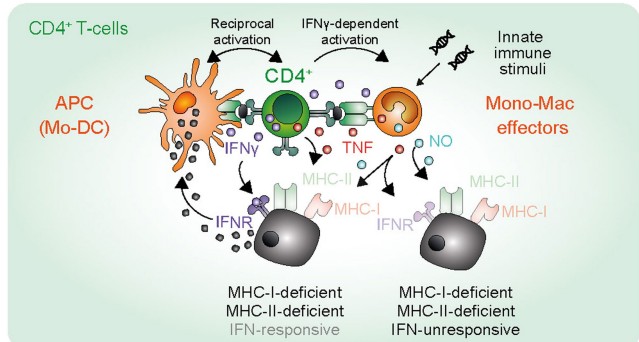

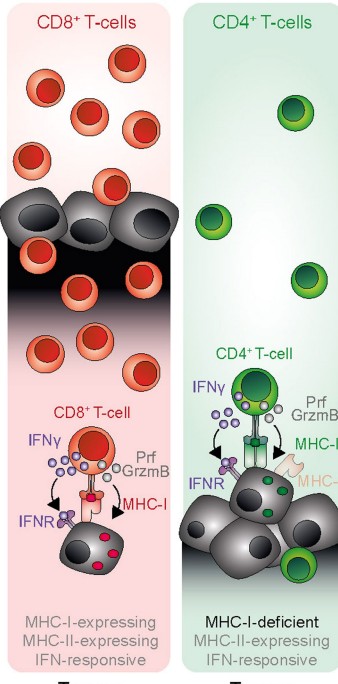

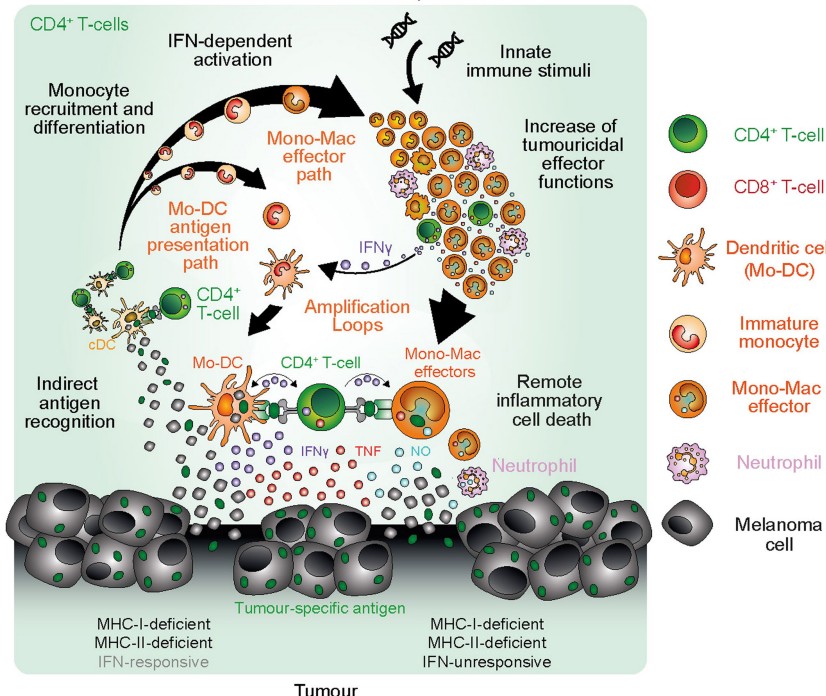

**Extended Data Fig. 10 | Spatial organisation and dynamics of T-cell effector functions in tumour tissues.** a, Graphical representation of direct antigen recognition and induction of cytolytic cell death. CD8+ and CD4+ effector T-cells can recognise their antigens as peptide epitopes presented by MHC-molecules on tumour cell surfaces and initiate direct killing through the release of cytolytic granules. b, Graphical representation of indirect antigen recognition and remote induction of inflammatory cell death. CD4+ effector T-cells also efficiently recognise tumour antigen on the surface of antigen-presenting cells (APC) including monocyte-derived dendritic cells (Mo-DC) and engage tumouricidal effector cells of the monocyte-macrophage lineage (Mono-Mac effectors) to initiate indirect cell death through the release of pro-inflammatory mediators. c, Spatial organisation and dynamics of direct induction of cytolytic cell death. CD8+ effector T-cells briskly infiltrate tumour tissues, where they directly interact with tumour cells (left), while CD4+ effector T-cells directly interact with tumour cells mainly near the invasive margin (right). d, Spatial organisation and dynamics of remote induction of inflammatory cell death. CD4+ effector T-cells cluster locally at the tumour invasive margin, where they indirectly recognise tumour antigen phagocytosed, processed and presented by dendritic cells. Activated CD4+ T-cells secrete IFNγ leading to the recruitment and activation of monocytes into the tumour tissue. Recruited monocytes phenotypically develop along differentiations path towards IFN-activated antigen-presenting (monocyte-derived dendritic cells, Mo-DCs) and tumouricidal effector phenotypes (monocyte-macrophage effector cells, Mono-Mac effectors). Mo-DCs additionally activate CD4+ T-cells and amplify monocyte recruitment, activation and differentiation. Innate immune stimulation promotes the Th1-directed differentiation of CD4+ T-cells and increases the tumouricidal functions of Mono-Mac effectors. CD4+ T-cell-derived IFNγ sensitises IFN-responsive melanoma cells for TNF-induced cell death. Myeloid cell-derived nitric oxide (NO) contributes to inflammatory cell death of IFN-unresponsive melanoma cells. Taken together, the induction of remote inflammatory cell death by CD4+ T-cells and tumouricidal myeloid cells eradicate IFN-responsive as well as IFN-unresponsive, MHC-deficient tumours that evade direct recognition and cytolytic killing.

## Reporting Summary

## Statistics

For all statistical analyses, confirm that the following items are present in the figure legend, table legend, main text, or Methods section.

| n/a | Confirmed | |
|---|---|---|
| ☐ | ☒ | The exact sample size (*n*) for each experimental group/condition, given as a discrete number and unit of measurement |
| ☐ | ☒ | A statement on whether measurements were taken from distinct samples or whether the same sample was measured repeatedly |
| ☐ | ☒ | The statistical test(s) used AND whether they are one- or two-sided<br>*Only common tests should be described solely by name; describe more complex techniques in the Methods section.* |
| ☐ | ☒ | A description of all covariates tested |
| ☐ | ☒ | A description of any assumptions or corrections, such as tests of normality and adjustment for multiple comparisons |
| ☐ | ☒ | A full description of the statistical parameters including central tendency (e.g. means) or other basic estimates (e.g. regression coefficient) AND variation (e.g. standard deviation) or associated estimates of uncertainty (e.g. confidence intervals) |
| ☐ | ☒ | For null hypothesis testing, the test statistic (e.g. *F*, *t*, *r*) with confidence intervals, effect sizes, degrees of freedom and *P* value noted<br>*Give P values as exact values whenever suitable.* |
| ☒ | ☐ | For Bayesian analysis, information on the choice of priors and Markov chain Monte Carlo settings |
| ☒ | ☐ | For hierarchical and complex designs, identification of the appropriate level for tests and full reporting of outcomes |
| ☒ | ☐ | Estimates of effect sizes (e.g. Cohen's *d*, Pearson's *r*), indicating how they were calculated |

*Our web collection on statistics for biologists contains articles on many of the points above.*

## Software and code

Policy information about availability of computer code

| Data collection | Attune NxT, Axio Imager.M2, LaVision TrimScope, Zeiss ZSM700 and Illumina Novaseq. |
|---|---|
| Data analysis | Graphpad Prism v8, Imaris v8.3 to 9.7, Flowjo v10.8.1, ImageJ 1.52i.<br><br>Python (version 3.9.9)<br>anndata (version 0.7.6)<br>anndata2ri (version 1.0.6)<br>attrs (version 21.2.0)<br>backcall (version 0.2.0)<br>certifi (version 2020.12.5)<br>cffi (version 1.14.5)<br>chardet (version 4.0.0)<br>CITE-seq-Count (version 1.4.4)<br>cmake (version 3.22.0)<br>cutadapt (version 3.5 )<br>cycler (version 0.10.0)<br>Cython (version 0.29.24)<br>decorator (version 4.4.2)<br>get-version (version 2.2)<br>gprofiler-official (version 1.0.0)<br>h5py (version 3.2.1)<br>idna (version 2.10)<br>iniconfig (version 1.1.1) |

ipykernel (version 5.5.4)
ipython (version 7.23.1)
ipython-genutils (version 0.2.0)
jedi (version 0.18.0)
Jinja2 (version 2.11.3)
joblib (version 1.0.1)
jupyter-client (version 6.1.12)
jupyter-core (version 4.7.1)
kiwisolver (version 1.3.1)
legacy-api-wrap (version 1.2)
llvmlite (version 0.36.0)
MarkupSafe (version 1.1.1)
matplotlib (version 3.4.2)
matplotlib-inline (version 0.1.2)
natsort (version 7.1.1)
networkx (version 2.5.1)
numba (version 0.53.1)
numexpr (version 2.7.3)
numpy (version 1.21.4)
openpyxl (version 3.0.9)
packaging (version 20.9)
pandas (version 1.2.4)
parso (version 0.8.2)
patsy (version 0.5.1)
pexpect (version 4.8.0)
pickleshare (version 0.7.5)
Pillow (version 8.2.0)
pluggy (version 0.13.1)
prompt-toolkit (version 3.0.18)
ptyprocess (version 0.7.0)
py (version 1.10.0)
pycparser (version 2.20)
Pygments (version 2.9.0)
pynndescent (version 0.5.2)
pyparsing (version 2.4.7)
python-dateutil (version 2.8.1)
pytz (version 2021.1)
pyxlsb (version 1.0.9)
pyzmq (version 22.0.3)
requests (version 2.25.1)
rpy2 (version 3.4.2)
scanpy (version 1.7.2)
scikit-learn (version 0.24.2)
scipy (version 1.6.3)
scvelo (version 0.2.4)
seaborn (version 0.11.1)
sinfo (version 0.3.4)
six (version 1.16.0)
statsmodels (version 0.12.2)
stdlib-list (version 0.8.0)
tables (version 3.6.1)
threadpoolctl (version 2.1.0)
toml (version 0.10.2)
tornado (version 6.1)
tqdm (version 4.60.0)
traitlets (version 5.0.5)
tzlocal (version 2.1)
umap-learn (version 0.5.1)
urllib3 (version 1.26.4)
wcwidth (version 0.2.5)
xlrd (version 1.2.0)
python-igraph (version 0.9.1)
leidenalg (version 0.8.4)
pytest (version 6.2.3)

R (version 4.0.4)
scran (version 1.18.7)
MAST (version 1.16.0)
SingleCellExperiment (version 1.12.0)
RcppAnnoy (version 0.0.16)
SummarizedExperiment (version 1.20.0)
Biobase (version 2.50.0)
GenomicRanges (version 1.42.0)
GenomeInfoDb (version 1.26.7)
IRanges (version 2.24.1)
S4Vectors (version 0.28.1)

BiocGenerics (version 0.36.1)
MatrixGenerics (version 1.2.1)
matrixStats (version 0.63.0)
gam (version 1.22)
foreach (version 1.5.1)
slingshot (version 1.8.0)
princurve (version 2.1.6)
glmnet (version 4.1-6)
Matrix (version 1.3-2)
RColorBrewer (version 1.1-3)
plyr (version 1.8.8)
ggplot2 (version 3.4.0)

For manuscripts utilizing custom algorithms or software that are central to the research but not yet described in published literature, software must be made available to editors and reviewers. We strongly encourage code deposition in a community repository (e.g. GitHub). See the Nature Portfolio guidelines for submitting code & software for further information.

## Data

Policy information about availability of data

All manuscripts must include a data availability statement. This statement should provide the following information, where applicable:
- Accession codes, unique identifiers, or web links for publicly available datasets
- A description of any restrictions on data availability
- For clinical datasets or third party data, please ensure that the statement adheres to our policy

The raw sequencing mouse scRNA-seq data are available at the NCBI GEO under the accession GSE230427 without restrictions. The normalized and logarithmised count matrix used for the subsequent analyses is also available at the NCBI GEO under the accession GSE230427 without restrictions.
Human scRNAseq data used in this study are available at the European Genome-Phenome Archive (EGA) with the identifier EGAS00001006488, available for non-commercial research purposes upon reasonable request and subject to review of a project proposal that will be evaluated by the VIB-UZL Data Access Committee.

# Field-specific reporting

Please select the one below that is the best fit for your research. If you are not sure, read the appropriate sections before making your selection.

☒ Life sciences ☐ Behavioural & social sciences ☐ Ecological, evolutionary & environmental sciences

For a reference copy of the document with all sections, see nature.com/documents/nr-reporting-summary-flat.pdf

# Life sciences study design

All studies must disclose on these points even when the disclosure is negative.

| | |
|---|---|
| Sample size | The determination of sample sizes for animal experiments was based on our experience with success of tumour engraftment and efficacy of therapeutic intervention in order to adhere to the 3R guidelines of the local Ethics Committee of the Office for Veterinary Affairs. Tumour treatment experiments involved 4-6 mice per group and were performed at least twice. Analyses of tumour immune cell infiltrates and intravital microscopy experiments involved a minimum of 3 mice per group. This yielded consistently reproducible and statistically significant results. Similarly, group sizes for in vitro experiments were determined based on prior knowledge of variation. |
| Data exclusions | No data was excluded from analysis. |
| Replication | Experiments were reliably reproduced and the number of experiments performed stated in methods and legends. Culminated and pooled data are shown where possible. Where representative data is shown, relevant experiments were repeated successfully at least twice with the exact number of repeats indicated in each case. Most experiments were repeated at least twice if not three or more times to verify that experimental findings were reproducible. |
| Randomization | For in vivo tumour treatment experiments, mice were randomized into different groups when the tumours reached between 3-5 mm in diameter. |
| Blinding | Blinding was not performed in this study. The experimental observations presented would be consistent irrespective of blinding and therefore blinding was not relevant in this study. |

# Reporting for specific materials, systems and methods

We require information from authors about some types of materials, experimental systems and methods used in many studies. Here, indicate whether each material, system or method listed is relevant to your study. If you are not sure if a list item applies to your research, read the appropriate section before selecting a response.

## Materials & experimental systems

| n/a | Involved in the study |
|-----|----------------------|
| ☐ | ☒ Antibodies |
| ☐ | ☒ Eukaryotic cell lines |
| ☒ | ☐ Palaeontology and archaeology |
| ☐ | ☒ Animals and other organisms |
| ☐ | ☒ Human research participants |
| ☒ | ☐ Clinical data |
| ☒ | ☐ Dual use research of concern |

## Methods

| n/a | Involved in the study |
|-----|----------------------|
| ☒ | ☐ ChIP-seq |
| ☐ | ☒ Flow cytometry |
| ☒ | ☐ MRI-based neuroimaging |

## Antibodies

| Antibodies used | MOUSE |
|-----------------|-------|
| | Flow cytometry (Antibody, Supplier, Clone, Colour (Catalogue #) Dilution, Lot No.) |
| | |
| | Anti-mouse CD45, Biolegend, 30-F11, APC Fire 750 (Cat #103154) 1:1600, B226658 |
| | Anti-mouse CD11c, Biolegend, N418, APC (Cat #117310) 1:200, B206713 |
| | Anti-mouse F4/80, Thermo Fisher, BM8, PE (Cat #12-4801-82) 1:300, 4299805 |
| | Anti-mouse CD11b, Biolegend, M1/70, BV711 (Cat #101242) 1:200, B379696 |
| | Anti-mouse Ly6C, Biolegend, HK1.4, PE-Cy7 (Cat #128018) 1:2000, B200606 |
| | Anti-mouse CD45R (B220), Biolegend, RA3-682, PE (Cat #103208) 1:1000, B224683 |
| | Anti-mouse CD3ε, Biolegend, 17A2, BV421 (Cat #100228) 1:500, B295089 |
| | Anti-mouse CD4, BD Biosciences, RM4-5, BV605 (Cat #563151) 1:500, 8039838 |
| | Anti-mouse NK1.1, Biolegend, PK136, APC (Cat #108710) 1:400, B191787 |
| | Anti-mouse CD45, BD Biosciences, 30-F11, FITC (Cat # 553079) 1:1000, 0030912 |
| | Anti-mouse F4/80, Biolegend, BM8, APC (Cat #123116) 1:200, B205476 |
| | Anti-mouse Ly6C, Biolegend, HK1.4, BV421 (Cat #128031) 1:800, B284703 |
| | Anti-mouse iNOS, Thermo Fisher, CXNFT, PE (Cat #12-5920-80 ) 1:300, 2283975 |
| | Anti-mouse I-A/I-E, Biolegend, M5/114.15.2, BV510 (Cat #107635) 1:800, B336985 |
| | Anti-mouse CD45, Biolegend, 30-F11, BV711 (Cat #103147) 1:200, B339309 |
| | Anti-mouse CD11c, Biolegend, N418, APC Fire 750 (Cat #117352) 1:100, B367888 |
| | Anti-mouse Siglec H, Biolegend, 551, FITC (Cat #129603) 1:400, B292161 |
| | Anti-mouse CD4, Thermofisher GK1.5, PE (Cat #12-0041-82) 1:1600, E01010-1635 |
| | Anti-mouse CD11b, Biolegend, M1/70, PE-Cy7 (Cat #101216) 1:2000, B203625 |
| | Anti-mouse Ly6G, BD Biosciences, 1A8, PE (Cat #551461) 1:800, 4246573 |
| | Anti-mouse CD3ε, Biolegend, 145-2C11, BV711 (Cat #100349) 1:100, B275433 |
| | Anti-mouse CD8α, Biolegend, 53-6.7, APC Fire 750 (Cat #100766) 1:1600, B247625 |
| | Anti-mouse H2-Kb, Biolegend, AF6-88.5, PE (Cat #116508) 1:500, B179854 |
| | Anti-mouse I-A/I-E, Biolegend, M5/114.15.2, APC (Cat #107614) 1:2000, B191785 |
| | Anti-mouse CD3ε, Biolegend, 145-2C11, FITC (Cat #100306) 1:100, B241616 |
| | Anti-mouse CD335 (NKp46), Biolegend, 29A1.4, APC (Cat #137608) 1:100, B375108 |
| | Anti-mouse CD8α, BD Biosciences, 53-6.7, PE (Cat #137608) 1:800, 5047674 |
| | Anti-mouse Vβ14 T cell receptor, BD Biosciences, 14-2, FITC (Cat # 553258) 1:2000, 6259505 |
| | Anti-mouse T-bet, Biolegend, 4B10, PeCy7, (Cat# 644824) 1:200, B214294 |
| | Anti-mouse Foxp3, Biolegend, MF-14, Alexa Fluor 647 (Cat# 126408) 1:100, B358685 |
| | Anti-mouse CD16/32, BioLegend, 93 (Cat # 101320) 1:300, B266362 |
| | |
| | Western blot (Antibody, Supplier, Clone (Catalogue #) Dilution, Lot No.) |
| | |
| | Anti-mouse β-Actin (C4), Santa Cruz Biotechnology (Cat #sc-47778 HRP) 200 µg/ml, L3112 |
| | Anti-mouse TRP1 (M-19), Santa Cruz Biotechnology, Goat polyclonal (Cat #sc-10448) 1:1000, 0593-100808W5 |
| | Anti-goat HPR, Santa Cruz Biotechnology (Cat #sc-2354) 1:2000, A0319 |
| | |
| | In vivo depletion (Antibody, Supplier, Clone (Catalogue #) Lot No.) |
| | |
| | Anti-mouse MHC-II, BioXCell, Y3P (Cat #BE0178), 796422M2 |
| | Anti-mouse NK1.1, BioXCell, PK136 (Cat #BE0036), 796521N1 |
| | Anti-mouse CD8, BioXCell, 2.43 (Cat #BE0061), 666418M1 |
| | Anti-mouse Ly6G, BioXCell, 1A8 (Cat #BE0075-1), 673218J1 |
| | Anti-mouse IFNg, BioXCell, XMG1.2 (Cat #BE0055), 791321M1 |
| | Anti-mouse CCR2, Matthias Mack, MC21 |
| | |
| | Single cell RNA sequencing hashtags (Antibody, Supplier, Clone (Catalogue #) Dilution, Lot No.) |
| | |
| | TotalSeq™-B0301 anti-mouse Hashtag 1, Biolegend, M1/42; 30-F11, (Cat # 155831), 1:300, B324862 |
| | TotalSeq™-B0302 anti-mouse Hashtag 2, Biolegend, M1/42; 30-F11, (Cat # 155833), 1:300, B329819 |
| | TotalSeq™-B0303 anti-mouse Hashtag 3, Biolegend, M1/42; 30-F11, (Cat # 155835), 1:300, B324863 |
| | TotalSeq™-B0304 anti-mouse Hashtag 4, Biolegend, M1/42; 30-F11, (Cat # 155837), 1:300, B327527 |
| | TotalSeq™-B0305 anti-mouse Hashtag 5, Biolegend, M1/42; 30-F11, (Cat # 155839), 1:300, B318761 |

TotalSeq™-B0306 anti-mouse Hashtag 6, Biolegend, M1/42; 30-F11, (Cat # 155841), 1:300, B319551
TotalSeq™-B0307 anti-mouse Hashtag 7, Biolegend, M1/42; 30-F11, (Cat # 155843), 1:300, B326966
TotalSeq™-B0308 anti-mouse Hashtag 8, Biolegend, M1/42; 30-F11, (Cat # 155845), 1:300, B318319
TotalSeq™-B0309 anti-mouse Hashtag 9, Biolegend, M1/42; 30-F11, (Cat # 155847), 1:300, B326544
TotalSeq™-B03010 anti-mouse Hashtag 10, Biolegend, M1/42; 30-F11, (Cat # 155849), 1:300, B318317

Immunofluorescence (Antibody, Supplier, Colour, (Catalogue #) Dilution, Lot No.)

Rat anti-mouse I-A/I-E, BD Bioscience, M5/114.15.2, Purified (Cat #556999) 1:50, 6104526
Donkey anti-rat IgG (H+L), Jackson ImmunoResearch, Alexa Fluor 594 (Cat #712-585-150) 1:100, 126246
_________________________________________________________________________________________

HUMAN

Immunohistochemistry (Antibody, Supplier, Clone (Catalogue #) Dilution, Lot No.)

Anti-human MHC-I (HLA-Class 1 ABC), Abcam, EMR8-5 (Cat #ab70328), 1:100, 20064861
Anti-human MHC-II (HLA-DP,DQ,DR), Abcam, CR3/43 (Cat #ab7856), 1:200, 12253498
Anti-human CD8, VENTANA, SP57 (Cat #05937248001), Undiluted, J16713
Anti-human MART-1 (MelanA), VENTANA, A103, (Cat #05278350001), Undiluted, J29957
Anti-human gp100, VENTANA, HMB45, (Cat #05479282001), Undiluted J27017
Anti-human S100, VENTANA, 4C4.9, (Cat #05278104001), Undiluted, J27878
Anti-human SOX10, Vitro Master Diagnostica, EP268 (Cat #MAD-000656QD-12), Undiluted, 06560046S

MILAN (Antibody, Supplier, Clone (Catalogue #) Dilution, Lot No.)

Anti-human CD3, Sigma Aldrich, polyclonal (Cat# C7930) 1 μg/mL, WB3189161
Anti-human panCK, Santa Cruz Biotechnology, LP5K (Cat# sc-53264) 1 μg/mL, 11246817
Anti-human CD4, Abcam, EPR6855, (Cat# ab133616) 1:200, GR3276764-17
Anti-human Foxp3, Abcam, 236A/E7, (Cat# ab20034) 1 μg/mL, GR3409148-10
Anti-human MHC-II, Novus Biologicals, SPM288 (Cat# NBP2-45312) 1 μg/mL, G0615
Anti-human CD68, Thermo Fischer Scientific, PGM1 (Cat# MA5-12407) 1:200, VB2949567
Anti-human Melan-A, Novus Biologicals, A19-P (Cat# NBP1-30151) 1:500, 41343161
Anti-human CD31, Santa Cruz Biotechnology, JC70 (Cat# sc-53411) 1 μg/mL, D1913
Anti-human CD11c, Santa Cruz, ITGAX (Cat# SC-46677), 1 μg/mL, H2416
Anti-human MITF, Dako, DS (Cat#M3621), 1 μg/mL, 10051273

**Validation**

All antibodies were obtained from commercial vendors and specificity was based on descriptions and information provided in corresponding data sheets provided by the manufacturers, and confirmed via in-house antibody titrations.

Validation statement for each antibody is provided on the manufacturer's website:

MOUSE

Flow cytometry

Anti-mouse CD45-APC Fire 750
https://www.biolegend.com/nl-be/products/apc-fire-750-anti-mouse-cd45-antibody-13049
Anti-mouse CD11c-APC
https://www.biolegend.com/de-at/products/apc-anti-mouse-cd11c-antibody-1813
Anti-mouse F4/80-PE
https://www.thermofisher.com/antibody/product/F4-80-Antibody-clone-BM8-Monoclonal/12-4801-82
Anti-mouse CD11b-BV711
https://www.biolegend.com/en-us/products/brilliant-violet-711-anti-mouse-human-cd11b-antibody-7927?GroupID=BLG10552
Anti-mouse Ly6C-PeCy7
https://www.biolegend.com/en-gb/products/pe-cyanine7-anti-mouse-ly-6c-antibody-6063
Anti-mouse CD45R, (B220) - PE
https://www.biolegend.com/de-de/products/pe-anti-mouse-human-cd45r-b220-antibody-447
Anti-mouse CD3ε -BV421
https://www.biolegend.com/de-de/products/brilliant-violet-421-anti-mouse-cd3-antibody-7326
Anti-mouse CD4-BV605
https://www.bdbiosciences.com/en-de/products/reagents/flow-cytometry-reagents/research-reagents/single-color-antibodies-ruo/bv605-rat-anti-mouse-cd4.563151
Anti-mouse NK1.1-APC
biolegend.com/en-us/products/apc-anti-mouse-nk-1-1-antibody-427?GroupID=GROUP20
Anti-mouse CD45-FITC
https://www.bdbiosciences.com/zh-cn/products/reagents/flow-cytometry-reagents/research-reagents/single-color-antibodies-ruo/fitc-rat-anti-mouse-cd45.553079
Anti-mouse F4/80-APC
https://www.biolegend.com/en-us/products/apc-anti-mouse-f4-80-antibody-4071?GroupID=BLG5319
Anti-mouse Ly6C-BV421
https://www.biolegend.com/de-de/products/brilliant-violet-421-anti-mouse-ly-6c-antibody-8586
Anti-mouse iNOS-PE
https://www.thermofisher.com/antibody/product/iNOS-Antibody-clone-CXNFT-Monoclonal/12-5920-82
Anti-mouse I-A/I-E-BV510
https://www.biolegend.com/en-us/products/brilliant-violet-510-anti-mouse-i-a-i-e-antibody-7997?GroupID=BLG11931

Anti-mouse CD45-BV711
https://www.biolegend.com/nl-nl/products/brilliant-violet-711-anti-mouse-cd45-antibody-10439
Anti-mouse CD11c-APC Fire 750
https://www.biolegend.com/en-us/products/apc-fire-750-anti-mouse-cd11c-antibody-13050?6664
Anti-mouse Siglec H- FITC
https://www.biolegend.com/nl-be/products/fitc-anti-mouse-siglec-h-antibody-5177
Anti-mouse CD4-PE
https://www.thermofisher.com/antibody/product/CD4-Antibody-clone-GK1-5-Monoclonal/12-0041-82
Anti-mouse CD11b-PE-Cy7
https://www.biolegend.com/en-us/products/pe-cyanine7-anti-mouse-human-cd11b-antibody-1921?GroupID=BLG10427
Anti-mouse Ly6G-PE
https://www.bdbiosciences.com/en-eu/products/reagents/flow-cytometry-reagents/research-reagents/single-color-antibodies-ruo/
pe-rat-anti-mouse-ly-6g.551461
Anti-mouse CD3ε,-BV711
https://www.biolegend.com/fr-fr/products/brilliant-violet-711-anti-mouse-cd3epsilon-antibody-11975
Anti-mouse CD8α-APC Fire 750
https://www.biolegend.com/de-de/products/apc-fire-750-anti-mouse-cd8a-antibody-13048
Anti-mouse H2-Kb-PE
https://www.biolegend.com/en-us/products/pe-anti-mouse-h-2kb-antibody-1749?GroupID=BLG2539
Anti-mouse I-A/I-E-APC
https://www.biolegend.com/en-us/products/apc-anti-mouse-i-a-i-e-antibody-2488
Anti-mouse CD3ε-FITC
https://www.biolegend.com/en-us/products/fitc-anti-mouse-cd3epsilon-antibody-23
Anti-mouse CD335 (NKp46) - APC
https://www.biolegend.com/de-at/products/apc-anti-mouse-cd335-nkp46-antibody-6676?GroupID=BLG8849
Anti-mouse CD8α-PE
https://www.bdbiosciences.com/en-de/products/reagents/flow-cytometry-reagents/research-reagents/single-color-antibodies-ruo/
pe-rat-anti-mouse-cd8a.553032
Anti-mouse Vβ14 T cell receptor-FITC
https://www.bdbiosciences.com/en-eu/products/reagents/flow-cytometry-reagents/research-reagents/single-color-antibodies-ruo/
fitc-rat-anti-mouse-v-14-t-cell-receptor.553258
Anti-mouse T-bet-PeCy7
https://www.biolegend.com/en-us/products/pe-cyanine7-anti-t-bet-antibody-8328?GroupID=BLG6433
Anti-mouse Foxp3-Alexa Fluor 647
https://www.biolegend.com/en-us/products/alexa-fluor-647-anti-mouse-foxp3-antibody-4662
Anti-mouse CD16/32
https://www.biolegend.com/nl-be/products/trustain-fcx-anti-mouse-cd16-32-antibody-5683

Western blot

Anti-mouse TRP1
https://datasheets.scbt.com/sc-10448.pdf
Anti-mouse β-Actin
https://datasheets.scbt.com/sc-47778.pdf
Anti-goat HPR
https://www.scbt.com/p/mouse-anti-goat-igg-hrp

In vivo depletion

Anti-mouse MHC-II
https://bioxcell.com/invivomab-anti-mouse-mhc-class-ii-i-a-be0178
Anti-mouse NK1.1
https://bioxcell.com/invivomab-anti-mouse-nk1-1-be0036
Anti-mouse CD8
https://bioxcell.com/invivomab-anti-mouse-cd8a-be0061
Anti-mouse Ly6G
https://bioxcell.com/invivomab-anti-mouse-ly6g
Anti-mouse IFNg
https://bioxcell.com/invivomab-anti-mouse-ifng-be0055
Anti-mouse CCR2
Mack et al., 2001 Journal of Immunology

Single cell RNA sequencing hashtags
TTotalSeq™-B0301 anti-mouse Hashtag 1
https://www.biolegend.com/en-us/products/totalseq-b0301-anti-mouse-hashtag-1-antibody-17771
TotalSeq™-B0302 anti-mouse Hashtag 2
https://www.biolegend.com/en-us/products/totalseq-b0302-anti-mouse-hashtag-2-antibody-17772
TotalSeq™-B0303 anti-mouse Hashtag 3
https://www.biolegend.com/en-us/products/totalseq-b0303-anti-mouse-hashtag-3-antibody-17773
TotalSeq™-B0304 anti-mouse Hashtag 4
https://www.biolegend.com/en-us/products/totalseq-b0304-anti-mouse-hashtag-4-antibody-17774
TotalSeq™-B0305 anti-mouse Hashtag 5
https://www.biolegend.com/en-us/products/totalseq-b0305-anti-mouse-hashtag-5-antibody-17775
TotalSeq™-B0306 anti-mouse Hashtag 6
https://www.biolegend.com/en-us/products/totalseq-b0306-anti-mouse-hashtag-6-antibody-17776
TotalSeq™-B0307 anti-mouse Hashtag 7

https://www.biolegend.com/en-us/products/totalseq-b0307-anti-mouse-hashtag-7-antibody-17777
TotalSeq™-B0308 anti-mouse Hashtag 8
https://www.biolegend.com/en-us/products/totalseq-b0308-anti-mouse-hashtag-8-antibody-17778
TotalSeq™-B0309 anti-mouse Hashtag 9
https://www.biolegend.com/en-us/products/totalseq-b0309-anti-mouse-hashtag-9-antibody-17779
TotalSeq™-B03010 anti-mouse Hashtag 10
https://www.biolegend.com/en-us/products/totalseq-b0310-anti-mouse-hashtag-10-antibody-18225

Immunofluorescence

Rat anti-mouse I-A/I-E-Purified
https://www.bdbiosciences.com/en-us/products/reagents/flow-cytometry-reagents/research-reagents/single-color-antibodies-ruo/purified-rat-anti-mouse-i-a-i-e.556999
Donkey anti-rat IgG (H+L)
https://www.jacksonimmuno.com/catalog/products/712-585-150
______________________________________________________________________________________

HUMAN

Immunohistochemistry

Anti-MHC-I (HLA-Class 1 ABC)
https://www.abcam.com/products/primary-antibodies/hla-class-1-abc-antibody-emr8-5-ab70328.html
Anti-MHC-II (HLA-DP,DQ,DR)
https://www.abcam.com/products/primary-antibodies/hla-dr--dp--dq-antibody-cr343-ab7856.html
Anti-human CD8
https://shop.roche-diagnostics.ch/labor/05937248001
Anti-human MART-1 (MelanA)
https://shop.roche-diagnostics.ch/labor/05278350001
Anti-human gp100
https://shop.roche-diagnostics.ch/labor/05479282001
Anti-human S100
https://shop.roche-diagnostics.ch/labor/05278104001
Anti-human SOX10
https://www.medac-diagnostika.de/index.php?controller=product&id_product=10566

MILAN

Anti-human CD3
https://www.sigmaaldrich.com/DE/en/product/sigma/c7930
Anti-human panCK
https://www.scbt.com/p/cytokeratin-7-antibody-lp5k
Anti-human CD8
https://www.thermofisher.com/antibody/product/CD8-Antibody-clone-SP16-Monoclonal/MA5-16345
Anti-human CD4
https://www.abcam.com/products/primary-antibodies/cd4-antibody-epr6855-ab133616.html
Anti-human Foxp3
https://www.abcam.com/products/primary-antibodies/foxp3-antibody-236ae7-ab20034.html
Anti-human MHC-II
https://www.novusbio.com/products/hla-drb1-antibody-spm288_nbp2-45312
Anti-human CD68
https://www.thermofisher.com/antibody/product/CD68-Antibody-clone-PG-M1-Monoclonal/MA5-12407
Anti-human MLANA
https://www.novusbio.com/products/melan-a-mart-1-antibody-a19-p_nbp1-30151
Anti-human CD31
https://www.scbt.com/p/pecam-1-antibody-jc70
Anti-human CD11c
https://www.scbt.com/p/integrin-alphax-antibody-b-6
Anti-human MITF
https://www.agilent.com/en/product/immunohistochemistry/antibodies-controls/primary-antibodies/mitf-(concentrate)-76592#productdetails

# Eukaryotic cell lines

Policy information about cell lines

| Cell line source(s) | The mouse HCmel12 cell line and all variants were generated in the Tüting Laboratory. The mouse B16 melanoma cell line was purchased from ATCC.<br><br>The human melanoma cell lines MaMel04 and MaMel102 were kindly provided by Dirk Schadendorf. The human melanoma cell lines, Skmel28 and A375, and HEK293T cells were purchased from ATCC. The 911 human embryonic retinoblast cell line was obtained from Crucell. |
|---|---|
| Authentication | B16, Skmel28, A375 and HEK293T cells were originally obtained from ATCC respectively and were therefore authenticated by |

| Authentication | the manufacturer. Furthermore, all cell lines were subjected to STR fingerprinting analysis. Successful gene knock-out for the CRISPR-variants of HCmel12 were all confirmed on a genomic (next-generation sequencing), on a transcriptomic (q-PCR)/ proteomic (western blot) and functional level. Fluorescence was always confirmed by flow cytometry. |
| Mycoplasma contamination | Cell lines regularly tested negative for mycoplasma contamination. |
| Commonly misidentified lines (See ICLAC register) | No commonly misidentified cell lines were used in this study. |

## Animals and other organisms

Policy information about studies involving animals; ARRIVE guidelines recommended for reporting animal research

| Laboratory animals | Mice were housed in an ambient temperature- and humidity-controlled environment on a 12-hour light/dark cycle to mimic natural conditions. Laboratory mouse (Mus musculus) strains C57BL/6J mice were purchased from Janvier or Charles River. Pmel-1, TRP1, OT-I and OT-II mice were purchased from Jackson Laboratories and bred in Central Animal Laboratory, House 65, University Hospital Magdeburg. Pmel-1-Venus mice were generated by crossing CAG-Venus mice with pmel-1 mice. TRP-1-eGFP mice were generated by crossing B6-eGFP mice into the TRP-1-deficient Rag1-KO background of TRP-1 mice. OT-I-Venus mice were generated by crossing CAG-Venus mice with OT-I mice. OT-II-dsRed were generated by crossing OT-II mice with hCD2-dsRed mice (kindly provided by Cornelia Harlin). Pmel-Venus, TRP1-GFP, OT-I-Venus, OT-II-dsRed and CD11c-Venus mice were bred in Central Animal Laboratory, House 65, University Hospital Magdeburg. All transgenic strains were maintained on a C57BL/6 background. All mice were aged between 8 and 12 weeks of age at the time experiments commenced. All animal experiments were conducted with male mice on the C57BL/6 background under specific pathogen-free conditions in individually ventilated cages according to the institutional and national guidelines for the care and use of laboratory animals. |
| Wild animals | No wild animals were used in the study. |
| Field-collected samples | No field collected samples were included in the study. |
| Ethics oversight | Approval by the Ethics Committee of the Office for Veterinary Affairs of the State of Saxony-Anhalt, Germany (permit license numbers 42502-2-1393 Uni MD, 42502-2-1586 Uni MD, 42502-2-1615 Uni MD and 42502-2-1672 Uni MD) in accordance with legislation of both the European Union (Council Directive 499 2010/63/EU) and the Federal Republic of Germany (according to § 8, Section 1 TierSchG, and TierSchVersV). |

Note that full information on the approval of the study protocol must also be provided in the manuscript.

## Human research participants

Policy information about studies involving human research participants

| Population characteristics | Melanoma metastases of 12 male and 8 female patients with a median age of 76 years (range 35-88 years) were biopsied in the Department of Dermatology at the Univeristy Hospital Magdeburg. Melanoma metastases of 9 male and 11 female patients with a median age of 66 years (range 34-82 years) were biopsied the Department of Oncology at the UZ Leuven. |
| Recruitment | From the University Hospital Magdeburg, samples from melanoma metastases were collected as part of a non-interventional single-centre study investigating the dynamics of the inflammatory immune cell composition.<br><br>From UZ Leuven, biopsies from melanoma metastases were collected as part of a non-interventional single-center prospective study investigating transcriptomic changes upon immune checkpoint inhibition (Prospective Serial biopsy collection before and during immune-checkpoint inhibitor therapy in patients with malignant melanoma (SPECIAL). Biopsies were taken from easily accessible sites (skin, subcutis, lymph node). |
| Ethics oversight | Participants from the University Hospital Magdeburg: Ethical approval for the observational study under the title "Dynamics of inflammatory responses during the initiation and progression of skin cancer"(Study No. 162/20).<br><br>Participants from UZ Leuven: Ethical approval from the UZ Leuven Medical Ethical Committee. |

Note that full information on the approval of the study protocol must also be provided in the manuscript.

## Flow Cytometry

### Plots

Confirm that:

☒ The axis labels state the marker and fluorochrome used (e.g. CD4-FITC).

☒ The axis scales are clearly visible. Include numbers along axes only for bottom left plot of group (a 'group' is an analysis of identical markers).

☒ All plots are contour plots with outliers or pseudocolor plots.

☒ A numerical value for number of cells or percentage (with statistics) is provided.

## Methodology

**Sample preparation**

Blood samples were resuspended in red cell lysis buffer and incubated for 15 minutes at room temperature. The samples were then centrifuged at 350g for 5 minutes and the supernatant was discarded. This process was repeated. FC-Blocking was performed by incubation of the samples with anti-CD16/32 (1:300) for 10 minutes at 4°C. After washing the samples, the cells were then stained with antibodies for 15 minutes at 4°C. The samples were subsequently washed and resuspended in FACS buffer prior to analysis.

Tumours, spleens and lymph nodes were homogenised through a 70 µm strainer to generate single cell suspensions. The samples were then centrifuged at 350g for 5 minutes and the supernatant was discarded. The samples were then resuspended in red cell lysis buffer and incubated for 5 minutes at room temperature. The samples were then centrifuged again and the supernatant was discarded prior to FC-Blocking of the samples with anti-CD16/32 (1:300) for 10 minutes at 4°C. After washing the samples, the cells were then stained with antibodies for 15 minutes at 4°C. The samples were subsequently washed and resuspended in FACS buffer prior to analysis.

**Instrument**

Attune NxT flow cytometer.

**Software**

Attune NxT for collection and Flowjo v10.8.1 (Treestar) for analysis.

**Cell population abundance**

To quantify the abundance of immune cell subpopulations in tumour tissues, 2000 cells of interest per biological sample were concatenated to a single FCS file. t-SNE plots were generated in FlowJo using the opt-SNE learning configuration (https://www.nature.com/articles/s41467-019-13055-y). The vantage-point tree KNN algorithm and the Barnes-Hut gradient algorithm set to 1000 iterations, 30 perplexity and 840 learning rate.

**Gating strategy**

Please refer to Supplementary Figure 1.
For blood, Pmel cells were identified using the following gating strategy: FSCH lo/SSCH intermediate (lymphocytes) --> FSCA/FSCWlo (singlet gate) --> +/- CD45+ (lymphocytes) --> CD8+ Venus (transgenic Pmel).
For blood, Trp1 cells were identified using the following gating strategy: FSCH lo/SSCH intermediate (lymphocytes) --> FSCA/FSCWlo (singlet gate) --> +/- CD45+ (lymphocytes) --> CD8+ eGFP+(transgenic Trp1).
For blood, OT.I cells were identified using the following gating strategy: FSCH lo/SSCH intermediate (lymphocytes) --> FSCA/FSCWlo (singlet gate) --> +/- CD45+ (lymphocytes) --> CD8+ Venus+ (transgenic OT.I).
For blood, OT.II cells were identified using the following gating strategy: FSCH lo/SSCH intermediate (lymphocytes) --> FSCA/FSCWlo (singlet gate) --> +/- CD45+ (lymphocytes) --> CD4+ dsRED+ (transgenic OT.II).
To assess cell death, melanoma cells were identified using the following gating strategy: FSCA/SCCA --> FSCA/FSCWlo (singlet gate) --> PI+ Annexin+ (dead cells) .
To quantitate MHC expression, melanoma cells were identified using the following gating strategy: FSCA/SCCA --> FSCA/FSCW lo (singlet gate) --> MHC-I hi or MHC-II hi (histogram gate).
To phenotype intratumoural CD4+ T cells in Figure 1: FSCH lo/SSCH intermediate (lymphocytes) --> FSCA/FSCW lo (singlet gate) --> +/- CD45+ (lymphocytes) --> CD3+ CD4+ --> eGFP+ (transferred) --> T-bet+ (Th1), Foxp3+ (Treg).
To quantitate immune subsets in Figure 1 and Extended Data Figure 3, leukocytes were identified using the following gating strategy: FSCA/SSCA --> FSCA/FSCWlo (singlet gate) --> CD45+ 7AAD- (live leukocytes) --> Immature monocytes (CD11b+ Ly6C hi), mature macrophages, (CD11b+ F4/80+), mature monocytes (CD11b+ Ly6C lo), TRP1 CD4 (GFP+), dendritic cells (CD11b+ MHC-II+ CD11c+), endogenous lymphocytes (CD11b- CD11c-), neutrophils (CD11b+ Ly6G+).
To quantitate immune subsets in Figure 1 and Extended Data Figure 5, leukocytes were identified using the following gating strategy: FSCA/SSCA --> FSCA/FSCWlo (singlet gate) --> CD45+ 7AAD- (live leukocytes) --> Immature monocytes (CD11b+ Ly6C hi), mature macrophages (CD11b+ F4/80+), mature monocytes (CD11b+ Ly6c lo), TRP1 CD4 (GFP+), Pmel CD8 (Venus+), dendritic cells (CD11b+ MHC-II+ CD11c+ F4/80-), endogenous lymphocytes (CD11b- CD11c-), neutrophils (CD11b+ Ly6G+).
To quantitate immune subsets in Figure 3 and Extended Data Figure 9, leukocytes were identified using the following gating strategy: FSCA/SSCA --> FSCA/FSCW lo (singlet gate) --> CD45+ 7AAD- (live leukocytes) --> Immature monocytes (CD11b+ Ly6C hi), mature macrophages (CD11b+ F4/80+ iNOS-), iNOS+ mono/macs (CD11b+ Ly6C hi iNOS+) mature monocytes (CD11b+ Ly6c lo), dendritic cells (CD11b+ MHC-II+ CD11c+ F4/80-), endogenous lymphocytes (CD11b- CD11c-), neutrophils (CD11b+ Ly6G+).
To quantitate immune subsets in Extended Data Figure 10, leukocytes were identified using the following gating strategy: FSCA/SSCA --> FSCA/FSCWlo (singlet gate) --> CD45+ 7AAD- (live leukocytes) --> Monocytes (CD11b+ Ly6C hi) and neutrophils (CD11b+ Ly6G+).

☒ Tick this box to confirm that a figure exemplifying the gating strategy is provided in the Supplementary Information.

