## [Peer Review File · Nature]

Manuscript Title: CD4+ T-cell-induced inflammatory cell death controls immune evasive tumours

Reviewer Comments & Author Rebuttals

Reviewer Reports on the Initial Version:

Referees' comments:

Referee #1 (Remarks to the Author):

This paper is a follow up of previous work, where the authors study adoptively transferred effector T cells acting on tumours in vivo, and found that TNF α may cause reversible dedifferentiation of melanoma. In this manuscript they focus on CD4 T cells and how they exert their anti-tumour function.

While it is known that CD4 T cells can directly kill target cells, also other, more indirect mechanisms have been published throughout the past decades, including the secretion of IFN γ and the production of nitric oxide by macrophages.

In this manuscript the authors provide more insight in the spatiotemporal dynamics of these phenomena and show that T cells clearly differ from CD4 T cells. While CD4 cells cluster at tumour invasive margins where they engage with myeloid cells, CD8 cells infiltrate tumour tissues. They suggest that the CD4 T cells myeloid cell interactions result in reprogramming of the tumour-microenvironment stimulating tumouricidal effector phenotypes. Previously described IFN γ secreted by CD4 T cells, and myeloid cell-derived nitric oxide, play an important role in driving apoptotic cell death, which might be of especial importance in cases where MHC deficient can escape direct killing by CD8 T cells.

The authors exploit an adenovirus encoded peptide harbouring both a CD8 and CD4 epitope to stimulate CD8 and CD4 T cells respectively. They exploit a quite complicated ACT model where prior to ACT mice are preconditioned by cyclophosphamide. Subsequently, they are vaccinated with the adenovirus and receive TCRtg T cells. In addition, they also receive several injections of adjuvants intratumourally. With this protocol the authors aim to mimic acute viral infection.

While getting more insight in the role of CD4 T cells in tumour eradication is certainly interesting, I have several reservations studying their models. Especially regarding, the MHC I KO models where NK cells are eliminated, the intratumoural adjuvant administration, how this would relate to systemic disease, and the impact of their work towards clinical translation.

1. First of all the treatment model that they have chosen is quite complicated and certainly less suited to execute in a patient setting. Questions that arise immediately is what do the individual components of this treatment lead to. This is not clear. In Figure 1 untreated mice are compared to mice that receive both ACT, vaccine and adjuvants. Do authors also have data of mice only stimulated with either ACT or innate stimulation alone? It would be interesting to see the effect of each of both arms, adaptive and innate. Perhaps they can show data on what happens with adjuvant

treatment intratumourally alone as a control. Furthermore, as adjuvant is given intratumourally, is the response systemic? What would happen if an untreated tumour is present on the other flank?

2. More CD8 T cells were found in the blood compared to CD4 T cells, but CD4 T cells were at least equally effective in eradicating the tumour. How do the authors see this, are CD4 T cells more effective, as it seems they are not present in higher numbers in the tissues (extended Fig 3). Have they studied the numbers also in the draining lymph nodes / spleen? If they transfer equal numbers of CD4 and CD8 T cells, is the many fold difference due to expansion, cell death, or differences in tissue distribution?
3. To demonstrate that CD4 T cells were also able to eradicate MHC I deficient tumours Hcme12 MHC-I-KO / JAK-1-KO / Ciit-KO were exploited. The problem is that to grow these tumours in the first place they need to deplete NK cells which otherwise prevent tumour outgrowth. While indeed the CD4 T cells are therapeutically active in these mice, the question arises how this compares to the effect of NK cells in a wild type setting once MHC I deficient tumour cells arise? How do the authors see the contribution of the CD4 compared to NK cells?
4. In Figs 1G and 1H, where direct and indirect killing are compared, how do T cells behave when exposed to a tumour lysate instead of DC as a control experiment? In other words, how dependent is this on DC?
5. Extended Fig 3F shows that indeed CD4 T cells accumulate more at the invasive margin whereas CD8 T cells seem to be everywhere. Also CD4 are at least 10fold less in number. What would happen if just many more CD4 T cells would be transferred (or 10 fold less CD8)? Suppose that the tumour margin is a barrier that can only be taken if sufficient cells are present. Could it be a numbers game?
6. In Fig 2B, how many CD8 and CD4 T cells were analysed?, equal numbers or many more CD8 T cells? Could this play a role?
7. In itself it is not so surprising that local and long term accumulation of CD4 T cells with myeloid cells requires expression of the TRP2 antigen by the tumour. Only then antigen can be processed and presented by myeloid cells. As the CD4 cells secrete IFN γ upon activation, could this also lead to activation of NK cells that may be locally present or recruited? A major difference with human ACT in patients is that mostly T cells used for ACT are expanded and activated in vitro prior to infusion. In this model activation is taking place in vivo and probably completely dependent on the intratumoural injection of adjuvant. Related to my remarks above (1), how do authors envision translation? Another related question is, in how far is the CD8 T cell response equally dependent on adjuvant administration?
8. Authors refer to viral (COVID-19 patients) infections, comparing this with the mechanisms effective in their tumour models. At the same time, loss of MHC I is often seen in melanoma in a more chronic setting of established disease. Is there evidence that also in humans patients (melanoma lesions can be very heterogeneous), in particular CD4 T cells accumulating at the invasive margin and interacting with myeloid cells?

Referee #2 (Remarks to the Author):

In a B6 model, the authors show Class II-restricted CD4 T cells can lead to the destruction of a Class II, Class I and IFN γ pathway deficient tumor by recruiting activated monocytes for NO-directed tumor cell killing. Overall this shows the importance of recruiting Class II restricted CD4 T cells for adoptive

T cell therapy, especially in the context of CD8 T cell resistance (Class I and IFN γ non-responsiveness by the tumor). This work very elegantly demonstrates a multicellular signaling network that substantially adds to the field. The use of the complex mouse model is appreciated. The authors have made a huge effort towards clear schematics and spelling out the complexity of the models but this needs some additional clarification and explanations. Comments below.

General:

1. The mouse model needs to be clarified throughout the manuscript. For example clarification of whether the transferred cells are human or mouse, clarification of the Venus DC studies.
2. The authors reference their model throughout the manuscript sending the reader to other papers. Would suggest having everything in one place for ease.
3. For relevance, the authors should better clarify why these studies are relevant for humans. For example, I think it would be helpful to show/mention/cite how many tumors are either IFN γ resistant or MHC deficient.
4. The model is very specific to a tumor that recruits monocytes. However, other models lack this immune landscape but still have CD4 T cell infiltration suggesting other mechanisms of resistance are at play. The authors should discuss/mention other models in their discussion. Also, are all human melanomas infiltrated by DCs?
5. Please clarify the selection of the 2 adjuvants used (instead of a TLR4 agonist for example). Some of this is described in Extended Figure 7 but would benefit the reader to know the rationale earlier.
6. The figure legends need more precision. For example, Ext Fig 2A- what does the pie chart represent? (I think MHC Class II expression in primary human melanomas – needs clarification); Figure 4 – please spell out the acronyms.
7. In the text, supplementary figures are mentioned before the actual figures and the numbering is not chronological. Understand this is due to a large amount of data but very confusing for the reader.

Specific:

8. Extended Data Fig 1, Extended data Fig 3: The authors mention that they adoptively transfer 2×10^6 and 0.5×10^6 CD8 and CD4 T cell respectively in their experiments. Can they clarify the rationale for this discrepancy? Could this in part explain why less CD4⁺ T cells are present in the blood and tumor?
9. Line 109 in reference to extended Fig 1A: the authors mention that the vaccine “activates” the transferred T cells. Could they clarify? I am not seeing evidence of activation. Should the term be “expand” – if so maybe show expansion dynamics.
10. Can the authors comment on the differences in tumor control and survival of HCMel12 by CD4 T cells MHC1 KO vs Stat1 KO vs Wild type (CRISPR control)? Control and survival by CD4 T cells seems less in the MHC KO – not sure if this is significant. If so, would be great to comment on what are the biological implications. Also, In extended Figure 1g, 2 mice experience tumor relapse when treated with CD4 T cells. Could the authors comment on what may be happening.
11. The authors mention they need to deplete NK cells for effective tumor growth of HCMel12 MHC and Stat1 KO. The biological implication of depleting NKs in the model and in humans should be discussed. Conversely, in humans NK depletion is not usually feasible – what would be the effect of the adjuvants on the NK cells in the TME? Should be discussed especially since in Ext. Fig 5 there are NK cells present in the untreated tumor.
12. Line 184-186: The authors mention the CD4 T cells secrete IFN γ but do not show this. Please clarify.

13. Lines 189-204 in reference to Extended data 5 and Fig 3 are confusing. It would be helpful to have a methods paragraph describing these experiments. By “bone fide” do they mean that the Venus + myeloid DCs transformed into inflammatory macrophages? Extended Fig 5b also showed the recruitment of Venus negative cells. Could they comment on what that means?
14. Extended Figure 5: This is the first time the authors describe the immune landscape of the tumor. This data – or at least the baseline of the tumor landscape- might benefit to be moved up and referenced earlier in the text so the reader has a good comprehension of the baseline immune profiling. For example, in Figure 1 the authors introduce DCs but don’t show how many DCs are actually in the baseline tumor model.
15. Extended Fig 5d: NK cells are present and decrease after adoptive CD4 T cell transfer as do the DCs. Are the NKs decreasing as a function of the treatment/ the CD4 T cells would be pushing the NKs out?
16. Clarification of the kinetics of the monocyte and DC populations after CD4 T cell infiltration. The DCs (cDCs and pDCs), decrease acutely – does this limit the activation of the adoptively transferred CD4 T cells?
17. This figure/paragraph (ext. data 5) would also benefit from an explanation of the function of the different macrophage/monocyte subsets in relation to their phenotype/gene expression profile.
18. Extended Fig. 6, Fig 3b: Please explain the gating strategy using Ly6C and Ly6G more comprehensively. Please clarify the relationship between Ly6C high and low. Are the Ly6C high NO producing? In general, explaining the selection of mouse markers and gating strategies would help readers that are not deeply familiar with the mouse models used.
19. Extended Figure 10 is very informative. the manuscript would benefit if this schema was a figure of the paper as opposed of being buried at the end of the extended data.

Referee #3 (Remarks to the Author):

The paper by Kruse et al entitled “CD4+ T cell-induced inflammatory killing controls immune evasive tumours” presents an interesting mechanism of how tumor antigen specific CD4 T cells can eradicate tumor cells indirectly independently of their MHC expression and their IFN responsiveness. Further, the authors also present in vivo imaging data illustrating the spatio-temporal arrangement of CD4 T cells and their importance in recruiting anti-inflammatory myeloid cells. Using single cell RNA seq, the authors also illustrate the ability of tumor specific CD4 T cells along with PRR stimulation to reprogram the myeloid network and stimulate anti-inflammatory responses in the tumor microenvironment. They present a good model to study bystander killing of escape variants. The intravital imaging data are interesting but not very informative in the context of this study and in answering why innate cells eliminate tumor escape variants. The mechanistic insight underlying the observation are limited except for iNOS contribution in the monocyte function. The relevance of the proposed findings in clinical samples is key and are required to show the relevance of the proposed mechanism. Overall, the paper is describing an important potential mechanism by which CD4 T cells eliminate tumor escape variants in concert with innate cells. However, as presented, this study is a bit preliminary and would require a considerable amount further in-depth explorations.

Specific comments:

1-The Trp1 TCR is a high affinity TCR with unusual cytolytic capabilities. The authors should validate some of the key findings in other antigen models. Given that they have access to the Ovalbumin/OTI/OTII system, showing the main findings are consistent with a different antigen is targeted is necessary to exclude potential model dependent observations.

2-Figure 1- While the use of CRISPR/Cas9 to edit genes involved in MHC class II machinery is interesting. The authors should add an extra data set where they simply block MHC class I or MHC class II using a blocking antibody in their experimental setting in order to orthogonally confirm their findings.

3-Supplemental Figure 1: How would the dynamics of CD4+ vs CD8+ T cell expansion change if both populations are injected alone. Would CD8+ T cell outcompete CD4+ T cells during homeostatic proliferation.

4-Figure 2- An additional group of MHC class II deficient mice would make the 2 photon microscopy data more convincing.

5-Would the therapy be effective in mice bearing a mixture of Class II+/+ or Trp1+/+ tumors with Class II-/- or Trp1-/- negative tumors. This scenario will recapitulate the heterogeneity found in patients' tumors.

6-Cyclophosphamide is known to promote homeostatic proliferation of transferred T cells, direct tumor cytotoxicity, releasing antigen and creating a "vaccination" effect, and profoundly modifying the tumor microenvironment. A control where Cyclophosphamide is not included in the therapy shall be considered. The authors should also consider the experiment in a setting where lymphopenia is present with the cyclophosphamide tumoricidal effect.

7-Extended data 5- Authors do not confirm if NT (Non-treated) here means with or without cyclophosphamide pre-conditioning. This is important to mention clearly as that would affect the way the data is interpreted.

8-Figure 3- The single cell RNA seq data for the characterization of monocytes, DCs and macrophages reveals intriguing data. However, considering authors claims towards the cytolytic activity of CD4 T cells, it would have been appropriate to sequence the ACT TRP1 CD4 T cells as well to prove the same.

9-The reviewer is unsure why neutrophils were not included in analysis of Figure 3. Neutrophils can express iNOS and kill tumors in a context-dependent manner. Including an experiment where neutrophils are depleted with antibodies will clarify this issue. Conversely, and to verify that Ly6C monocytes are responsible for by stander killing of escape clones, including experiments where monocytes are depleted with anti-Ly6C antibodies should confirm their necessary function. Additionally, Ly6C monocytes can be FACS sorted from tumors of treated mice and their killing

capabilities can be tested ex vivo by co-culture with tumor cells.

10-Figure 4- The indirect killing of MHC deficient tumors through IFN and iNOS dependent mechanism is clear. However, it is highly unlikely that this killing is specific? Perhaps mixing different tumor cell types fluorescently labeled or tagged will prove this. In addition, given that the central mechanism of monocyte cytotoxicity of tumors is iNOS dependent the experiment shown in Figure 4h shall be performed using iNOS deficient mice.

11-The authors might want to consider analyzing the CD4 T cells in a comprehensive flow cytometric analysis. Recent data suggest the existence of double positive (CD4 and CD8) T cells that, depending on their programming either gain or lose expression of the opposite marker and attain either regulatory or cytolytic function (<https://rupress.org/jem/article-abstract/219/6/e20212169/213237/Tumor-induced-double-positive-T-cells-display?redirectedFrom=fulltext>). Although the indirect mechanisms of apoptotic tumor cell death such as iNOS is well demonstrated, it is important to address this aspect of CD4 T cells as well.

12-This study will greatly benefit if biopsies from patients treated with immunotherapies show higher levels of iNOS positive monocytes. In addition, the authors claim that this is relevant for melanoma while melanoma tumors are known for Class II expression.

13-Methods: Not clear how T cells were isolated from donor mice. Was FACS or MACS used? Were whole splenocytes injected? The interpretation of experiments shall be clearer if purified T cell populations are used.

14-Minor Edits

Line 119- "Lack MHC I- deficient" Reword either to lacks MHC I or MHC I deficient.

Line 273- Therefore, (comma) not . (period)

Nature manuscript 2022-06-09463

Kruse et al “CD4+ T-cell-induced inflammatory killing controls immune evasive tumours”

General remarks:

We thank the referees for their constructive feedback that helped us to significantly improve our manuscript. We have performed a series of extensive new experiments and additional analyses to address all the referee's questions. In the completely revised current manuscript, we incorporated their suggestions and comments carefully. Furthermore, we have taken much care to present the complex topic in a way that is understandable to a broad audience. In the revised manuscript, we sought to strengthen the following key messages and conceptual advances of our work:

- We describe an adoptive cell transfer (ACT) therapy protocol that unleashes the unique ability of CD4+ effector T-cells to effectively eradicate established tumours, although they represent only a small subpopulation of tumour-infiltrating immune cells.
- These few CD4+ T-cells engage with IFN-activated mononuclear phagocytes and initiate indirect inflammatory tumour cell killing remotely.
- We show that this unique indirect mode of action, that has first been characterised in infectious disease models, operates independently and in parallel to the well-known mechanisms of direct cytolytic tumour destruction exerted by T- and NK-cells.
- Unlike CD8+ T- and NK-cells, CD4+ effector T-cells do not require direct interactions with tumour cells, as their effector functions can also be activated in proxy by antigen-presenting cells at the tumour invasive margins.
- Mechanistically, CD4+ T-cells and innate immune stimulation reprogram the tumour-associated monocyte network towards IFN-activated antigen-presenting and tumouricidal effector phenotypes.
- This results in an amplification loop in which T-cell-derived IFN γ drives the release of myeloid cell-derived nitric oxide to eradicate IFN-unresponsive, MHC-deficient tumour cells that evade direct recognition and direct destruction by CD8+ cytolytic T-cells.

Our results have immediate consequences for patient care, as they strongly support the clinical development of ACT therapy strategies that include activating stimuli for mononuclear phagocytes to unleash the full potential of CD4+ T-cell effector functions against immune-evasive tumours. This opens new avenues of research to advance cancer immunotherapies.

We would like to briefly summarise the main changes and new experimental data that address the main points (a-g) raised by the reviewers:

- a. Using MHC-deficient and IFN-unresponsive melanoma cells, we provide additional experimental evidence for the unique ability of adoptively transferred CD4+ T-cells to indirectly recognise and eradicate established immune evasive tumours that is independent from their ability to directly kill MHC-II-expressing tumour cells and to provide help for the cytolytic activity of CD8+ T- and NK-cells (revised Figure 1; revised Extended Data Figure 5).
- b. We validated our main experimental findings using ovalbumin as a second model antigen system (revised Figure 1r; revised Extended Data Figure 5h,i; revised Extended Data Fig. 6f,g).
- c. In new experiments, we adoptively transferred different numbers of T-cells and characterised the role of the individual components of our treatment protocol for the therapeutic efficacy against established tumours (revised Figure 1f-k; new Extended Data Figure 3).
- d. We performed additional experiments to dissect the dynamic alterations of the tumour immune landscape in response to CD4 and CD8 ACT treatment (revised Figures 1j,p and 3f; revised Extended Data Figures 3, 5, and 9).
- e. In several new experiments, we provide additional insights into the mechanism how CD4+ T-cells orchestrate indirect inflammatory killing of tumour cells remotely (revised Figure 2j-l; new Supplementary Video 6; revised Extended Data Figures 4i-k; 10b,d; 11c,d,e).
- f. To demonstrate the clinical relevance of our work, we show immune evasion through MHC-I downregulation in human melanoma metastases and clustering of CD4+ T-cells with MHC-II+ antigen-presenting dendritic cells at tumour invasive margins (revised Figure 1; new Extended Data Figures 1 and 2; new Extended Data Figure 8).
- g. Finally, we provide a more detailed vision for the translational relevance of our work.

For your convenience, we also provide a list of the changes in the Main and Extended Data Figures as well as the Videos at the end of our point-to-point reply.

Response to referee #1:

This paper is a follow up of previous work, where the authors study adoptively transferred effector T-cells acting on tumours in vivo, and found that TNF α may cause reversible dedifferentiation of melanoma. In this manuscript they focus on CD4 T-cells and how they exert their anti-tumour function. While it is known that CD4 T-cells can directly kill target-cells, also other, more indirect mechanisms have been published throughout the past decades, including the secretion of IFN γ and the production of nitric oxide by macrophages.

In this manuscript the authors provide more insight in the spatiotemporal dynamics of these phenomena and show that T-cells clearly differ from CD4 T-cells. While CD4 cells cluster at tumour invasive margins where they engage with myeloid cells, CD8 cells infiltrate tumour tissues. They suggest that the CD4 T-cells myeloid cell interactions result in reprogramming of the tumour-microenvironment stimulating tumouricidal effector phenotypes. Previously described IFN γ secreted by CD4 T-cells, and myeloid cell-derived nitric oxide, play an important role in driving apoptotic cell death, which might be of especial importance in cases where MHC deficient can escape direct killing by CD8 T-cells.

The authors exploit an adenovirus encoded peptide harbouring both a CD8 and CD4 epitope to stimulate CD8 and CD4 T-cells respectively. They exploit a quite complicated ACT model where prior to ACT mice are preconditioned by cyclophosphamide. Subsequently, they are vaccinated with the adenovirus and receive TCRtg T-cells. In addition, they also receive several injections of adjuvants intratumourally. With this protocol the authors aim to mimic acute viral infection.

While getting more insight in the role of CD4 T-cells in tumour eradication is certainly interesting, I have several reservations studying their models. Especially regarding, the MHC I KO models where NK-cells are eliminated, the intratumoural adjuvant administration, how this would relate to systemic disease, and the impact of their work towards clinical translation.

1. First of all the treatment model that they have chosen is quite complicated and certainly less suited to execute in a patient setting. Questions that arise immediately is what do the individual components of this treatment lead to. This is not clear. In Figure 1 untreated mice are compared to mice that receive both ACT, vaccine and adjuvants. Do authors also have data of mice only stimulated with either ACT or innate stimulation alone? It would be interesting to see the effect of each of both arms, adaptive and innate. Perhaps they can show data on what happens with adjuvant treatment intratumourally alone as a control. Furthermore, as adjuvant is given intratumourally, is the response systemic? What would happen if an untreated tumour is present on the other flank?

Response #1

We now performed several additional experiments to dissect the impact of individual components of our ACT protocol on the treatment efficacy of adoptively transferred CD4⁺ T-cells, on the immune cell composition of the tumour microenvironment, and on the phenotype of tumour infiltrating CD4⁺ T-cells. In the revised Figure 3i and the new Extended Data Figure 3e,f, we dissect the contributions of the individual components of our treatment regimen, and show that the full ACT protocol is required to eradicate established tumours. In contrast, adenoviral vaccination and adoptively transferred CD4⁺ T-cells (V+T), or cyclophosphamide pre-treatment and adjuvant innate immune stimuli (C+I) alone induce monocyte recruitment into the tumour microenvironment, but are not efficient for tumour control. Importantly, adjuvant innate immune stimuli are required for full induction of iNOS expression (revised Extended Data Figure 10a,b).

In the new Extended Data Figure 10d,e, we show that adjuvant innate immune stimulation predominantly acts locally in injected tumours and to a lesser extent systemically effecting an untreated tumour in the opposite flank. We expect that the systemic activity of polyI:C/CpG injections can be significantly increased with the development of more sophisticated approaches and delivery methods for innate immune stimulation.

Please see also our response to comment #7 below regarding our vision for clinical translation, and our response to comment #14 of referee #2 regarding the impact of cyclophosphamide and innate immune stimulation on the tumour immune landscape and the rationale to visualise the multi-parametric flow cytometry data in a two-dimensional space using the t-SNE algorithm.

Furthermore, please see also our response to comment #6 of referee #3 regarding the effect of cyclophosphamide and innate immune stimulation on the phenotype of intra-tumoural CD4+ T-cells.

2. More CD8 T-cells were found in the blood compared to CD4 T-cells, but CD4 T-cells were at least equally effective in eradicating the tumour. How do the authors see this, are CD4 T-cells more effective, as it seems they are not present in higher numbers in the tissues (extended Fig 3). Have they studied the numbers also in the draining lymph nodes / spleen? If they transfer equal numbers of CD4 and CD8 T-cells, is the many fold difference due to expansion, cell death, or differences in tissue distribution?

Response #2

In response to the referee, we have now performed additional experiments to directly compare the expansion of equal numbers of transferred CD4+ and CD8+ T-cells in the blood, spleen and draining lymph node following our ACT regimen. The data confirm the different *in vivo* expansion dynamics of CD4+ and CD8+ T-cells regardless of the number of cells used for adoptive transfer. Even low numbers of CD8+ T-cells expand to a major cell population in the CD8+ T-cell compartment within a few days after adoptive transfer, while CD4+ T-cells remain comparatively few in number (revised Extended Data Figure 3a,b). Importantly, the expansion of adoptively transferred CD8+ T-cells appears to peak at 500,000 cells. In all other new experiments, we therefore directly compared 500,000 CD4+ and CD8+ T-cells.

Our results demonstrate that the *in vivo* expansion dynamics of adoptively transferred CD4+ T-cells differs fundamentally from that of adoptively transferred CD8+ T-cells. The differential expansion of CD4+ and CD8+ T-cells is well known in the field and presumed to result largely from differential proliferative expansion. Our data also provide some insights into the differential tissue distribution of adoptively transferred CD8+ and CD4+ T-cells. However, the molecular mechanisms underlying the different behaviour of CD4+ and CD8+ T-cells is poorly understood.

Of note, the ability of a comparatively small number of CD4+ T-cells to eradicate established tumours as effective as a much larger number of CD8+ T-cells represents a key finding of our work. On a per cell basis, CD4+ T-cells indeed appear to eradicate tumours more effectively when compared to CD8+ T-cells.

Please see also our response to comment #8 of referee #2.

3. To demonstrate that CD4 T-cells were also able to eradicate MHC I deficient tumours HCmel12 MHC-I-KO / JAK-1-KO / Ccl4-KO were exploited. The problem is that to grow these tumours in the first place they need to deplete NK-cells which otherwise prevent tumour outgrowth. While indeed the CD4 T-cells are therapeutically active in these mice, the question arises how this compares to the effect of NK-cells in a wild type setting once MHC I deficient tumour cells arise? How do the authors see the contribution of the CD4 compared to NK-cells?

Response #3

We fully agree that NK-cells also contribute to the control of MHC-deficient cancers. In particular, a crosstalk between CD4+ T-cells and NK-cells has been shown to enable the elimination of MHC-I^{low} tumours upon treatment with innate immune stimulators targeting TLR7/8 (Doorduyn et al, 2017 <https://doi.org/10.1158/2326-6066.CIR-16-0334>). Furthermore, experiments in a mouse model of herpesvirus infection showed that NK-cells are a component of CD4+ T-cell-dependent protection that is important to control viruses that evade CD8+ T-cell-mediated control (Lawler and Stevenson, 2020 <https://doi.org/10.1128/JVI.01545-19>). Finally, recent clinical and experimental investigations highlighted not only a role for CD4+ T-cells, but also a role for gamma-delta T-cells in mediating checkpoint blockade immunotherapy responses of MMR-deficient carcinomas that lack MHC-I expression (Germano et al, 2021 <https://doi.org/10.1158/2159-8290.CD-20-0987>; de Vries et al, 2023 <https://doi.org/10.1038/s41586-022-05593-1>).

However, in our work we focused on the ability of adoptively transferred CD4+ T-cells to indirectly recognise and eradicate established MHC-deficient and IFN-unresponsive tumours that evade direct recognition and destruction by adoptively transferred CD8+ T-cells. To unambiguously demonstrate this independent anti-tumour effector function of CD4+ T-cells, we took advantage of the unique properties of HCmel12 melanoma cells that constitutively lack expression of both MHC-I and MHC-II molecules unless exposed to IFNs. Accordingly, CRISPR/Cas9-mediated disruption of the Jak1 gene coding for a central mediator of the IFN signalling pathway leads to IFN-unresponsive and MHC-deficient tumour cells (revised Extended Data Figure 5a).

Depletion of NK-cells in these experiments ensured homogenous tumour take and growth kinetics of transplanted MHC-deficient Hcme12 Jak1-KO melanomas in treatment groups that was similar to Hcme12 CRISPR-ctrl melanomas (revised Extended Data Figure 5c). We considered this experimental requirement as an advantage, as it allowed us to clearly establish the unique mode of action for CD4+ T-cell mediated tumour control that is independent from their other contributions to anti-tumour immunity, specifically their ability to directly kill MHC-II-expressing tumour cells and to provide help for the cytolytic activity of CD8+ T and NK-cells (revised Figure 1n,o, revised Extended Data Figure 5b,c). We have clarified this point in the revised manuscript.

Based on results reported in the literature, we would expect that the treatment of MHC-deficient tumours arising in the presence of NK-cells with our ACT protocol would be even more potent, as the crosstalk between CD4+ T-cells and NK-cells would engage an additional mode of anti-tumour immunity.

4. In Figs 1G and 1H, where direct and indirect killing are compared, how do T-cells behave when exposed to a tumour lysate instead of DC as a control experiment? In other words, how dependent is this on DC?

Response #4

In this experiment we compared how efficient CD4+ T-cells recognise antigen presented directly by MHC II+ tumour cells versus antigen presented in proxy by tumour lysate-pulsed dendritic cells. Of note, even MHC II-expressing tumour cells activated CD4+ T-cells less efficiently than tumour lysate-pulsed dendritic cells. Since CD4+ T-cells can only recognise antigen as short peptides in the binding groove of MHC-II molecules, they cannot recognize tumour lysate in the absence of an MHC-II+ antigen-presenting cell that is capable of proteolytic processing of antigenic proteins and presentation on MHC-II molecules on their surface.

5. Extended Fig 3F shows that indeed CD4 T-cells accumulate more at the invasive margin whereas CD8 T-cells seem to be everywhere. Also CD4 are at least 10fold less in number. What would happen if just many more CD4 T-cells would be transferred (or 10 fold less CD8)? Suppose that the tumour margin is a barrier that can only be taken if sufficient-cells are present. Could it be a numbers game?

Response #5

As described in our response to comment #2 above, CD4+ and CD8+ T-cells fundamentally differ in their *in vivo* expansion dynamics regardless of the cell number used for adoptive transfer. Based on these results, we repeated the confocal and intravital microscopy experiments using equal numbers of 500,000 adoptively transferred CD4+ and CD8+ T-cells and now also included MHC-deficient and IFN-unresponsive Hcme12 Jak1-KO melanomas. We observed a similar density, spatial distribution, and migratory dynamics of Pmel-1 CD8+ T-cells in Hcme12 CRISPR-ctrl melanomas as previously described for 2,000,000 CD8+ T-cells. We presented a more detailed description of these experiments in our response to comment #3 above.

Taken together, our additional experimental data provide further evidence that CD8+ and CD4+ T-cells fundamentally differ in their spatial distribution and migratory behaviour in tumour tissues and strongly argue against an impact of the number of transferred T-cells.

6. In Fig 2B, how many CD8 and CD4 T-cells were analysed? Equal numbers or many more CD8 T-cells? Could this play a role?

Response #6

Please see our responses to the comments #2 and #5 above.

*7. In itself it is not so surprising that local and long term accumulation of CD4 T-cells with myeloid cells requires expression of the TRP1 antigen by the tumour. Only then antigen can be processed and presented by myeloid cells. As the CD4 cells secrete IFN γ upon activation, could this also lead to activation of NK-cells that may be locally present or recruited? A major difference with human ACT in patients is that mostly T-cells used for ACT are expanded and activated *in vitro* prior to infusion. In this model activation is taking place *in vivo* and probably completely dependent on the intratumoural injection of adjuvant. Related to my remarks above (1), how do authors envision translation? Another related question is, in how far is the CD8 T-cell response equally dependent on adjuvant administration?*

Response #7

As described in our response to comment #3 above, we would expect that our CD4 ACT protocol also activates the anti-tumour activity of NK-cells. However, the focus of the work presented here was to investigate the capacity of CD4+ T-cells to indirectly recognise and kill MHC-deficient and IFN-unresponsive tumour cells independent from their other contributions to anti-tumour immunity, specifically from their ability to directly target and lyse MHC-II-expressing tumour cells and to provide help for the cytolytic activity of CD8+ T- and NK-cells.

We agree that current clinical protocols for ACT in cancer patients use tumour-infiltrating T-cells that are expanded *ex vivo*. Of note, these patients receive chemotherapeutic preconditioning prior to infusion of T-cells similar to the ACT protocol we used experimentally. We would like to highlight that ACT immunotherapies using CD4+ T-cells have already been implemented in a few clinical studies. For example, ACT with NY-ESO-1 specific CD4+ T-cells were able to treat metastatic melanoma (Hunder et al 2008 <https://doi.org/10.1056/NEJMoa0800251>) and ACT with CD4+ T-cells genetically engineered to express an MHC class II-restricted T-cell receptor that recognizes the cancer germline antigen MAGE-A3 demonstrated clinical efficacy (Lu et al 2017 <https://doi.org/10.1200/JCO.2017.74.5463>). Furthermore, adjuvant intra-lesional injections of immunostimulatory CpG oligonucleotides that target TLR9 have been combined with checkpoint blockade immunotherapy in early clinical trials with promising results (Ribas et al, 2021 <https://doi.org/10.1158/2159-8290.CD-21-0425>; Davar et al, 2022 <https://doi.org/10.21203/rs.3.rs-2235839/v1>).

Our results strongly support the clinical development of immunotherapies that combine ACT approaches using CD4+ T-cells together with stimuli activating innate immunity to unleash their full multifunctional therapeutic efficacy and treat immune evasive tumours. This opens new avenues of research to advance cancer immunotherapies.

Looking further into the future, we would envision ACT approaches that include CD4+ T-cells genetically engineered to express MHC-II-restricted T-cell receptors that recognise T-cell epitopes deriving from immunogenic tumour antigens, including new epitopes that arise as a consequence of tumour-specific mutations. Gene-modified CD4+ T-cells could be stimulated *in vivo* using RNA vaccines that express the respective mutated tumour antigen. Their effector functions could be boosted in tumour tissues using immunostimulatory oligonucleotides.

8. Authors refer to viral (COVID-19 patients) infections, comparing this with the mechanisms effective in their tumour models. At the same time, loss of MHC I is often seen in melanoma in a more chronic setting of established disease. Is there evidence that also in humans patients (melanoma lesions can be very heterogeneous), in particular CD4 T-cells accumulating at the invasive margin and interacting with myeloid cells?

Response #8

We now show in the revised Figure 1a-e and the new Extended Data Figures 1 and 2, that MHC-I downregulation on tumour cells is a rather frequent event in the evolution of melanoma. Importantly, MHC-I downregulation is associated with the absence of tumour-infiltrating CD8+ T-cells (revised Figure 1a, new Extended Data Figure 1b) and with a poor response to immunotherapeutic intervention (revised Figure 1e). Please see also our response to comment #3 of referee #2 regarding our new data for human melanomas.

To further support the clinical relevance of our findings, we analysed the distribution of CD4+ T-cells in relation to myeloid immune cells in a representative multiplex immunofluorescence microscopic image of a human melanoma in cooperation with the group of Dr. Bosisio at the UZ Leuven. As shown in the New Extended Data Figure 8, CD4+ effector T-cells indeed accumulate at invasive margins where they can be found to cluster with MHC-II+ antigen-presenting dendritic cells and macrophages.

Response to referee #2:

In a B6 model, the authors show Class II-restricted CD4 T-cells can lead to the destruction of a Class II, Class I and IFN γ pathway deficient tumor by recruiting activated monocytes for NO-directed tumor cell killing. Overall this shows the importance of recruiting Class II restricted CD4 T-cells for adoptive T-cell therapy, especially in the context of CD8 T-cell resistance (Class I and IFN γ non-responsiveness by the tumor). This work very elegantly demonstrates a multicellular signaling network that substantially adds to the field. The use of the complex mouse model is appreciated. The authors have made a huge effort towards clear schematics and spelling out the complexity of the models but this needs some additional clarification and explanations. Comments below.

General:

1. The mouse model needs to be clarified throughout the manuscript. For example clarification of whether the transferred cells are human or mouse, clarification of the Venus DC studies.

Response #1

In our revised manuscript we took great care to clarify our experimental approaches. Please see also our response to comment #2 below.

2. The authors reference their model throughout the manuscript sending the reader to other papers. Would suggest having everything in one place for ease.

Response #2

We completely revised the sections of our work describing the establishment of the CD4 ACT protocol together with the references of our previous work. In particular, we completely revised Main Figure 1. The first part of this Figure (1a-e) and the associated Extended Data Figures 1 and 2 show completely new data. Here we provide evidence that human melanomas evade CD8+ T-cell immunity through MHC-I downregulation early in the disease course (please see our response to comment #3 below). In the second part of this Figure (1f-k) and the associated new Extended Data Figures 3 and 4 we present the establishment of our experimental adoptive CD4+ T-cell therapy strategy including the impact of the individual components of our treatment protocol on the treatment efficiency and the phenotype of the transferred and endogenous CD4+ T-cells in tumour tissues. In the third part of the Figure (1l-r), we then show that adoptively transferred CD4+ T-cells can indirectly recognise and kill MHC-I deficient and IFN-unresponsive melanoma cells that evade recognition and destruction by adoptively transferred CD8+ T-cells in the absence of NK-cells. We hope that Figure 1 and the associated new and revised Extended Data Figures 1-5 now comprehensively illustrate the clinical relevance of our work and facilitates the understanding of our experimental approach.

3. For relevance, the authors should better clarify why these studies are relevant for humans. For example, I think it would be helpful to show/mention/cite how many tumors are either IFN γ resistant or MHC deficient.

Response #3

In support of the clinical relevance of MHC-I loss as a mechanism of immune evasion, we reassessed the expression of MHC-I and MHC-II by immunohistochemistry in skin metastases of 20 melanoma patients that have been diagnosed and treated in the Department of Dermatology at the University Hospital in Magdeburg. Our analyses show that MHC-I downregulation on tumour cells is an a rather frequent event in the evolution of melanoma. Importantly, MHC-I downregulation is associated with the absence of tumour-infiltrating CD8+ T-cells (revised Figure 1a and new Extended Data Figure 1).

Next, we validated our findings in scRNAseq data obtained from skin and lymph node metastases in a different cohort of 20 melanoma patients that were diagnosed and treated in the Department of Oncology at the University Hospital in Leuven, Belgium. These analyses were performed in cooperation with the group of Jean-Christoph Marine at the Center for Cancer Biology of the VIB in Leuven (revised Figure 1b-e and new Extended Data Figure 2). Importantly, we observed an association of MHC-I downregulation with a poor response to checkpoint blockade immunotherapy (revised Figure 1e), indicating an immune-evasive phenotype. We also found that the expression of MHC-II is mostly restricted to antigen-presenting immune cells and only rarely found on tumour cells.

4. The model is very specific to a tumor that recruits monocytes. However, other models lack this immune landscape but still have CD4 T-cell infiltration suggesting other mechanisms of resistance are at play. The authors should discuss/mention other models in their discussion. Also, are all human melanomas infiltrated by DCs?

Response #4

Recent experimental data demonstrate that genomic instability, a hallmark of progressively growing cancers, activate innate and adaptive cellular tumour immunity. This initiates the recruitment of dendritic cells and monocytes from precursors in the blood. Monocytes further differentiate into tumour-associated macrophages. However, individual tumours differ widely in their capacity to attract dendritic cells, activate immune recognition, and evade immune destruction. This results in the different patterns of immune cell infiltration that are clinically observed (see Extended Data Figure 1).

We also present new data showing the distribution of CD4+ T-cells in relation to dendritic cells, monocytes and macrophages in a human melanoma. A representative multiplex immunofluorescence microscopic image of a human melanoma that was generated in cooperation with the group of Dr. Bosisio at the UZ Leuven demonstrates that CD4+ effector T-cells and CD4+ regulatory T-cells accumulate at invasive margins where they can be found to cluster with MHC-II+ antigen-presenting dendritic cells, monocytes, and macrophages (new Extended Data Figure 8). Our experimental work establishes a therapeutic strategy that can eliminate regulatory CD4+ T-cells and reprogram the tumour-associated myeloid network towards tumouricidal effector phenotypes.

5. Please clarify the selection of the 2 adjuvants used (instead of a TLR4 agonist for example). Some of this is described in Extended Figure 7 but would benefit the reader to know the rationale earlier.

Response #5

In our work we used the immunostimulatory oligonucleotides polyI:C and CpG that activate TLR3 and TLR9, respectively. This imitates an acute viral infection and activates the innate arm of pathogen defence. Similar oligonucleotides are currently explored in early clinical trials. Promising signs of therapeutic efficacy have recently been reported for adjuvant intralesional injections of a CpG oligonucleotide that is equivalent to the one used in our experiments in combination with checkpoint blockade immunotherapy in early clinical trials (Ribas et al, 2021 <https://doi.org/10.1158/2159-8290.CD-21-0425>; Davar et al, 2022 <https://doi.org/10.21203/rs.3.rs-2235839/v1>). We now incorporated this information in the section describing the establishment of our ACT protocol and the final paragraph that describe our vision for clinical translation.

6. The figure legends need more precision. For example, Ext Fig 2A- what does the pie chart represent? (I think MHC Class II expression in primary human melanomas – needs clarification); Figure 4 – please spell out the acronyms.

Response #6

As described in our response to comment #3 above, we greatly expanded our analyses of human melanomas and now focus on the expression of MHC molecules in early metastases. The results are presented in the revised Figure 1a-e and the new Extended Data Figures 1 and 2.

7. In the text, supplementary figures are mentioned before the actual figures and the numbering is not chronological. Understand this is due to a large amount of data but very confusing for the reader. Specific:

Response #7

In the revised version of our manuscript, we took great care to present the main and supplementary figures in chronological order to avoid confusion of the reader.

8. Extended Data Fig 1, Extended data Fig 3: The authors mention that they adoptively transfer 2×10^6 and 0.5×10^6 CD8 and CD4 T-cell respectively in their experiments. Can they clarify the rationale for this discrepancy? Could this in part explain why less CD4+ T-cells are present in the blood and tumor?

Response #8

In our previous work we routinely used 2,000,000 Pmel-1 CD8+ TCRtg T-cells for adoptive transfer experiments. We now performed additional experiments to directly compare the expansion of equal numbers of transferred CD4+ and CD8+ T-cells in the blood, spleen and draining lymph node following our ACT regimen. Our results confirm that the *in vivo* expansion dynamics of adoptively transferred CD4+ T-cells differs fundamentally from that of adoptively transferred CD8+ T-cells (revised Extended Data Figure 3a,b). Of note, the expansion of adoptively transferred CD8+ T-cells appears to peak at 500,000 cells. In all other new experiments, we therefore directly compared 500,000 CD4+ and CD8+ T-cells. Please see also our response to comment #2 of referee #1 for more details.

9. Line 109 in reference to extended Fig 1A: the authors mention that the vaccine “activates” the transferred T-cells. Could they clarify? I am not seeing evidence of activation. Should the term be “expand” – if so maybe show expansion dynamics.

Response #9

We agree with the reviewer and now use the term expansion dynamics.

10. Can the authors comment on the differences in tumor control and survival of HcMel12 by CD4 T-cells MHC1 KO vs Jak1 KO vs Wild type (CRISPR control)? Control and survival by CD4 T-cells seems less in the MHC KO – not sure if this is significant. If so, would be great to comment on what are the biological implications. Also, In extended Figure 1g, 2 mice experience tumor relapse when treated with CD4 T-cells. Could the authors comment on what may be happening.

Response #10

In the revised version of our manuscript, we focused on experiments with HcMel12 Jak1-KO as a model of an immune evasive tumour. We took advantage of the unique properties of HcMel12 melanoma cells that constitutively lack expression of both MHC-I and MHC-II molecules unless exposed to IFNs. The few MHC-deficient HcMel12 tumours that recur late after successful TRP-1 CD4 ACT immunotherapy showed an amelanotic, dedifferentiated phenotype, similar to HcMel12 CRISPR-ctrl tumours that escaped immunotherapeutic intervention. We, and others, previously described this mechanism of immunotherapy resistance through inflammatory dedifferentiation that is driven by the cytokines TNF and IFN γ (Landsberg et al, 2012 <https://doi.org/10.1038/nature11538>, Effern et al. 2020 <https://doi.org/10.1016/j.immuni.2020.07.007>; Mehta et al, 2018 <https://doi.org/10.1158/2159-8290.CD-17-1178>, Kim et al. 2021 <https://doi.org/10.1172/JCI145859>). In contrast, MHC-deficient HcMel12 Jak1-KO tumours that resist Pmel-1 CD8 ACT immunotherapy, as well as HcMel12 Trp1-KO tumours that escaped TRP-1 CD4 ACT immunotherapy, showed a heavily pigmented phenotype and high expression of melanocytic differentiation markers, indicating that adoptively transferred T-cells failed to recognise their antigen.

11. The authors mention they need to deplete NK-cells for effective tumor growth of HcMeL12 MHC and Stat1 KO. The biological implication of depleting NKs in the model and in humans should be discussed. Conversely, in humans NK depletion is not usually feasible – what would be the effect of the adjuvants on the NK-cells in the TME? Should be discussed especially since in Ext. Fig 5 there are NK-cells present in the untreated tumor.

Response #11

We agree that depletion of NK-cells in human patients is not feasible, but we would not expect that this is required. Indeed, our CD4 ACT protocol is fully active in the presence of NK-cells. The depletion of NK-cells in our experiments with MHC-deficient HcMel12 Jak1-KO melanoma cells ensured homogenous tumour take and growth kinetics in treatment groups that was similar to HcMel12 CRISPR-ctrl melanoma cells (revised Extended Data Figure 5c). We considered this experimental requirement as an advantage, as it allowed us to investigate the capacity of CD4+ T-cells to eradicate MHC-deficient and IFN-unresponsive tumour cells independent from their other contributions to anti-tumoural immunity, specifically their ability to directly kill MHC-II-expressing tumour cells and to provide help for the cytolytic activity of CD8+ T- and NK-cells.

In patients, MHC-deficient cancer cell subpopulations arise in the presence of NK-cells. Injections with innate immune stimulators that imitate a viral infection will activate NK-cells. We would expect that treatment of MHC-deficient tumours that arise in the presence of NK-cells with our ACT protocol additionally activates a crosstalk between CD4+ T-cells and NK-cells that supports antigen-specific anti-

tumour immunity. This has already been shown in experimental models (Doorduyn et al, 2017 <https://doi.org/10.1158/2326-6066.CIR-16-0334>). Therefore, we would not consider NK-cell depletion in patients.

Please see also our response to comment #3 of referee #1 for an in-depth discussion of this issue.

12. Line 184-186: The authors mention the CD4 T-cells secrete IFNgamma but do not show this. Please clarify.

Response #12

In the experiments presented in the new Extended Data Figure 4d, we first isolated TRP-1 CD4+ T-cells from mice that were treated with CD4+ ACT therapy. Following *in vitro* restimulation with tumour-lysate pulsed DC, 40% of TRP-1 CD4+ T-cells expressed intracellular IFN γ , consistent with their Th1-directed differentiation (see revised Figure 1k and revised Extended Data Figure 3i).

13. Lines 189-204 in reference to Extended data 5 and Fig 3 are confusing. It would be helpful to have a methods paragraph describing these experiments. By "bone fide" do they mean that the Venus + myeloid DCs transformed into inflammatory macrophages? Extended Fig 5b also showed the recruitment of Venus negative cells. Could they comment on what that means?

Response #13

The cellular dynamics of mononuclear phagocytes in inflammatory tissues of CD11c-Venus transgenic mice is complicated to follow, as the expression of CD11c is not restricted to dendritic cells and can also be detected on inflammatory monocytes that are recruited from the bone marrow and differentiate into macrophage effectors in mice that received CD4 ACT immunotherapy.

In order to clarify this in the revised version of our manuscript, we decided to rearrange the results of our transcriptomic and flow cytometric investigations that we presented in the former Figure 3. As suggested by the referee, we now already describe the alterations of the tumour immune landscapes following ACT treatment in Figure 1. Following up on these observations, we now first present our scRNAseq data of mononuclear phagocytes in the revised Figure 3 and the associated revised Extended Data Figure 9. Subsequently, we now present our flow cytometry data using t-SNE algorithms as described in our response to comment #14 below. The results support our analyses of the transcriptional dynamics in recruited monocytes at the protein level.

14. Extended Figure 5: This is the first time the authors describe the immune landscape of the tumor. This data – or at least the baseline of the tumor landscape- might benefit to be moved up and referenced earlier in the text so the reader has a good comprehension of the baseline immune profiling. For example, in Figure 1 the authors introduce DCs but don't show how many DCs are actually in the baseline tumor model.

Response #14

We thank the referee for this very helpful suggestion. For the revision we performed several additional experiments to analyse the impact of CD4 and CD8 ACT immunotherapies on the tumour immune landscape using flow cytometry. To provide a comprehensible holistic view of the immune cell composition in tumour tissues, we now chose to visualise the multi-parametric flow cytometry data in a two-dimensional space using the t-SNE algorithm. As suggested, we incorporated the results in the revised Figure 1 and the associated revised Extended Data Figures 3 and 5.

Using this approach, we found that adoptively transferred TRP-1 CD4+ T-cells represented only 1% of tumour-infiltrating immune cells in the microenvironment of B16 or HcMel12 melanomas after treatment with CD4 ACT (revised Extended Data Figures 3h and 5e). ACT immunotherapy and injections of innate immune stimuli independently increased the immune infiltrate in B16 melanoma cells, predominantly through the recruitment of monocytes. The full ACT protocol further increased the number of monocytes and diminished the number of dendritic cells (revised Figure 1j, revised Extended Data Figure 3h).

Visualisation of the immune landscapes as t-SNE plots in HcMel12-CRISPR-ctrl tumours on day 2, 5, and 8 following CD4 ACT immunotherapy confirms the dynamic recruitment of inflammatory monocytes and their differentiation towards iNOS+ effector phenotypes (revised Extended Data Figure 9h).

Please see our responses to comment #1 of referee #1 and to comments #6 and #11 of referee #3 for more information on the impact of the individual components of our ACT protocol on the treatment efficacy of CD4 ACT therapy and on the phenotype of tumour infiltrating TRP-1 CD4+ T-cells.

15. *Extended Fig 5d: NK-cells are present and decrease after adoptive CD4 T-cell transfer as do the DCs. Are the NKs decreasing as a function of the treatment/ the CD4 T-cells would be pushing the NKs out?*

Response #15

We did not follow up on the dynamics of NK-cells, as we specifically focused on the indirect effector functions of CD4+ T-cells that are exerted remotely and are independent of direct anti-tumour effects mediated by cytolytic T- and NK-cells.

16. *Clarification of the kinetics of the monocyte and DC populations after CD4 T-cell infiltration. The DCs (cDCs and pDCs), decrease acutely – does this limit the activation of the adoptively transferred CD4 T-cells?*

Response #16

The reduction of MHC-II+ dendritic cell populations in tumour tissues becomes prominent on day 8 after ACT therapy (new Extended Data Figure 9h). This might indeed serve to limit the activation of adoptively transferred CD4 T-cells and prevent excessive damage. However, we feel that exploring the possible shutdown of the transferred CD4+ T-cells due to deprivation of dendritic cells would be beyond the scope of our present study, which focuses on the delivery of CD4+ T-cell effector function and its mode of action for tumour control.

17. *This figure/paragraph (ext. data 5) would also benefit from an explanation of the function of the different macrophage/monocyte subsets in relation to their phenotype/gene expression profile.*

Response #17

We now annotated the different macrophage/monocyte subsets in relation to their phenotype in the text. Monocyte-derived dendritic cells (cluster ACT1) have an antigen-presentation phenotype, monocyte-macrophage effectors (cluster ACT2a-c) have a tumouricidal phenotype, and Ly6c-lo mature monocytes (cluster ACT3a,b) have a patrolling phenotype.

18. *Extended Fig. 6, Fig 3b: Please explain the gating strategy using Ly6C and Ly6G more comprehensively. Please clarify the relationship between Ly6C high and low. Are the Ly6C high NO producing? In general, explaining the selection of mouse markers and gating strategies would help readers that are not deeply familiar with the mouse models used.*

Response #18

We hope that the presentation of the multi-parametric flow cytometry data in a two-dimensional space using the t-SNE algorithm helps to comprehend the baseline immune cell composition of the different tumour models and the dynamic alterations in response to our ACT immunotherapy approaches. We selected the mouse markers and gating strategies to characterise the main immune cell types present in the tumour microenvironment. Ly6c-hi are the equivalent of CD14+CD16+ classical monocytes in humans. A detailed list of antibodies and gating strategies is found in the Nature reporting summary.

19. *Extended Figure 10 is very informative. The manuscript would benefit if this schema was a figure of the paper as opposed of being buried at the end of the extended data.*

Response #19

We thank the referee for this suggestion and included a schematic that illustrates the indirect inflammatory killing process of adoptively transferred CD4+ T-cells (revised Figure 4g).

Response to referee #3:

The paper by Kruse et al entitled "CD4+ T-cell-induced inflammatory killing controls immune evasive tumours" presents an interesting mechanism of how tumor antigen specific CD4 T-cells can eradicate tumor cells indirectly independently of their MHC expression and their IFN responsiveness. Further, the authors also present in vivo imaging data illustrating the spatio-temporal arrangement of CD4 T-cells and their importance in recruiting anti-inflammatory myeloid cells. Using single cell RNA seq, the authors also illustrate the ability of tumor specific CD4 T-cells along with PRR stimulation to reprogram the myeloid network and stimulate anti-inflammatory responses in the tumor microenvironment. They present a good model to study bystander killing of escape variants. The intravital imaging data are interesting but not very informative in the context of this study and in answering why innate cells eliminate tumor escape variants. The mechanistic insight underlying the observation are limited except for iNOS contribution in the monocyte function. The relevance of the proposed findings in clinical samples is key and are required to show the relevance of the proposed mechanism. Overall, the paper is describing an important potential mechanism by which CD4 T-cells eliminate tumor escape variants in concert with innate cells. However, as presented, this study is a bit preliminary and would require a considerable amount further in-depth explorations.

Specific Comments:

1-The Trp1 TCR is a high affinity TCR with unusual cytolytic capabilities. The authors should validate some of the key findings in other antigen models. Given that they have access to the Ovalbumin/OTI/OTII system, showing the main findings are consistent with a different antigen is targeted is necessary to exclude potential model dependent observations.

Response #1

We thank the referee for this helpful suggestion. We have performed additional experiments using ovalbumin as a second model tumour antigen and included the data in the revised manuscript. Specifically, we show that a comparatively small subpopulation of adoptively transferred ovalbumin-specific OT-II TCRtg CD4+ T-cells was also able to indirectly recognise and eradicate ovalbumin-expressing MHC-deficient HcMel12 Jak1-KO tumours that evade direct recognition and destruction by a much larger subpopulation of adoptively transferred ovalbumin-specific OT-I TCRtg CD8+ T-cells (revised Figure 1r, revised Extended Data Figure 5h,i), consistent with our results using TRP-1 TCRtg CD4+ T-cells and Pmel-1 TCRtg CD8+ T-cells (revised Figure 1o, revised Extended Data Figure 5b,c). We also present results of additional confocal and intravital fluorescence microscopy experiments with amelanotic (Tyr-KO) HcMel12 Jak1-KO tumours that express tagBFP-Ova and were treated with adoptively transferred ovalbumin-specific dsRed+ OT-II and Venus+ OT-I TCRtg T-cells (revised Extended Data Figure 6e-g). These experiments confirm the fundamental difference in the spatial distribution and the migratory behaviour of CD4+ and CD8+ T-cells in MHC-deficient tumours that we observed in experiments with eGFP+ TRP-1 TCRtg and Venus+ Pmel-1 TCRtg T-cells (presented in the revised Figure 2b-f and in the Supplementary Videos 1-4).

Of note, the spatial distribution of CD8+ T-cells in MHC-deficient HcMel12 Jak1-KO tumours imitated the pattern of CD8+ T-cells observed in immune-excluded human melanomas with downregulated MHC expression (compare Extended Data Figure 6e-g and Extended Data Figure 1b).

Together, our experimental results support our main conclusion that very few adoptively transferred CD4+ T-cells infiltrate the invasive margin of MHC-competent and -deficient tumours, where they preferentially interact with MHC-II+ antigen-presenting cells and initiate an inflammatory tumour cell killing process. In contrast, large numbers of adoptively transferred CD8+ T-cells can be found in the tumour parenchyma of MHC-competent, but not of MHC-deficient tumours, since they preferentially interact with MHC-I+ malignant-cells. This explains their differential efficacy against MHC-deficient tumours.

2-Figure 1- While the use of CRISPR/Cas9 to edit genes involved in MHC class II machinery is interesting. The authors should add an extra data set where they simply block MHC class I or MHC class II using a blocking antibody in their experimental setting in order to orthogonally confirm their findings.

Response #2

We thank the referee for this helpful suggestion. To address the role of MHC-II-restricted antigen recognition, we performed additional experiments using a mAb that blocks MHC-II-restricted antigen recognition and investigated the impact on the intra-tumoural migratory behaviour of adoptively

transferred CD4+ T-cells. Our results demonstrate that MHC-II blockade abrogates the interaction between CD4+ T-cells and Venus+ myeloid cells, confirming the specificity of our findings (revised Figure 2j-l; new Supplementary Video 6).

3-Supplemental Figure 1: How would the dynamics of CD4+ vs CD8+ T-cell expansion change if both populations are injected alone? Would CD8+ T-cell outcompete CD4+ T-cells during homeostatic proliferation?

Response #3

We carefully considered this aspect in the different experiments that we performed for the revision. We could not find significant differences of the *in vivo* expansion dynamics when CD4+ and CD8+ T-cells were transferred together (e.g. in the additional experiments for intravital microscopy shown in the revised Figure 2a-f) or separately (e.g. in the additional experiments for the flow cytometric analyses of the tumour-immune cell composition shown in the revised Figure 1j and the revised Extended Data Figure 5c,d).

Regarding the T-cell expansion please see also our response to comment #2 of referee #1.

4-Figure 2- An additional group of MHC class II deficient mice would make the 2 photon microscopy data more convincing.

Response #4

As the interaction of CD4+ T-cells with antigen presenting cells in the tumour microenvironment requires CD11c-Venus transgenic mice, this experiment would require crossing the MHC-II deficient alleles homozygously into CD11c-Venus mice. As an alternative to address the role of MHC-II-restricted antigen recognition, we instead used a mAb that blocks MHC-II recognition (see our response to comment #2 above).

5-Would the therapy be effective in mice bearing a mixture of Class II+/+ or Trp1+/+ tumors with Class II-/- or Trp1-/- negative tumors. This scenario will recapitulate the heterogeneity found in patients' tumors.

Response #5

We thank the referee for this helpful suggestion. To replicate the heterogeneity of tumours found in patients, we performed new experiments in which a mixture of HCmel12 CRISPR-Ctrl (~75%) and HCmel12 Trp1-KO (~25%) cells were implanted into wild-type mice. Treatment of tumours consisting of HCmel12 CRISPR-Ctrl and HCmel12 Trp1-KO mixtures demonstrated that TRP-1 CD4+ T-cells also exerted significant bystander killing but could not prevent the outgrowth of HCmel12 Trp1-KO cells in all mice (revised Extended Data Figure 4i-k).

6-Cyclophosphamide is known to promote homeostatic proliferation of transferred T-cells, direct tumor cytotoxicity, releasing antigen and creating a "vaccination" effect, and profoundly modifying the tumor microenvironment. A control where Cyclophosphamide is not included in the therapy shall be considered. The authors should also consider the experiment in a setting where lymphopenia is present with the cyclophosphamide tumoricidal effect.

Response #6

Chemotherapeutic preconditioning is a part of the clinical protocols currently used to treat melanoma patients with an adoptive transfer of *ex vivo* expanded tumour-infiltrating lymphocytes. For this reason we have established a cyclophosphamide pre-treatment in our experimental ACT protocol. We now show in additional experiments that all components of our ACT protocol are also required for the eradication of established tumours by adoptively transferred TRP-1 CD4+ T-cells (revised Figure 1i; revised Extended Data Figure 3e, f).

Please see also our response to comment #11 below regarding the effect of cyclophosphamide and innate immune stimulation on the phenotype of intra-tumoural CD4+ T-cells.

7-Extended data 5- Authors do not confirm if NT (Non-treated) here means with or without cyclophosphamide pre-conditioning. This is important to mention clearly as that would affect the way the data is interpreted.

Response #7

The groups designated NT (~ Non-Treated) are untreated controls. We apologise for omitting this explanation in the respective Figure legends.

8-Figure 3- The single cell RNA seq data for the characterization of monocytes, DCs and macrophages reveals intriguing data. However, considering authors claims towards the cytolytic activity of CD4 T-cells, it would have been appropriate to sequence the ACT TRP1 CD4 T-cells as well to prove the same.

Response #8

Unfortunately, an attempt to study the phenotype of CD4+ T-cells in a scRNAseq experiment failed, likely because only very few adoptively transferred CD4+ T-cells could be sorted from treated tumours. In alternative experiments, we studied the phenotype of tumour-infiltrating CD4+ T-cells by flow cytometry. Please see our response below to comment #11 for the results.

We want to emphasise that we describe the ability of CD4 T-cells to orchestrate inflammatory killing of MHC-II-deficient tumours without the need for cytolytic CD8+ T-cells or NK-cells. This mechanism does not require CD4+ T-cells to exert direct cytolytic activity and is active against MHC-II-deficient tumours.

9-The reviewer is unsure why neutrophils were not included in analysis of Figure 3. Neutrophils can express iNOS and kill tumors in a context-dependent matter. Including an experiment where neutrophils are depleted with antibodies will clarify this issue. Conversely, and to verify that Ly6C monocytes are responsible for by stander killing of escape clones, including experiments where monocytes are depleted with anti-Ly6C antibodies should confirm their necessary function. Additionally, Ly6C monocytes can be FACS sorted from tumors of treated mice and their killing capabilities can be tested ex vivo by co-culture with tumor cells.

Response #9

We thank the referee for this helpful suggestion. To further assess the contribution of inflammatory monocytes and neutrophils to CD4+ T-cell anti-tumour immunity we used anti-CCR2 (Mack et al., 2001, Journal of Immunology) and/or anti-Ly6G mAb that have been shown to deplete monocytes and neutrophils, respectively. Although antibody-mediated depletion of CCR2+ monocytes could only be achieved transiently, this impaired the efficacy of CD4 ACT therapy to a greater extent than the longer lasting antibody-mediated depletion of Ly6G+ neutrophils (revised Extended Data Figure 11c,d,e). This supports a predominant role of iNOS-expressing mononuclear phagocytes over neutrophils for inflammatory killing of MHC-deficient tumour cells.

10-Figure 4- The indirect killing of MHC deficient tumors through IFN and iNOS dependent mechanism is clear. However, it is highly unlikely that this killing specific? Perhaps mixing different tumor cell types fluorescently labeled or tagged will prove this. In addition, given that the central mechanism of monocyte cytotoxicity of tumors is iNOS dependent the experiment shown in Figure 4h shall be preform using iNOS deficient mice.

Response #10

We agree that the inflammatory killing mechanism is per se non-specific. However, the induction of this mechanism depends on antigen-specific activation of CD4+ T-cells by tumour-associated MHCII+ immune cells. As suggested, we now performed a mixing experiment using TRP1-deficient HcMel12 melanoma cells. We found that TRP-1 CD4 ACT treatment of tumours consisting of HcMel12 CRISPR-Ctrl and HcMel12 Trp1-KO mixtures exerted significant bystander killing, but could not prevent the outgrowth of HcMel12 Trp1-KO cells in all mice (revised Extended Data Fig. 4i-k). Unfortunately, we do not currently have iNOS-deficient mice available for experiments. However, we believe that the results of the new experiments with antibody-mediated depletion of CCR2+ monocytes described in our response to comment #9 above confirm our results obtained with the chemical iNOS inhibitor.

11-The authors might want to consider analyzing the CD4 T-cells in a comprehensive flow cytometric analysis. Recent data suggest the existence of double positive (CD4 and CD8) T-cells that, depending

on their programming either gain or lose expression of the opposite marker and attains either regulatory or cytolytic function (<https://rupress.org/jem/article-abstract/219/6/e20212169/213237/Tumor-induced-double-positive-T-cells-display?redirectedFrom=fulltext>). Although the indirect mechanisms of apoptotic tumor cell death such iNOS is well demonstrated, it is important to address this aspect of CD4 T-cells as well.

Response #11

In our previous work, we found that the combination of cyclophosphamide and innate immune stimulation boosted the effector functions of adoptively transferred Pmel-1 CD8+ T-cells (Kohlmeyer et al. 2009 <https://doi.org/10.1158/0008-5472.CAN-09-0579>). We now show that cyclophosphamide and innate immune stimulation strongly promoted the differentiation of both transferred and endogenous CD4+ T-cells towards a Th1-directed phenotype in tumour tissues. Furthermore, the combination prevented the accumulation of regulatory T-cells (revised Figure 1k; revised Extended Data Figure 3i). In our experiments, we did not observe CD4+ and CD8+ double positive T-cells.

12-This study will greatly benefit if biopsies patients treated with immunotherapies show higher levels of iNOS positive monocytes. In addition the authors claim that this is relevant for melanoma while melanoma tumors are known for Class II expression.

Response #12

Regarding iNOS expressing monocytes in human melanomas: In our work we demonstrate the *in vivo* dynamics of inflammatory monocytes following CD4 and CD8 ACT in mice. Of note, the recruitment of inflammatory monocytes is a rather rapid response driven by T-cells secreting IFN γ , typically observed in acute viral infections (see for example Simpson et al. 2022 <https://doi.org/10.1016/j.immuni.2022.01.003>). We expect a similar *in vivo* dynamics in melanoma patients receiving ACT. Our work provides the scientific rationale to investigate the dynamics of myeloid immune cells under therapy in patients. Unfortunately, we do not (yet) have access to on treatment biopsies of melanomas in patients that received ACT. Such studies would be a valuable consequence of our work.

In support of the clinical relevance of our experimental results, we now present a representative multiplex immunofluorescence microscopic image of a human melanoma that demonstrates the accumulation of CD4+ effector T-cells and CD4+ regulatory T-cells at invasive margins in association with MHC-II+ antigen-presenting dendritic cells, monocytes, and macrophages (new Extended Data Figure 8). Please see also our response to comment #4 of referee #2.

Regarding the expression of MHC-II on melanoma cells: Early reports more than 30 years ago showed that some melanoma cells unexpectedly express MHC-II molecules (comprehensively reviewed by Ferrone and Campoli 2006, <https://doi.org/10.1016/j.immuni.2022.01.003>). However, our analyses in two different cohorts of patients demonstrate that MHC-II expression by melanoma cells is rather the exception than the rule (new Extended Data Figure 1c and 2b), consistent with a recent report in the literature (Oliveira et al 2022, <https://doi.org/10.1038/s41586-022-04682-5>).

13-Methods: Not clear how T-cell were isolated from donor mice. Was FACS or MACS used? Were whole splenocytes injected? The interpretation of experiments shall be clearer if purified T-cell populations are used.

Response #13

We agree that the interpretation of experiments using purified T-cell populations would be more clear. However, we found that the labour intensive and expensive isolation procedures do not yield different results when compared to the use of whole splenocytes. Of note, the TRP-1 TCRtg mice are Rag-1 KO mice and only contain TRP-1 TCRtg CD4+ T-cells.

14-Minor Edits

Line 119- "Lack MHC I- deficient" Reword either to lacks MHC I or MHC I deficient.

Line 273- Therefore, (comma) not . (period)

Response #14

In the now completely revised current manuscript we tried to avoid spelling errors as best as possible.

List of revised/new Main Figures and Extended Data Figures with explanation of changes:

General remarks. All Figures were revised according to the suggestions of the reviewers. Figures 1 and 2 were completely revised and contain new results obtained in a number of additional *in vivo* experiments in mice and in additional analyses of human melanoma metastases obtained from two independent cohorts of melanoma patients. Figures 3 and 4 were substantially rearranged.

Revised Figure 1. A small population of CD4⁺ effector T-cells can indirectly recognise and eradicate MHC-deficient and IFN-unresponsive melanomas that resist destruction by CD8⁺ cytotoxic T-cells.

Figure 1a-e (and the associated new Extended Data Figures 1 and 2) now present IHC and scRNAseq analyses of human melanomas that demonstrate the clinical and translational relevance of our work. Figure 1f-k (and the associated revised Extended Data Figures 3 and 4) now present the establishment of our experimental adoptive CD4⁺ T-cell therapy strategy including the impact of the individual components of our treatment protocol on the treatment efficiency and the phenotype of the transferred and endogenous CD4⁺ T-cells in tumour tissues. Figure 1l-r (and the associated revised Extended Data Figure 5) now show that adoptively transferred CD4⁺ T-cells can indirectly recognise and kill MHC-I deficient and IFN-unresponsive melanoma cells that evade recognition and destruction by adoptively transferred CD8⁺ T-cells in the absence of NK-cells.

Revised Figure 2. CD4⁺ effector T-cells interact with MHC-II⁺ CD11c⁺ antigen-presenting cells within the tumour invasive margins.

Figure 2d-f (and the associated revised Extended Data Figure 6) now additionally present the results of several new experiments performed with MHC-deficient and IFN-unresponsive tumours (including a second model antigen system), that demonstrate how the loss of MHC-I and -II expression dramatically alters the spatial distribution and migratory behaviour of CD8⁺ T-cells, but only marginally affects that of CD4⁺ T-cells. Figure 2j-l shows another new intravital microscopy experiment, confirming that the interaction of CD4⁺ T cells with CD11c-Venus cells is MHC-II-dependent by using an MHC-II-blocking antibody.

Revised Figure 3. CD4⁺ effector T-cells and innate immune stimulation promote the infiltration of IFN-activated inflammatory monocytes and synergise to eradicate established tumours.

Figure 3a-f (and the associated revised Extended Data Figure 9) now presents our scRNAseq and flow cytometry analyses that reveal the major differentiation trajectories of inflammatory monocytes in tumour tissues of ACT-treated mice towards IFN-activated antigen-presenting and tumouricidal effector phenotypes.

Figure 3g-l (and the associated revised Extended Data Figure 10) show the impact of innate immune stimuli and CD4⁺ T-cell-derived IFN γ on the activation of tumour infiltrating inflammatory monocytes (and now also on neutrophils) and on the treatment efficacy of established tumours (previously shown in Fig. 4).

Revised Figure 4. CD4⁺ effector T-cells cooperate with IFN-activated mononuclear phagocytes to orchestrate indirect recognition and inflammatory killing of IFN-unresponsive, MHC-deficient tumours at a distance.

Revised Figure 4 (and the associated revised Extended Data Figures 11 and 12) now focuses on the results of our experiments that show how CD4⁺ effector T-cells and IFN-activated mononuclear phagocytes cooperate to enable indirect recognition and inflammatory killing of MHC-deficient and IFN-unresponsive tumours at a distance. We now also include a graphical illustration of this mechanism that acts independently and in parallel to direct recognition and cytolytic destruction of tumour cells.

New Extended Data Figure 1. Landscapes of MHC expression in human melanoma metastases and distribution of tumour-infiltrating CD8⁺ T-cells.

New Extended Data Figure 1 presents our greatly expanded immunohistochemical analyses of human melanoma skin metastases, demonstrating that MHC-I expression is a frequent event in tumour evolution, which is associated with the absence of tumour-infiltrating CD8⁺ T-cells.

New Extended Data Figure 2. Single cell RNA-sequencing analysis of human melanoma metastases and response to immune checkpoint blockade.

New Extended Data Figure 2 presents single cells RNA-sequencing data of human melanoma metastases, confirming that MHC-I downregulation on melanoma cells is a frequent event in tumour evolution that is associated with resistance to immune checkpoint blockade. The single cell RNA-sequencing data further confirm that melanoma cells mostly lack MHC-II expression, which is restricted to antigen-presenting cells immune cells.

New Extended Data Figure 3. Establishment of an experimentally tractable adoptive cell transfer model to compare CD4⁺ and CD8⁺ T-cell effector functions against tumours.

This Figure presents results of several new experiments that show the differential dynamic expansion of adoptively transferred CD4⁺ and CD8⁺ T-cells (a,b), as well as the impact of the different ACT therapy components on tumour growth (e,f), on the composition of the tumour immune composition (g,h), and on the phenotype of tumour-infiltrating CD4⁺ T-cells (i).

Revised Extended Data Figure 4 (formerly ED2). CD4⁺ effector T-cells indirectly recognise and eradicate established MHC-II-deficient melanomas through antigen presentation on MHC-II⁺ tumour-infiltrating immune cells.

This Figure additionally shows results of a new experiment where mixtures of fluorescently labelled HCmel12 CRISPR-ctrl and Trp1-KO melanoma cells were treated with CD4 ACT to evaluate bystander killing (i-k).

Revised Extended Data Figure 5 (formerly ED1). Comparative evaluation of CD4⁺ and CD8⁺ T-cell effector functions against IFN-unresponsive tumours lacking MHC-I and MHC-II.

This Figure additionally presents results of new flow cytometric analyses of the immune landscape of IFN-responsive HCmel12 CRISPR-ctrl and MHC-deficient, IFN-unresponsive Jak1-KO melanomas following treatment with CD4 or CD8 ACT (d,e). It concludes with new experimental results that show the ability of ovalbumin-specific CD4⁺ effector T-cells to eradicate ovalbumin-expressing HCmel12 Jak1-KO Tyr-KO tag-BFP melanomas that evade recognition and destruction by ovalbumin-specific CD8⁺ cytolytic T-cells (f-l), confirming our results in a second independent tumour antigen model.

Revised Extended Data Figure 6 (formerly ED3). CD4⁺ effector T-cells show a different spatial distribution and migratory behaviour in tumour tissues when compared to CD8⁺ effector T-cells.

This Figure shows results of additional experiments with mice bearing MHC-deficient HCmel12 Jak1-KO Tyr-KO tag-BFP melanomas that were treated with equal numbers of 500,000 transferred CD4⁺ and CD8⁺ T-cells (a,c,e). Furthermore, it presents results of additional experiments with mice bearing ovalbumin-expressing HCmel12 Jak1-KO Tyr-KO tag-BFP melanomas that were treated ovalbumin-specific CD4⁺ and CD8⁺ T-cells as an independent tumour antigen model (f,g).

Revised Extended Data Figure 7 (formerly ED4). CD4⁺ effector T-cells cluster with MHC-II-expressing CD11c⁺ immune cells at the invasive tumour margin.

This figure received minor adjustments to improve comprehensibility.

New Extended Data Figure 8. CD4⁺ effector T-cells cluster with MHC-II⁺ dendritic cells and monocytes at the invasive margin of a human melanoma skin metastasis.

The new Extended Data Figure 8 contains a representative multiplex immunofluorescence microscopic image providing evidence that CD4⁺ effector T-cells also cluster with MHC-II⁺ dendritic antigen-presenting cells and monocytes in a human melanoma skin metastasis.

Revised Extended Data Figure 9 (formerly ED5 and ED6). Tumour-infiltrating inflammatory monocytes acquire IFN-activated effector phenotypes upon CD4 ACT therapy

This Figure now begins with the single cell RNA-sequencing data (a-f) and now shows a Pseudotime analysis. The transcriptional differentiation trajectories were confirmed by the flow cytometric analyses of the tumour immune microenvironment 2, 5 and 8 days after CD4 ACT. To provide a comprehensive view of the immune landscape we re-analysed the flow cytometry and now show t-SNE plots (g,h).

Revised Extended Data Figure 10 (formerly ED7). Robust IFNg-dependent eradication of established melanomas requires local adjuvant innate immune stimulation.

This figure now additionally contains iNOS-expression data of neutrophils (b,g) and presents the results of a new CD4 ACT treatment experiment that addresses the impact of local versus distal injections of immunostimulatory oligonucleotides (d,e).

Revised Extended Data Figure 11 (formerly ED8). IFNg-induced nitric oxide production by myeloid cells is essential for indirect recognition and destruction of established IFN-unresponsive, MHC-deficient melanomas by CD4+ effector T-cells.

This figure now presents the results of another CD4 ACT treatment experiment that includes the in vivo antibody-mediated depletion of CCR2-expressing monocytes and Ly6G-expressing neutrophils (c-e).

Revised Extended Data Figure 12 (formerly ED9). Nitric oxide complements the ability of IFNg to sensitise melanoma cells for TNF-induced cell death, enabling efficient killing of IFN-unresponsive melanoma cells.

This figure received minor adjustments to improve comprehensibility.

Revised Extended Data Figure 13 (formerly ED10). Spatial organisation and dynamics of direct and indirect tumour cell recognition and delivery of T-cell effector functions.

This graphical summary figure was updated and now also includes neutrophils.

New List of Supplementary Videos

Video 1 (Formerly Video 2) – TRP-1 CD4+ T cells arrest in both the tumour and the stroma of HCmel12 CRISPR-ctrl tumours

Video 2 (Formerly Video 1) – Pmel-1 CD8+ T cells arrest in proximity to HCmel12 CRISPR-ctrl tumour cells

Video 3 (New) – TRP-1 CD4+ T cells arrest in both the tumour and the stroma of HCmel12 Jak1-KO tumours

Video 4 (New) – Pmel-1 CD8+ T cell do not arrest in proximity to HCmel12 Jak1-KO tumour cells

Video 5 (Formerly Video 3) – TRP-1 CD4+ T cells arrest in contact to CD11c+ immune cells at the invasive tumour margin in an antigen-dependent manner

Video 6 (New) – TRP-1 CD4+ T cells arrest in contact to CD11c+ immune cells which is dependent on MHC-II

Reviewer Reports on the First Revision:

Referees' comments:

Referee #1 (Remarks to the Author):

The revised manuscript by Thomas Tüting has been significantly improved. Not only have they carried out a substantial amount of experiments as requested by the reviewers, but they also restructured the paper and improved/ replaced a number of Figures. In particular the new Figure 1 is very helpful as it also links to the relevance of their work for patients.

New added experiments show that indeed the full ACT protocol is required to eradicate established tumours.

It is also intriguing that the ability of such a small number of CD4+ T-cells are capable to eradicate established tumours as effective as a much larger number of CD8+ T-cells and that this is not related to the cell numbers transferred by ACT.

The new histochemical analyses support the clinical relevance of their findings, and it is interesting to see that also in patients CD4+ effector T-cells accumulate at invasive margins where they can be found to cluster with MHC-II+ myeloid cells. This adds to the findings that CD8+ and CD4+ T-cells clearly differ in their spatial distribution.

Their findings that CD4+ T-cells and innate immune stimulation can reprogram the tumor microenvironment and induce tumouricidal, are sufficiently substantiated by the additional experiments and may have direct consequences for the development of novel immunotherapies, especially in patients that do not respond to ICB, lack MHC on their tumor cells and with only very few CD8+ cells present in the tumor.

Referee #2 (Remarks to the Author):

The authors have added substantially to the initial submission and thoughtfully addressed the major comments raised. The re-ordering of some data, addition of schematics (esp Figure 4g) and the requested clarifications of models throughout is a great improvement for the reader. Great efforts were made to demonstrate these findings in patient tissue, and extended data in Figures 1, 2 & 8 highlight the clinical relevance, adding further impact to this already impressive manuscript.

Referee #3 (Remarks to the Author):

We thank the authors for their efforts in providing additional data to address the points raised in the original review. I particularly appreciate that the authors provided a second tumor model that substantiate their original observation with the Trp1/Pmel model.

We understand that the possibility that the authors may have challenges to acquire iNOS KO mice although they are commercially available. Since iNOS is proposed as a central mechanism by which monocytes eliminate IFN resistant tumor clones, I suggest that they tone down the claim of a nitric oxide-dependent mechanism. The in vitro data presented, while supportive, is not entirely convincing. The authors should include language in a revised version of the manuscript concerning the clinical relevance of their findings. Finally, in response to my concern regarding the transfer of pure population while I agree that they are using Trp1 TCR tg mice, they are not transferring a pure T cell population. They are also transferring other cells (Other than mature T cells). This should be noted.

Author Rebuttals to First Revision:

Referee #1:

The revised manuscript by Thomas Tüting has been significantly improved. Not only have they carried out a substantial amount of experiments as requested by the reviewers, but they also restructured the paper and improved/ replaced a number of Figures. In particular the new Figure 1 is very helpful as it also links to the relevance of their work for patients.

New added experiments show that indeed the full ACT protocol is required to eradicate established tumours.

It is also intriguing that the ability of such a small number of CD4+ T-cells are capable to eradicate established tumours as effective as a much larger number of CD8+ T-cells and that this is not related to the cell numbers transferred by ACT.

The new histochemical analyses support the clinical relevance of their findings, and it is interesting to see that also in patients CD4+ effector T-cells accumulate at invasive margins where they can be found to cluster with MHC-II+ myeloid cells. This adds to the findings that CD8+ and CD4+ T-cells clearly differ in their spatial distribution.

Their findings that CD4+ T-cells and innate immune stimulation can reprogram the tumor microenvironment and induce tumoricidal, are sufficiently substantiated by the additional experiments and may have direct consequences for the development of novel immunotherapies, especially in patients that do not respond to ICB, lack MHC on their tumor cells and with only very few CD8+ cells present in the tumor.

Referee #2:

The authors have added substantially to the initial submission and thoughtfully addressed the major comments raised. The re-ordering of some data, addition of schematics (esp Figure 4g) and the requested clarifications of models throughout is a great improvement for the reader. Great efforts were made to demonstrate these findings in patient tissue, and extended data in Figures 1, 2 & 8 highlight the clinical relevance, adding further impact to this already impressive manuscript.

Response to referees #1 and #2

We thank the referees #1 and #2 for their careful examination and positive evaluation of our work.

Referee #3:

We thank the authors for their efforts in providing additional data to address the points raised in the original review. I particularly appreciate that the authors provided a second tumor model that substantiate their original observation with the Trp1/Pmel model.

We understand that the possibility that the authors may have challenges to acquire iNOS KO mice although they are commercially available. Since iNOS is proposed as a central mechanism by which monocytes eliminate IFN resistant tumor clones, I suggest that they tone down the claim of a nitric oxide-dependent mechanism. The in vitro data presented, while supportive, is not entirely convincing. The authors should include language in a revised version of the manuscript concerning the clinical relevance of their findings. Finally, in response to my concern regarding the transfer of pure population while I agree that they are using Trp1 TCR tg mice, they are not transferring a pure T cell population. They are also transferring other cells (Other than mature T cells). This should be noted.

Response to referee #3:

We thank the referee for the constructive comments. In the revised version of our manuscript, we took great care to tone down the claim of a nitric oxide-dependent mechanism in the summary, the results, the figures, and the discussion.

In the summary, we replaced:

“Together with innate immune stimulation, CD4+ T-cells reprogram the tumour-associated monocyte network towards IFN-activated antigen-presenting and tumouricidal effector phenotypes. This results in an amplification loop in which T-cell-derived IFN γ drives the release of myeloid cell-derived nitric oxide to eradicate IFN-unresponsive, MHC-deficient tumour cells.”

with:

“We show that Th1-directed CD4+ T-cells and innate immune stimulation reprogram the tumour-associated myeloid cell network towards IFN-activated antigen-presenting and iNOS-expressing tumouricidal effector phenotypes. Together, CD4+ T-cells and tumouricidal myeloid cells orchestrate a remote inflammatory cell death process that indirectly eradicates IFN-unresponsive and MHC-deficient tumours.”

In the results section (Figure 4), we replaced:

“Together, our results provide evidence that the ability of adoptively transferred CD4+ T-cells to indirectly kill IFN-unresponsive, MHC-deficient tumour cells involved the remote action of nitric oxide released by IFN-activated tumouricidal myeloid cells.”

with:

“In aggregate, these results suggested that the ability of adoptively transferred CD4+ T-cells to indirectly eradicate IFN-unresponsive, MHC-deficient tumour cells involved the remote action of nitric oxide released by IFN-activated tumouricidal myeloid cells.”

We adapted the discussion in a similar manner.

In addition, we emphasised the clinical relevance of our findings.

We also included the statement “Unless otherwise indicated, we transferred splenocytes containing 5×10^5 antigen-specific T-cells” in the methods section to clarify that we do not transfer pure T-cell populations in our ACT protocols.